



# New submodel for emissions from Explosive Volcanic ERuptions (EVER v1.1) within the Modular Earth Submodel System (MESSy, version 2.55.1)

Matthias Kohl[1], Christoph Brühl[1], Jennifer Schallock[1], Holger Tost[2], Patrick Jöckel[3], Adrian Jost[4], Steffen Beirle[4], Michael Höpfner[5], and Andrea Pozzer[1,6]

[1]Atmospheric Chemistry Department, Max Planck Institute for Chemistry, Mainz, Germany
[2]Institute for Physics of the Atmosphere, Johannes Gutenberg University, Mainz, Germany
[3]Institut für Physik der Atmosphäre, Deutsches Zentrum für Luft- und Raumfahrt (DLR), Oberpfaffenhofen, Germany
[4]Satellite Remote Sensing, Max-Planck-Institut für Chemie, Mainz, Germany
[5]Institute of Meteorology and Climate Research, Karlsruhe Institute of Technology, Karlsruhe, Germany
[6]Climate and Atmosphere Research Center, The Cyprus Institute, Nicosia, Cyprus

**Correspondence:** Matthias Kohl (m.kohl@mpic.de)

**Abstract.** We present a methodological study to document the operation of a new submodel for tracer emissions from Explosive Volcanic ERuptions (EVER v1.1), developed within the Modular Earth Submodel System (MESSy, version 2.55.1). EVER calculates additional tendencies of gaseous and aerosol tracers based on volcanic emission source parameters, aligned to specific sequences of volcanic eruptions. It allows for the mapping of size-resolved volcanic ash to number and mass of different size modes, and the employment of various vertical emission profiles. The new submodel is evaluated in atmospheric simulations with the ECHAM/MESSy Atmospheric Model (EMAC) coupling the general circulation model ECHAM5 to EVER and other MESSy submodels, using satellite observations of $SO_2$ column amounts and mixing ratios as well as aerosol optical properties following the explosive eruption of the Nabro volcano (Eritrea) in 2011. Sensitivity studies explore perturbations of the emission source parameters, such as plume location, emitted mass, plume altitude, vertical distribution, and timing of the emission. We integrate information from a volcanic $SO_2$ emission inventory, additional satellite observations, and our findings from the sensitivity studies to establish a historical standard setup for volcanic eruptions impacting stratospheric $SO_2$ from 1990 to 2023. We advocate for this to be a standardized setup in all simulations within the MESSy framework concentrating on the upper troposphere and stratosphere in this period. Additionally, we demonstrate the applicability of the new submodel for the simulation of degassing volcanoes, with further potential applications in studies on volcanic ash, wildfires, solar geoengineering, and atmospheric transport processes.

## 1 Introduction

Volcanic eruptions strongly impact atmospheric chemistry, climate dynamics, and air pollution. The most explosive eruptions reach the upper troposphere and stratosphere, carrying primarily emitted particles (mostly volcanic ash) and volcanic gases, that eventually lead to the formation and growth of aerosol particles. The resulting additional stratospheric aerosol loading





exerts a substantial negative radiative forcing (Schallock et al., 2023; Schmidt et al., 2018) on the one hand, and serves as surfaces for heterogeneous reactions on the other hand, thus, impacting stratospheric composition in general and ozone in particular (Klobas et al., 2017; Tie and Brasseur, 1995). Moreover, in the troposphere, gaseous and particulate emissions from volcanic eruptions can affect the environment and public health via inhalation or acid rain (Durand and Grattan, 2001; Stewart et al., 2022).

The composition of volcanic plumes exhibits considerable variability and depends on the intricate mixture of chemical species in the magma. Although emitted gases are typically dominated by water vapor and carbon dioxide, the respective total amount is mostly negligible in comparison to global emissions and concentrations (Textor et al., 2004). Notable exceptions arise during eruptions of submarine volcanoes that can inject substantial amounts of water vapor into the stratosphere, thereby impacting climate, atmospheric dynamics, and radiative forcing. A recent and notable example is the eruption of the Hunga
Tonga-Hunga Ha'apai volcano in January 2022 (e. g. Vömel et al., 2022; Sellitto et al., 2022; Schoeberl et al., 2022; Xu et al., 2022).

However, in general, the third most abundant species in volcanic plumes, sulfur dioxide ($SO_2$) along with its precursor gases hydrogen sulfide ($H_2S$), carbonyl sulfide ($OCS$) and carbon disulfide ($CS_2$), exerts the most significant long-term impact on atmospheric chemistry and climate. The release of $SO_2$ can result in a substantial enrichment of atmospheric $SO_2$ levels, with
total yearly volcanic emissions spanning a range of $1 - 50 \, \text{Tg}$ with outliers due to exceptionally strong volcanic events (Textor et al., 2004). $SO_2$ plays a pivotal role in air pollution and the formation of acid rain close to the surface (Stewart et al., 2022). Moreover, $SO_2$ undergoes oxidation to form sulfuric acid ($H_2SO_4$), which rapidly converts to the aerosol phase forming sulfate under most atmospheric conditions.

The (global) impact of volcanic eruptions strongly depends on the strength of the eruption and its geographical location.
Emissions from degassing volcanoes and smaller eruptions, failing to reach the stratosphere, primarily influence the local environment, as emitted species and their products are usually removed in the troposphere within weeks. However, they can exert a strong risk for public health and even influence the short-term climate by altering cloud properties (e. g. the 2014-2015 fissure eruption in the Holuhraun vent of Bardarbunga, Iceland, leading to a global-mean radiative forcing of -0.2 $\text{Wm}^{-2}$; Malavelle et al., 2017).

When volcanic $SO_2$ emissions reach the stratosphere, the subsequently formed sulfate aerosols enhance the stratospheric aerosol burden and are distributed widely across the globe. The lifetime of stratospheric aerosols can reach up to more than 2 years, when injected in the tropics, leading to sustained impacts on atmospheric radiative balance and climate dynamics. The strongest eruption in recent times, Pinatubo in 1991, strongly increased the stratospheric Aerosol Optical Depth (sAOD) and resulted in radiative forcing of $-3$ to $-5 \, \text{Wm}^{-2}$ (Schallock et al., 2023; Schmidt et al., 2018). Remarkably, this magnitude of
radiative forcing is comparable to the positive anthropogenic radiative forcing attributed to greenhouse gas emissions (IPCC, 2021), underscoring the considerable influence of volcanic eruptions on global climate. Medium and small explosive eruptions contribute considerably to the "Junge" layer around 25 km altitude dominated by sulfate aerosol (Junge et al., 1961), while in periods of low volcanic activity a large contribution originates from oxidation of biogenic and anthropogenic OCS (Crutzen, 1976; Brühl et al., 2012).





Furthermore, the augmented aerosol surface area in the stratosphere enhances heterogeneous reactions. Most prominently, this impacts catalytic reactions, leading to depletion or enhancement of the stratospheric ozone layer, depending on atmospheric chlorine concentrations (Klobas et al., 2017; Tie and Brasseur, 1995). Eruptions in the recent past (including Pinatubo) have led to a depletion of ozone owing to the activation of chlorine species on the surfaces of volcanic aerosols. However, as atmospheric chlorine concentrations decrease, the increased aerosol surfaces likely exert a positive influence on the ozone

layer. This anticipated shift arises because, at sufficiently low halogen concentrations, the effect of removing ozone-destroying nitrogen oxides (NOx) species on aerosol surfaces can outweigh the catalytic ozone depletion effect associated with chlorine species (Klobas et al., 2017), depending on latitude and top altitude of the volcanic plume.

    In addition to gaseous emissions, volcanic eruptions release varying amounts of primary aerosols, mainly consisting of volcanic ash, directly into the atmosphere. These ash clouds pose severe hazards to aviation and affect public health and the

environment, as they deposit on the Earth's surface. While the lifetime of ash particles in the atmosphere is relatively short, resulting in mostly negligible climate effects, recent studies have indicated that ash can persist in the atmosphere for longer durations than previously anticipated (Vernier et al., 2016). Furthermore, ash particles can interact with other atmospheric constituents, such as $SO_2$ and $H_2SO_4$, thereby influencing atmospheric chemistry (Zhu et al., 2020) and reducing new particle formation due to the preferred condensation of sulfuric acid on existing ash particles.

The understanding of the aforementioned impacts of volcanic eruptions on climate and atmospheric chemistry heavily relies on atmospheric numerical modeling. Numerical simulations can study the impact of volcanoes on the radiative balance, atmospheric chemistry, and dynamics of the atmosphere. Furthermore, the incorporation of volcanic eruptions is indispensable for model-based studies on global atmospheric aerosol burdens, particularly in comparison to observations from satellites and aircraft campaigns. Without accurate accounting for volcanic eruptions, models may underestimate upper tropospheric and

stratospheric sulfate concentrations, thereby compromising the accuracy of simulated atmospheric aerosol distributions (e.g., Reifenberg et al., 2022).

    In global atmospheric models, the treatment of volcanic eruptions varies widely (e. g. Timmreck et al., 2018). For analyses focused on specific eruptions, mixing ratio tendencies of $SO_2$ are often added manually to existing mixing ratios at fixed points in time (e. g. Schallock et al., 2023; Brühl et al., 2018; Timmreck et al., 2018). Alternatively, longer-term analyses or studies

not centered on specific volcanic events often resort to potentially outdated climatologies (Carn et al., 2016). Another approach involves injecting point or column sources in one specific horizontal grid box (e. g. Mills et al., 2016; Schmidt et al., 2018). The objective of this study is to develop a general hybrid approach, using flexible one-dimensional emissions in altitude to facilitate volcanic studies.

    In this study, we expand the capabilities of the Modular Earth Submodel System, MESSy (version 2.55.1, Jöckel et al.,

2010), to incorporate the simulation of volcanic eruptions. Our primary aim is to provide recommendations regarding the implementation of stratospheric $SO_2$ injections from volcanic eruptions within the MESSy framework, as well as in atmospheric models more broadly. To achieve this goal, our work is structured into three essential components:

      1. **New MESSy submodel for Explosive Volcanic ERuptions (EVER; Sect. 2):** We present a novel submodel called EVER within the MESSy framework. EVER is designed to simulate gas and aerosol emissions following explosive



volcanic eruptions, allowing for flexible specification of vertical distributions over a defined period. This capability
increases the versatility of volcanic eruption simulations, enabling detailed studies of post-eruptive processes, both for
explosive and degassing volcanic events. Potential supplementary applications of EVER include simulations of volcanic
ash or water vapor dispersion, general atmospheric transport phenomena, wildfires, and solar geoengineering (Crutzen,
2006).

2. **Submodel evaluation and sensitivity studies on emission parameters: (Sect. 3):** We evaluate the EVER submodel us-
ing $SO_2$ emissions from the 2011 Nabro explosive volcanic eruption (Sect. 3.3) and a degassing event in June 2018 from
the Kilauea volcano (Sect. 3.4). In general, emissions of explosive volcanic eruptions can be introduced into simulations,
using either point sources, vertical columns, or 3D plumes. Moreover, the atmospheric composition following volcanic
eruptions depends on the exact timing, duration, and position of the $SO_2$ injection into the stratosphere. We conduct
sensitivity studies with EVER using the aforementioned approaches and evaluate them with satellite observations.

3. **Model setup for historic volcanic eruptions over the last three decades (Sect. 4):** Based on the stratospheric volcanic
$SO_2$ emission inventory developed by Schallock et al. (2023), we establish a default setup for the EVER submodel,
encompassing all listed volcanic eruptions from the past three decades (approximately 800). The emission inventory
provides emitted mass, maximum plume altitude as measured from satellites, date of initial satellite detection, and
geographical location of the volcano. Where available, observations of the IASI satellite instrument (Clarisse et al.,
2014) are used to identify the geographical location and timing of the plume entry into the stratosphere. We evaluate this
setup over the time interval from 2008 to 2011.

## 2   New MESSy submodel for Explosive Volcanic ERuptions (EVER)

The new submodel for Explosive Volcanic ERuptions (EVER) is developed as an extension to the second version of the
Modular Earth Submodel System, MESSy (version 2.55.1, Jöckel et al., 2010), which can be coupled with various basemodels,
i. e. General Circulation Models (GCM). MESSy employs strict coding structures across its submodels to ensure portability
and high flexibility in chemistry-climate simulations.

MESSy offers diverse submodels for numerically simulating atmospheric aerosols. The new EVER submodel is designed to
flexibly interface with modal aerosol submodels and was tested for the three aerosol submodels GMXe (Pringle et al., 2010),
MADE3 (Kaiser et al., 2014, 2019) and PTRAC (Jöckel et al., 2008).

In GMXe and MADE3, aerosol microphysics is represented through aerosol size distributions, which consist of interactive
lognormal modes covering the typical size spectrum of aerosols. All aerosols are assumed to be spherical particles, with each
mode's characteristics fully determined by parameters such as total mass (representing an internal mixture of contributing
species), density, number concentration, and width of the log-normal distribution. Following each simulation step, aerosols
may transfer between modes based on size changes.



In PTRAC, passive aerosol tracers can be defined. Unlike interactive aerosol submodels, the diameter and density of each aerosol tracer and mode remain fixed, and the tracers are externally mixed, meaning they do not interact with each other. Consequently, it is only necessary to track the molar mixing ratios, as number concentrations directly derive from this.

## 2.1 Submodel description

The core of EVER is based on the MESSy submodel TREXP (Jöckel et al., 2010), primarily employed for artificial tracer studies and capable of emitting point sources and linear columns of trace gases. To enable the simulation of volcanic eruptions, novel functionalities were introduced as part of the new EVER submodel, including the incorporation of different types of vertical distributions, emission of aerosol species, and seamless coupling with aerosol submodels.

An example of the Fortran95 namelist setup for the EVER submodel is illustrated in Figure 1. Global parameters, that
need to be defined, include channel and channel object names for grid mass, grid volume, and altitude above sea level for the conversion of mass to mixing ratio and the correct vertical distribution of the emissions. Additionally, the channel name of the coupled aerosol submodel has to be defined. This is only necessary, if primary aerosol emissions are considered, and will be outlined in more detail in Sect. 2.1.1.

Up to 800 volcanic eruptions or other emission points (POINT) can be simulated and controlled via the namelist. Each
volcanic eruption or emission is initiated using the following parameters:

- Geographical location (latitude and longitude)

- Type of vertical distribution (see Sect. 2.1.2)

- Altitude range (minimum to maximum altitude) [km asl]

- Midpoint [km asl] and width [km] of the eruption plume (only needed for Gaussian vertical distributions)

- Period of the (volcanic) emission (start and end date)

- List of emitted tracers (corresponding to tracer names)

- List of tracer masses (in Tg)

- List of aerosol parameter sets (the name of the respective AER container, see Sect. 2.1.1)

The tracer-related lists (emitted tracer, tracer masses and aerosol parameter sets) must contain the same number of items to
ensure proper emissions. Each tracer mass and aerosol parameter set is linked to a specific tracer. Aerosol parameter sets only need to be provided for aerosol species, and are defined as described in Sect. 2.1.1. In the namelist setup depicted in Fig. 1, both $SO_2$ and volcanic ash are emitted using the same vertical distribution and time frame. If the temporal or spatial extents differ, new emission points must be defined accordingly.

Emitted tracer masses are converted to mass mixing ratio tendencies $c_{trac}$ per second $s$:

$$\frac{c_{trac}}{s} = \frac{1}{dt} \cdot \frac{M_{trac}}{M_{mol,trac}} \cdot \frac{M_{mol,air}}{M_{grid}} \tag{1}$$



```
&CPL
!
! ### Channel name of aerosol submodel for mode sigma (empty if only gas phase species)
aermod_channel = 'gmxe_gp', ! tested for GMXe ('gmxe_gp') and MADE3 ('made3_gp')
!
inp_grmassdry    = 'grid_def', 'grmass', ! box properties: grid mass
inp_grvol        = 'grid_def', 'grvol',  !                 grid volume
inp_altitudei_msl = 'grid_def', 'altitudei_msl' ! height from main sea-level
!
! ### LIST OF AEROSOL EMISSION PARAMETERS SETS
! SYNTAX:
!   'parameter-set-name', density[g/cm3], median diameter[m], aer submodel mode[1]
!
AER(1) = 'Ash_4um', 2.5, 4e-06, 7,
!
! ### LIST OF VOLCANIC ERUPTIONS / EMISSION POINTS
! ### SHAPE:    = 1 : Uniform vertical distribution
! ###           = 2 : Gaussian vertical distribution
! ### (MEAN ALTITUDE and SIGMA only needed for Gaussian vertical distribution (SHAPE=2))
! SYNTAX:
!   LON [-180 ... 180], LAT [-90 ... 90]
!   SHAPE, MIN ALTITUDE [km], MAX ALTITUDE [km], MEAN ALTITUDE [km], SIGMA VERTICAL [km],
!   YYYY, MM, DD, HH, MI, SE, YYYY, MM, DD, HH, MI, SE,
!   |=====================|   |=====================|
!            START                     STOP
!   ';-sep. tracers',';-sep. tracer masses [Tg]',';-sep. aerosol param. sets (optional)'
!
! NABRO ERUPTION
POINT(1) = 29.74,22.85,
           2,16,18,17,2,
           2011,6,13,16,00,00,2011,6,14,22,00,00,
           'SO2;ASH_ci','0.406;0.1',';Ash_4um',
/
```

**Figure 1.** Example Fortran95 namelist for the new submodel EVER v1.1, emitting $SO_2$ and volcanic ash from the Nabro volcano (2011) in a Gaussian vertical distribution at the geographical location, where the $SO_2$ plume entered the stratosphere (refer to Sect. 3.3 for details).





The variables $M_{trac}$ and $M_{grid}$ represent the masses of the emitted species and dry air in the grid box, respectively. $M_{mol,trac}$ and $M_{mol,air}$ denote the molar mass of the tracer and of dry air, respectively, while $dt$ indicates the total emission period from the start to the end date in seconds. During runtime, the tracer mixing ratio tendencies are incrementally added to the total mixing ratios, uniformly distributed over the specified time range, within the horizontal grid cell containing the defined

emission point and following a user-specified vertical distribution. It is important to note that while all model tracers can be used in general, it is not possible to define new tracers within the EVER submodel, in contrast to the TREXP submodel.

### 2.1.1 Primary emissions

In the aerosol submodels GMXe and MADE3, emissions of primary aerosols are characterized by an increase in the mixing ratio of the corresponding tracer and an increase in number concentration of the corresponding aerosol size mode. For these

cases, the aerosol submodel channel name ("aermod_channel" in namelist) has to be provided.

The emitted number concentration is calculated based on aerosol parameter sets (the AER container), which can be defined via Fortran95 namelist using the following variables:

- Aerosol parameter set name (referenced by the volcanic eruption points)

- Density of the emitted aerosols $\rho$

- Median emission particle diameter of the emitted species $d_{md}$ [m]

- Aerosol submodel mode for the sigma of the log-normal distribution $\sigma_{ln}$

The median emission particle diameter should reside within the diameter boundaries of the corresponding mode to ensure proper treatment of the emission. The sigma of the log-normal distribution will be extracted from the previously defined aerosol submodel for the given mode.

In EVER v1.0, which is included in the latest MESSy releases 2.55.1 and 2.55.2, primary emissions were handled differently. Specifically, EVER could only be directly integrated with GMXe, requiring the GMXe mode as a string, while sigma had to be explicitly provided for other submodels. The improved coupling with aerosol submodels was introduced in EVER v1.1 and will be available in all subsequent releases.

On the basis of the aerosol parameter sets, number concentrations are calculated using

$$N_{aer} = \frac{6 \cdot M_{trac}}{\pi \cdot \rho \cdot \exp\left(3 \ln d_{md} + 4.5 \ln^2 \sigma_{ln}\right)} \qquad (2)$$

where $M_{trac}$ denotes the mass of the emitted species and $\rho$, $d_{md}$ and $\sigma_{ln}$ as described above.

In the PTRAC submodel, no aerosol parameter sets are required. If the tracer is specified as a mixing ratio in PTRAC, it is handled similarly to a gas-phase species in EVER, as there is no requirement for an increase in number concentration.



### 2.1.2 Vertical distributions

As volcanic plumes typically span a range of altitudes rather than being centered at a specific altitude, it is feasible to specify vertical distributions for the corresponding emissions. Presently, two types of vertical distributions are supported, with the potential for expansion in the future:

1. **Uniform distribution:** In this distribution, the mass is uniformly distributed between the minimum and maximum altitudes, proportionally to the height of the grid cell. The lowermost and uppermost grid cells within the altitude range are
filled based on the fraction of the grid cell covered by the altitude range.

2. **Gaussian distribution:** In this distribution, the mass follows a Gaussian-shaped profile with a mean altitude and width defined in the emission namelist. The emission amount in each grid cell is calculated by considering the fraction of the error function integrated from the bottom to the top of the grid cell (for the lowermost and uppermost grid cells, the minimum and maximum altitudes are used, respectively) relative to the integral of the error function across the entire
altitude range.

Large volcanic plumes entering the stratosphere can span several horizontal grid boxes as well. Although each emission point emits within a single grid box, multiple emission points can be defined to accurately reproduce the horizontal distribution of a single eruption. However, in this study we only used one emission point for each eruption.

## 3 Evaluation and sensitivity studies

### 3.1 Model setup

The ECHAM5/MESSy Atmospheric Chemistry (EMAC) model (Jöckel et al., 2006) couples MESSy2 (Jöckel et al., 2010) to the general circulation model ECHAM5 (version 5.3.02, Roeckner et al., 2003). For simulations of explosive volcanic eruptions, we employ a spectral, horizontal resolution of T63 ($1.875° \times 1.875°$) with 90 vertical levels, extending up to $0.1$ hPa altitude. Conversely, the study on degassing volcanoes (Sect. 3.4) is conducted at a horizontal resolution of T255 ($\sim$
$0.47° \times 0.47°$) with 31 vertical levels up to only 10hPa. Model simulations are "nudged" (Jeuken et al., 1996; Jöckel et al., 2006) towards meteorological reanalysis data (ERA5, Hersbach et al., 2020) from the European Centre for Medium-Range weather forecasts (ECMWF).

The newly developed EVER submodel is coupled to GMXe (Pringle et al., 2010), as described before. In our simulations, we apply four hydrophilic modes (Nucleation, Aitken, Accumulation, and Coarse) and three hydrophobic modes (Aitken, Accu-
mulation, and Coarse). Sulfuric acid-water nucleation follows the parameterization by Vehkamäki et al. (2002). Sedimentation, dry and wet deposition are simulated using the submodels SEDI, DDEP (both Kerkweg et al., 2006), and SCAV (Tost et al., 2006), respectively.

Gas-phase chemistry is addressed by the MECCA submodel (Sander et al., 2019), employing the Mainz Isoprene Mechanism (MIM1; Pöschl et al., 2000; Jöckel et al., 2006), while we use a simplified chemistry for the Kilauea studies (Sect. 3.4) due to



the high horizontal resolution. Emissions from anthropogenic and biogenic sources, and from biomass burning are introduced as described in Kohl et al. (2023). Carbonyl sulfide (OCS), a precursor for stratospheric aerosols, is constrained using monthly averaged surface concentrations as outlined by Montzka et al. (2007). Dimethyl sulfide (DMS) emissions from the ocean are computed using the MESSy submodel AIRSEA (Pozzer et al., 2006) and global ocean surface DMS concentrations derived by Lana et al. (2011) in the stratospheric setup. In the simplified degassing setup, we do not consider DMS and OCS (see

supplement), leading to a potential underestimation of background maritime $SO_2$ concentrations.

We calculate AERosol OPTical properties of the GMXe aerosol populations with the AEROPT submodel (Dietmüller et al., 2016), providing extinction coefficients and Aerosol Optical Depth (AOD) at various wavelengths. This information is sampled along the sun-synchronous satellite orbits using the submodel SORBIT (Jöckel et al., 2010) for comparison with satellite observations.

Namelist setup, chemical mechanism and runscript for the stratospheric simulation (*nml_strat*, *meccanism_strat.pdf* and *xmessy_mmd.stratVolc*, respectively) and the Kilauea simulation (*nml_Kilauea*, *meccanism_Kilauea.pdf* and *xmessy_mmd.Kilauea_T255*, respectively) can be found in the supplement.

### 3.2 Observations

$SO_2$ and optical properties of aerosols in the atmosphere are continuously monitored by satellites. We use observations of

different volcanic eruptions for the evaluation of the new submodel. Measurement techniques and geometries of the satellites are shortly introduced in the following.

### 3.2.1 Infrared Atmospheric Sounding Interferometer (IASI):

IASI (Blumstein et al., 2004; Clerbaux et al., 2009) is an integral part of the MetOp-A meteorological payload and was launched in October 2006. Operating in nadir geometry, IASI observes the Earth's atmosphere by measuring emitted radiation in the

thermal infrared range using a Fourier Transform Spectrometer. Advanced trace gas detection techniques have been developed to derive altitude-resolved profiles (Clarisse et al., 2012, 2014) of $SO_2$, which are used for this study.

### 3.2.2 Michelson Interferometer for Passive Atmospheric Sounding (MIPAS)

MIPAS was a component of the ENVISAT satellite, operational from 2002 to 2012 (Fischer et al., 2008). It was a mid-infrared emission spectrometer designed to perform limb sounding at different tangent heights, enabling the detection of radiation

emitted by the Earth's atmosphere. This geometric configuration allowed for the retrieval of three-dimensional information about the atmosphere. MIPAS could measure various trace gases, including $SO_2$. Data from MIPAS are obtained from Level 2 and Level 3 (5-day averages) retrievals (V5R_SO2_220 & V5R_SO2_221) as described by Höpfner et al. (2015, 2013).





### 3.2.3 Ozone Monitoring Instrument (OMI)

OMI (Levelt et al., 2006) was launched aboard the Aura satellite in 2004 and remains operational to this day. OMI observes
solar backscatter radiation from the Earth's surface across the ultraviolet and visible wavelength range in nadir position. Consequently, OMI provides measurements of the total column amount of various trace gases, including $SO_2$, $O_3$ and $NO_2$, but
does not offer vertical profiles. It achieves nearly global coverage within 24 hours. For our analysis, we used Level 2 $SO_2$ data
(V003; available from Li et al., 2020).

### 3.2.4 TROPOspheric Monitoring Instrument (TROPOMI)

In 2017, TROPOMI (Veefkind et al., 2012) was launched aboard the ESA Sentinel-5 Precursor satellite to measure key tropospheric trace gas columns. The TROPOMI spectrometer inherits components of OMI but is optimized for higher horizontal
resolution measurements, achieving resolutions as fine as $3.5 \times 7 \text{ km}^2$. Additionally, it encompasses spectral bands across the
ultraviolet, visible, near-infrared and shortwave infrared regions, enabling the measurement of a wide range of trace gases, including $SO_2$. Similar to OMI, TROPOMI achieves global coverage within a single day. The $SO_2$ retrieval algorithm is detailed
by Theys et al. (2017). We use data from the official TROPOMI Level 2 $SO_2$ product (repro, v1.1) for our analysis (ESA,
2018).

### 3.2.5 Optical, Spectroscopic and Infrared Remote Imaging System (OSIRIS)

The OSIRIS limb scatter instrument, aboard the ODIN satellite, has been operational since 2001, operating on a sun-synchronous
polar orbit. It performs measurements of the vertical profile of atmospheric limb radiance spectra at wavelengths ranging from
$274 \text{ nm}$ to $810 \text{ nm}$ (Bourassa et al., 2012a). The retrieval process provides information on aerosol extinction, assuming a
refractive index of $1.427 + 7.167 \times 10^{-8}$ and an aerosol mixture composed of $75\%$ $H_2SO_4$ and $25\%$ $H_2O$. We use the aerosol
product version 7.2 in this study (Rieger et al., 2019).

### 3.2.6 Averaging kernels

Satellites typically do not directly observe the retrieved variables but rather functions of these variables, which rely on certain
a priori information (Rodgers and Connor, 2003; Rodgers, 1990; Raspollini et al., 2006). Additionally, observed values are
influenced by the horizontal and vertical resolutions used for retrieval, leading to contributions from neighboring layers. To
account for this effect, averaging kernel matrices (AKM) can be applied to model data for comparison purposes. In this analysis,
AKMs are used for comparisons with MIPAS observations. It is important to note that it is not possible to invert the AKM to
correct the retrieved product.

## 3.3 SO₂ in explosive volcanic eruptions — Nabro (2011)

On 13 June 2011, the Nabro volcano in Eritrea erupted for the first time documented in history following an earthquake swarm
on 12 June 2011 (Global Volcanism Program, 2011). The resulting volcanic cloud predominantly comprised water and $SO_2$,





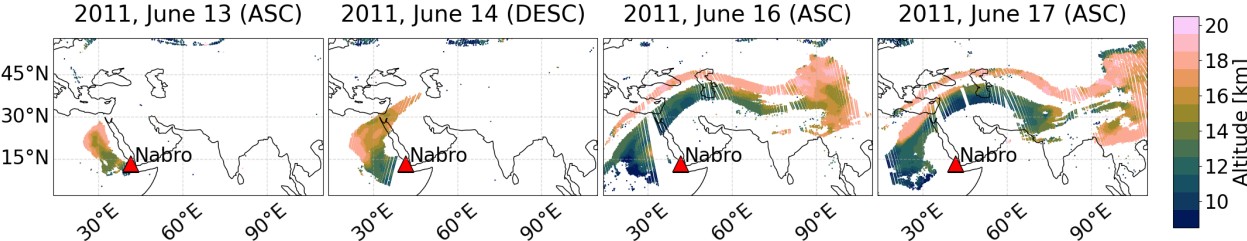

**Figure 2.** Estimated plume altitude for Nabro derived from IASI satellite observations during ascending (ASC) and descending (DESC) orbits on selected dates in the first week after the eruption. Data taken from Clarisse et al. (2014).

reaching up to 20 km in altitude. Thus, it offers a perfect case study to investigate the spatio-temporal evolution of volcanic $SO_2$ in the stratosphere. Notably, the eruption coincided with the Asian Monsoon Anticyclone (AMA), simultaneously probing the dynamics of the model.

The IASI satellite first recorded the $SO_2$ plume on 13 June 2011. Figure 2 illustrates the estimated altitude of the plume derived from the IASI satellite data on various days within the first two weeks after the initial eruption.

The initial stratospheric plume followed a northwestward trajectory during its ascent, entering the stratosphere at approximately $18°N$, $30°E$, and an altitude of up to 18 km. Within the Upper Troposphere / Lower Stratosphere (UTLS) region, the $SO_2$ plume is influenced by the AMA, subsequently evolving in a northeastward direction. In the night of June 15 to June 16, an additional plume entered the stratosphere, as evident in the IASI observations from June 16 (Fig. 2). Concurrently, a wind shear within the AMA around the tropical tropopause induced a separation between tropospheric and stratospheric $SO_2$, with the stratospheric component further north. This separation is notably pronounced in the IASI observations on 16 June 2011. Subsequently, the $SO_2$ originating from the two eruptions that reached the stratosphere mixed and was transported in the AMA. Approximately 10 days after the initial eruption, stratospheric $SO_2$ was distributed widely across the displayed region (not shown).

It is debated, if the eruption led to a direct injection of the volcanic plume into the stratosphere or if the initial plume failed to reach the stratosphere, subsequently being uplifted within the South Asian monsoon system (Bourassa et al., 2012b; Fromm et al., 2013; Vernier et al., 2013; Bourassa et al., 2013; Clarisse et al., 2014). Especially the second stratospheric plume on June 16 could comprise remnants of the tropospheric plume, that are uplifted. However, it is important to note that this study does not engage in this ongoing discussion, but exclusively concentrates on the stratospheric entrance points of the plume.

In addition to the altitude information, Clarisse et al. (2014) provide the amount of $SO_2$ within the plume, assuming that all $SO_2$ was concentrated at the derived altitude. This information is illustrated in Fig. 3 for the first week after the eruption, alongside the respective observations of the $SO_2$ column amount from the OMI satellite at the same dates. It is noteworthy that the precise timing of the satellite overpasses does not align for the IASI and OMIs satellites. $SO_2$ amounts derived from IASI are presented only for coordinates where the estimated plume altitude exceeds $14\,\mathrm{km}$, whereas OMI observes the total column amount. The difference between the stratospheric IASI observations and OMI total column amounts confirms the altitude-



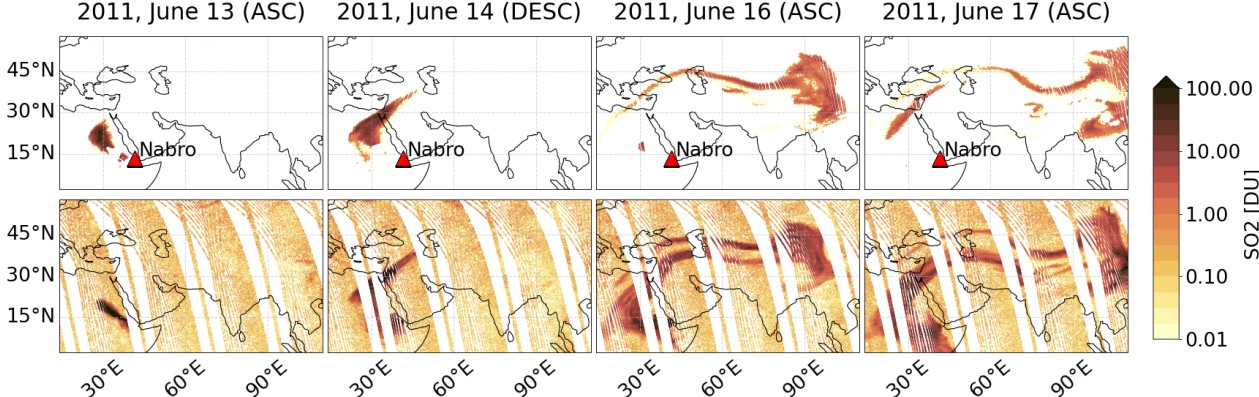

**Figure 3.** Derived SO$_2$ amount from IASI (top; Clarisse et al., 2014) and OMI (bottom; Li et al., 2020) observations. IASI SO$_2$ column amounts are calculated assuming that all SO$_2$ is centered at the plume altitude from Fig. 2 and we only display pixels where the SO$_2$ plume is detected in the stratosphere (altitude above 14 km), whereas OMI displays the total column. Note that the timing of the observations does not coincide in general.

specific findings depicted in Fig. 2, enabling a distinction between tropospheric and stratospheric contributions to the total column amount. Furthermore, it highlights that while the Nabro volcano continuously emitted, plumes entered the stratosphere

exclusively on June 13 and June 16.

To comprehensively evaluate the submodel and explore the impact of simplifications and adjustments to emission data, we conducted a reference simulation along with a series of sensitivity simulations of stratospheric SO$_2$ from the Nabro eruption. The reference simulation draws upon the emission inventory developed by Schallock et al. (2023). This inventory provides a detailed compilation of volcanic eruptions from 1990 to 2019 (extended for this study up to November 2023), along with

their geographical location, maximum plume altitude, stratospheric SO$_2$ amount and time of detection. However, it is worth noting that the location of the volcano does not coincide with the coordinates of the entrance to the stratosphere in general. This discrepancy is evident for the Nabro volcano in Fig. 2. Moreover, the detection time does not align with the actual eruption time or the moment of entry into the stratosphere.

To account for these disparities, we refined the timing and geographical location of the SO$_2$ injection, based on time and

coordinates of the maximum observed stratospheric mixing ratios derived from the IASI satellite. This analysis was performed within a spatial window of 20° x 30° in latitude and longitude around the volcano's location, considering a timeframe of 20 days around the detection time of the plume. Subsequently, the emissions were distributed across the respective model grid boxes over a 6-hour period around the time of the initial detection of the plume by IASI. This refined approach enables a more precise evaluation of the submodel's performance and a detailed exploration of the impact of perturbations in the emission

data.

The following simulations were performed:




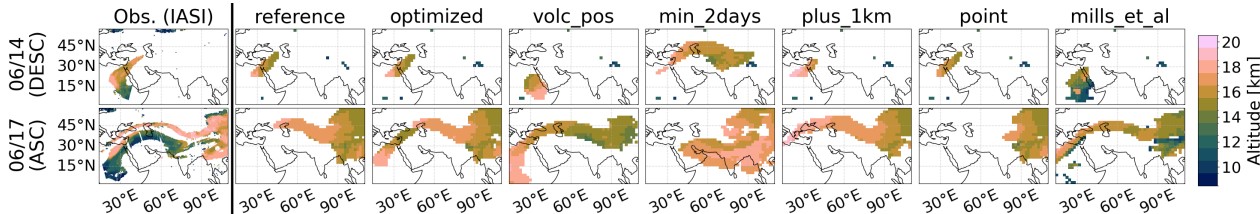

**Figure 4.** Derived plume altitude from the IASI satellite observations (left; compare Fig. 2) is compared with the altitude of maximum $SO_2$ mixing ratios from the sensitivity simulations, shortly after the eruption (top) and three days later (bottom). The injection of $SO_2$ in the simulations was only performed in the stratosphere, except for **mills_et_al**.

- **reference**: Column emission at altitude from the emission inventory minus 1 km (17 km); Gaussian vertical distribution with a width of 2 km; horizontal position and timing derived from IASI (one horizontal grid box encompassing 22.9°N, 29.7°E on June 13, 2011, 16:00 - 22:00 UTC); $SO_2$ amount (406 kt) from the emission inventory from Schallock et al. (2023).

- **optimized**: Same as **reference**, but with the total amount distributed on two stratospheric entry points (67% on June 13 and 33% on June 16 based on the qualitative findings from the IASI observations).

- **reduced**: Same as **reference**, but with reduced $SO_2$ emissions (280 kt).

- **volc_pos**: Same as **reference**, but emissions injected at the geographical location of the volcano (13°N, 41°E).

- **point**: Same as **reference**, but using only one vertical grid-box at the emission inventory altitude minus 1 km (17 km).

- **min_2days**: Same as **reference**, but emissions shifted by two days (June 11, 2011, 16:00 - 22:00 UTC).

- **plus_1km**: Same as **reference**, but emissions shifted by 1 km in altitude (18 km).

- **mills_et_al**: Emissions as described by Mills et al. (2016).

### 3.3.1 Short-term $SO_2$ plume evolution

Figures 4 and 5 present a comparison between the simulated altitude of maximum $SO_2$ mixing ratios and column $SO_2$ amounts from both, the reference and sensitivity simulations, and the corresponding IASI observations for the morning overpass on June 14 (immediately after the first eruption) and the afternoon overpass on June 17. $SO_2$ emissions in the simulation are confined to the stratosphere (except the approach by Mills et al. (2016)), and thus only stratospheric amounts (altitude $\geq$ 14 km) are shown. Indeed, this approach neglects the significant amount of tropospheric $SO_2$ injected during the eruption. However, tropospheric $SO_2$ typically has a much shorter lifetime compared to stratospheric $SO_2$, which we focus on in this study. The **reduced** simulation is not shown in these figures, as it is equivalent to the **reference** in altitude and only exhibits slightly reduced column amounts.





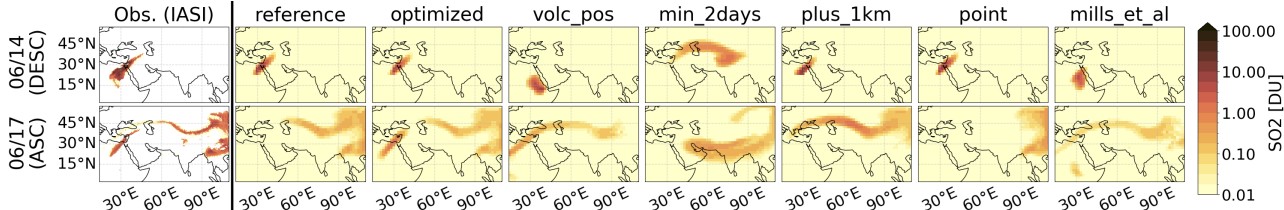

**Figure 5.** Column $SO_2$ amount derived from IASI satellite observations (left; compare Fig. 3) is compared with the respective column amount from the sensitivity simulations, shortly after the eruption (top) and three days later (bottom). The injection of $SO_2$ was only performed in the stratosphere, except for **mills_et_al**.

Overall, the simulated column amounts appear to be underestimated when compared to observations (see Fig. 5). However, this might be attributed to the retrieval procedure of the column amount from the IASI observations. The column amount

estimation assumes that all $SO_2$ of the plume is centered at the respective altitude depicted in Fig. 4. Hence, we only conduct a qualitative comparison between the simulated and observed column amounts.

The **reference** simulation appears to reasonably capture the stratospheric evolution when considering only pixels where the altitude $\geq 14$ km, although some discrepancies are evident. From Fig. 4, it seems that the simulated columns slightly broaden over time compared to the observations, with the plume appearing to sink. However, upon closer inspection of Fig. 5,

it becomes apparent that the simulated enhanced column amounts are actually narrower. The broader distribution observed in the data likely stems from simulated values that fall below the detection limit of IASI. Notably, as the **reference** simulation assumes only one stratospheric entrance, it fails to reproduce the observed second plume as expected.

In the **optimized** simulation the emissions are distributed across two space-time points, based on a detailed analysis of the IASI observations (refer to Fig. 2 and 3). As a result, the second plume is successfully reproduced, leading to better

agreement with the observations. By varying the geographical location of the stratospheric entrance (**volc_pos**), the plume encounters different meteorological patterns, resulting in a distinct evolution pattern. Similarly, adjusting the timing parameter (**min_2days**) leads to a more advanced evolution within the anticyclone. These sensitivity analyses highlight the importance of accurately representing the timing and location of stratospheric injections for capturing the short-term evolution of volcanic plumes in atmospheric models.

As previously mentioned, there exists a vertical wind shear, leading to a displacement of the stratospheric part of the plume towards higher latitudes. Moreover, there appears to be a vertical gradient in wind speed above 16 km. When the emission altitude is increased by 1 km (**plus_1km**), the evolution within the anticyclone is attenuated, suggesting lower wind speeds at higher altitudes. Additionally, the **point** simulation, where all emissions are centered at 17 km, only reproduces the rapid branch of the observed plume evolution. Consequently, it does not encounter the lower wind speeds experienced at higher altitudes.

In contrast, Mills et al. (2016) implemented emissions over several days and uniformly over altitudes ranging from 2.5 to 17 km, accounting for the continuous emissions that did not reach the stratosphere. However, in the **mills_et_al** simulation,





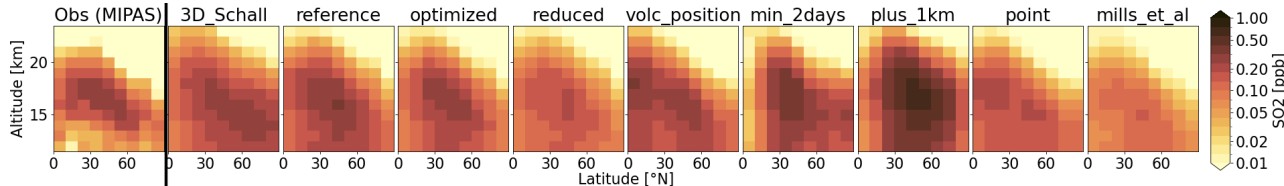

**Figure 6.** Zonal profile (5-day averaged) of northern hemispheric stratospheric $SO_2$ mixing ratios derived from the MIPAS satellite (leftmost panel) is compared with the respective $SO_2$ mixing ratios simulated in the reference and sensitivity studies, approximately one month after the eruption of the Nabro volcano (July 16, 2011).

stratospheric emissions are underestimated, leading to a substantial fraction of emissions being lost due to scavenging in the free and upper troposphere.

### 3.3.2 Long-term $SO_2$ mixing ratios

So far, we only studied the short-term evolution of the stratospheric plume, i. e. within the first week. However, IASI faces limitations in capturing the long-term evolution of volcanic plumes due to the dilution of the emitted $SO_2$, leading to column amounts that fall below the instrument's detection limit. Conversely, observations from the MIPAS satellite are not well-suited for short-term analysis as observed mixing ratios only slowly build up after strong volcanic eruptions. Höpfner et al. (2015) provide two main reasons for this behavior. First, the enhanced concentration of volcanic particles in the plume may lead to the

exclusion of the retrieved spectra, and thus non-plume air masses are favored. Second, the enhanced $SO_2$ mixing ratios saturate the spectral lines. Indeed, MIPAS observations become more reliable approximately three weeks after the initial eruption. Its ability to provide three-dimensional $SO_2$ observations makes it well-suited for long-term monitoring and understanding the evolution of volcanic plumes over extended periods.

The zonal and 5-day averaged profiles of stratospheric $SO_2$ mixing ratios on July 16, 2011, approximately one month after

the eruption, are shown in Figure 6 for the Northern Hemisphere. In addition to the sensitivity simulations discussed earlier, a simulation using 3D emission fields of $SO_2$ mixing ratios (**3D_Schall** in the following) is included for comparison (Schallock et al., 2023). These emission fields were derived from various satellite observations and applied several days after the initial eruption, specifically on June 21 for the Nabro eruption. Details on the methodology are described by Schallock et al. (2023). To simulate the effect of limited vertical resolution associated with MIPAS observations, averaging kernel matrices (AKM)

were applied to all simulated $SO_2$ mixing ratios.

The comparison between simulated and observed $SO_2$ distributions reveals some discrepancies, particularly in the vertical width of the distributions. After applying the AKM to the simulations, the $SO_2$ distributions exhibit a slightly larger vertical width compared to the observations. This discrepancy suggests a potential overestimation of the AKM or limitations in the vertical resolution of the simulation at the respective altitudes (approximately 500 meters). Interestingly, the **3D_Schall** sim-

ulation, which uses 3D emissions derived directly from satellite observations, shows the widest distribution. This discrepancy can be attributed to the fact that the 3D emissions already incorporate the smoothing effects introduced during the retrieval



process. Consequently, the simulated $SO_2$ mixing ratios in the **3D_Schall** simulation represent observed mixing ratios rather than actual mixing ratios. Therefore, after applying the AKM to these simulated mixing ratios, the resulting altitude resolution appears too wide.

The sensitivity studies exhibit consistent patterns, with the highest mixing ratios of $SO_2$ typically observed between 15 and 20 km in altitude and $20°$ and $60°$ N in latitude, and decreasing altitudes of highest $SO_2$ mixing ratios with increasing latitude. The distribution follows the typical stratospheric circulation pattern and resembles the observed distribution. Lower mixing ratios compared to observations are found in the **reduced**, **point**, and **mills_et_al** simulations. In the **reduced** simulation, the reduced emissions directly lead to lower mixing ratios. However, in the **mills_et_al** simulation, a significant portion of $SO_2$

is removed in the upper troposphere, as discussed earlier. In the **point** simulation, restricting emissions to a single grid box likely results in the accumulation and rapid growth of sulfate particles, followed by sedimentation out of the stratosphere. Conversely, the **plus_1km** simulation shows higher mixing ratios due to the increased stratospheric lifetime associated with higher injection altitudes. Interestingly, the **volc_pos** and **min_2days** simulations exhibit slightly different spatial distributions, with the former indicating higher and the latter lower $SO_2$ mixing ratios at low latitudes. Hence, the varied meteorological

conditions experienced in the initial days post-eruption consistently lead to diverse long-term evolutions of stratospheric $SO_2$.

     In addition to examining the spatial distributions at specific time points, we explored the long-term changes of stratospheric $SO_2$ mass and the long-term spatial agreement between observed and simulated $SO_2$ mixing ratios after the Nabro eruption. The top panel of Fig. 7 illustrates the total stratospheric $SO_2$ burden in the Northern Hemisphere from the sensitivity simulations compared to the observations. MIPAS observations exhibit a gradual increase following the volcanic eruption, unlike

the simulations, as discussed earlier. The **3D_Schall** simulation's onset is delayed by a week, as the simulation is based on observations after the first plume evolution in the stratosphere. The bottom panel of Fig. 7 illustrates the spatial correlation between the zonal profile of MIPAS observations and the sensitivity studies within the altitude-latitude window depicted in Fig. 6.

     The stratospheric $SO_2$ burdens of the **reference**, **optimized**, **volc_pos** and **min_2days** simulations follow the same log-

arithmic decay, mostly coinciding with the observations after mid-July. These simulations also exhibit very similar spatial correlations with some fluctuations, showing correlations around 0.9 approximately 3 weeks after the eruptions, which gradually decrease to values between 0.75 and 0.8 in late September. The **min_2days** simulation shows a weaker correlation in the initial phase but approaches the others in the long term, most likely due to the different meteorological conditions.

     The **reduced**, **point** and **mills_et_al** simulations exhibit consistently lower stratospheric $SO_2$ mass as observed earlier,

while the decay is parallel to the aforementioned. The spatial correlation of the **reduced** and **point** simulations is similar to the **reference** simulation, while the **mills_et_al** simulation shows a slightly smaller but comparable correlation. Emissions at higher altitudes (**plus_1km**) lead to higher $SO_2$ burden, longer lifetime, and lower spatial correlation with the observations. Initially, the simulation with three-dimensional emissions (**3D_Schall**) shows a comparable total stratospheric $SO_2$ burden to the reference simulation. However, it displays a slightly shorter lifetime as the burden decays more rapidly. Additionally, a

slightly lower spatial correlation is observed, potentially attributed to the wider distribution of emissions.



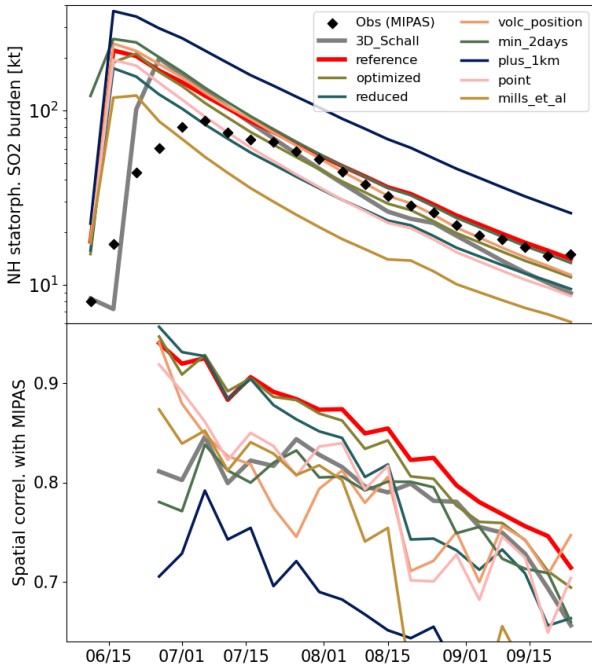

**Figure 7.** Timeline comparisons between MIPAS satellite observations and reference and sensitivity simulations. The top panel displays the total northern hemispheric stratospheric $SO_2$ burden as observed from the MIPAS satellite (black dots) and simulated (5-day averages). As stratospheric cutoff altitudes, we used 16 km from 0-30°N, 14 km from 30-60°N, and 12 km from 60-90°N. In the bottom panel, we depict the spatial correlation in the latitude-altitude plane between the simulations and MIPAS observations, i. e. the spatial correlation between the first and all other panels in Fig. 6.

The overall slightly faster decline observed in the simulation compared to the observations may be a consequence of the absence of primary particles, such as volcanic ash, in the simulations, resulting in a discrepancy between simulated and observed particle size distributions. This discrepancy appears consistent across all simulations, suggesting a systemic bias in the overall setup, regardless of emission intensity. Alternatively, the simulated particle sizes may grow excessively large too quickly, lead-
ing to an overestimation of sedimentation efficiency. Whether this discrepancy arises from nucleation rates versus condensation efficiency, the overall representation of the size distribution with only four modes, or the limitation to one horizontal grid box will be the topic of upcoming studies.

### 3.4 $SO_2$ from degassing volcanoes — Kilauea (2018)

In the following, we briefly discuss the additional EVER use case of degassing volcanoes by analyzing the emissions from the
Kilauea volcano in Hawaii, USA. Typically, $SO_2$ emissions from such volcanoes do not reach the stratosphere but impact the



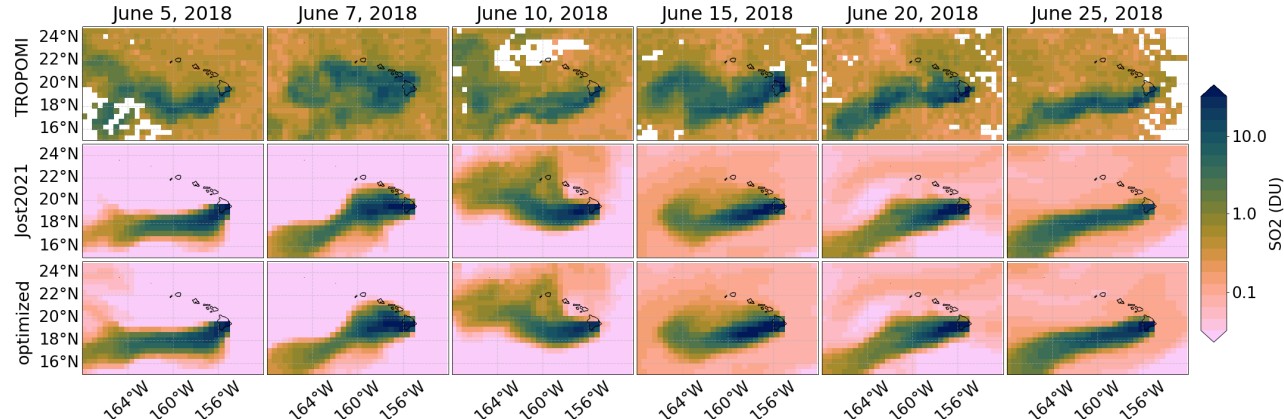

**Figure 8.** Observed (top) and simulated $SO_2$ column amounts resulting from the degassing of the Kilauea volcano at selected days in June 2018 at a model resolution of T255. The middle row represents a simulation with emission rates derived by Jost (2021, scaled by a factor of 4.3 – see text for details), the bottom row a simulation with optimized emission rates, based on a comparison between model and observations (refer to the text for more details).

atmosphere on a limited temporal and spatial scale. $SO_2$ emissions from Kilauea were studied with the EMAC model before for a different period (Beirle et al., 2014).

In summer 2018, a series of eruptive fissures opened at Kilauea, followed by intense degassing activity (Kern et al., 2020). The corresponding $SO_2$ emissions can be clearly seen from TROPOMI (ESA, 2018). These observations serve as a basis for comparison with simulated column amounts. In the top row of Figure 8, we present observations from selected days in June 2018, regridded to T255 model resolution (approximately $50 \times 50$ km at the equator). Generally, $SO_2$ column amounts are highest in proximity to the Kilauea volcano and disperse according to meteorological conditions.

Jost (2021) derived daily $SO_2$ emission rates for Kilauea based on TROPOMI observations (SO2 repro v1.1) and wind fields from ECMWF, using the divergence method developed by Beirle et al. (2019). Within this study, we use the original emission rates scaled up by a factor of 4.3 following recent updates, resulting in higher emission estimates. These adjustments include:

(a) A factor of 3.2 due to non-linear effects caused by strong $SO_2$ absorption in the Kilauea volcanic plume (compare Theys et al., 2017). While Jost (2021) initially deemed this effect negligible, we included it after re-examination.

(b) A factor of 1.12 due to the change of the assumed plume height from 2000 m to 1000 m (compare Kern et al., 2020).

(c) A factor of 1.19 from the consideration of topographic effects on the divergence calculation (Sun, 2022; Beirle et al., 2023).

We performed a simulation at T255 horizontal resolution using the derived emission rates for June 2018 and the model setup described in Section 5. $SO_2$ column amounts were sampled at the time of the satellite's overpass of the Kilauea volcano using the SORBIT submodel (Jöckel et al., 2010). Results are depicted in the middle row of Fig. 8. Qualitatively, we observe



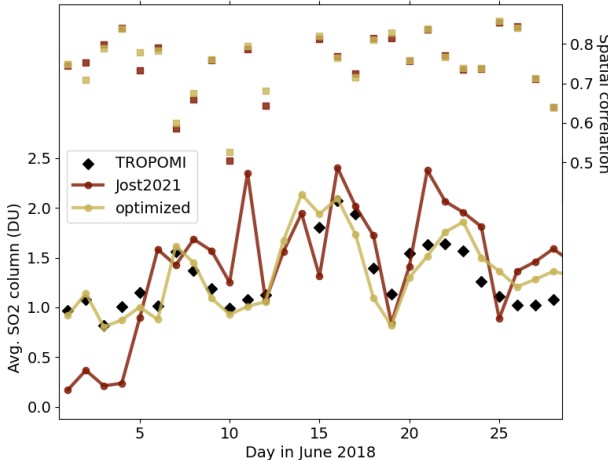

**Figure 9.** Derived spatially averaged $SO_2$ column amount from TROPOMI observations ($-168°$ to $-152°$ W, $15°$ to $25°$ N; horizontal window displayed in Fig. 8) in June 2018 compared with simulations, using emission rates from Jost (2021, scaled by a factor of 4.3 – see text for details) and optimized emission rates, based on a comparison between model and observations. On the top, we present spatial correlations in the same horizontal window between the observations and the respective simulations, using the same color as for the $SO_2$ column amount.

reasonable agreement in the horizontal dispersion of the plume on most days (Fig. 8), with some exceptions, such as June 7 and
June 10. However, there is stronger variability in the observations within the transported plume compared to the simulations, where we predominantly observe gradually decreasing $SO_2$ columns with distance from the volcano. This effect is attributed to the diurnal variability of the $SO_2$ emissions, which are not represented in the emission dataset.

Figure 9 provides a quantitative assessment of the horizontal extent depicted in Figure 8 (spanning from $-168°$ to $-152°$ W, $15°$ to $25°$ N). The lower graph illustrates the spatially averaged $SO_2$ column amount within this horizontal window,
$\overline{SO_{2(col,d)}}$, as observed and simulated at each day in June 2018 $d$. Additionally, the spatial correlation between simulated and observed logarithmic $SO_2$ column amounts within this window is presented in the upper graph of Figure 9.

The simulation exhibits some noticeable fluctuations in $\overline{SO_{2(col,d)}}$, with periods of underestimation in the initial days and on June 25, as well as instances of significant overestimation, such as on June 11 and June 21. The presence of a low-bias can be primarily attributed to missing or very low emission data from Jost (2021) due to missing orbits or cloud cover hindering
reasonable $SO_2$ retrieval. On most days, a strong spatial correlation between the simulated and observed horizontal dispersion of the $SO_2$ plume is evident, ranging from 0.6 to 0.9. However, the pronounced exceptions on June 7 and June 10 (also depicted in Fig. 8) may be attributed to misrepresentations of the meteorological conditions or generally lower wind speeds, leading to more turbulent flow.

After this comparison, we optimized the emission rates to improve the agreement between observed and simulated $\overline{SO_{2(col,d)}}$.
In the first step, we investigated the relation between the implemented emission rates and resulting simulated $\overline{SO_{2(col,d)}}$. Therefore, we considered the emission rates from the three preceding days of each sampled day in June 2018, denoted as



$\text{emis}_{\text{SO}_2;d-i}$ $(i = 0, 1, 2)$, and constructed a linear predictor of $\overline{\text{SO}_{2(\text{col},d)}}$:

$$\overline{\text{SO}_{2(\text{col},d)}} = \text{SO}_{2(\text{col},\text{BG})} + \sum_{i=0}^{3} a_{d-i} * \text{emis}_{\text{SO}_2;d-i} \tag{3}$$

The coefficients $a_{d-i}$ and the background $\text{SO}_2$ column amount, $\text{SO}_{2(\text{col},\text{BG})}$, represent the free parameters in the linear
predictor and were determined through a least squares fit. Subsequently, the linear predictor with these coefficients was utilized
to compute optimized emission rates for each day using a stochastic gradient descent (SGD) algorithm (in pseudocode):

---

**Algorithm 1** SGD optimizer for daily $\text{SO}_2$ emission rates

---

**Require:** `niter,lrate`

1:  **while** `j<niter` **do**

2:    **for** `d=0,d<30,k++` **do**

3:      **for** `k=0,k<3,k++` **do**

4:        `gradient=2*a`$_{\text{d-k}}$
          `*(SO`$_{2(\text{col,d})}$ $- \sum_{\text{i=0}}^{2}$ `a`$_{\text{d-i}}$`* emis`$_{\text{SO}_2;\text{d-i}}$`−SO`$_{2(\text{col,BG})}$`)`

5:        `emis`$_{\text{SO}_2;\text{d-k}}$`+=lrate*gradient`

6:      **end for**

7:    **end for**

8:    `j++`

9:  **end while**

---

The results of the simulation using these optimized emission rates are presented in the bottom row of Fig. 8 and are further
illustrated in Fig. 9. The most notable difference compared to the previous simulation is observed in $\overline{\text{SO}_{2(\text{col},d)}}$ (Fig. 9 bottom),
where the simulated column amount closely follows the observed pattern as expected, reflecting the optimization of emissions
based on observed $\overline{\text{SO}_{2(\text{col},d)}}$. However, barely any improvement is observed when examining the horizontal dispersion and
the corresponding spatial correlation. This lack of difference is primarily due to the intra-day fluctuations of $\text{SO}_2$ emissions,
which contribute to the observed variations and are also not accounted for in the optimized simulation.

Overall, this study demonstrates that degassing volcanic events can be effectively simulated using the new submodel EVER.
However, it is essential to apply high horizontal resolution (for a global model) to accurately capture the observed phenomena
and the small-scale wind fluctuations. Simulations performed at more standard horizontal resolutions, such as T63 and T106,
fail to reproduce the observations adequately (not shown), whereas these resolutions are mostly sufficient for stratospheric
simulations. Additionally, a thorough analysis of the emission rates is crucial for achieving reliable simulation results for
degassing volcanoes.



## 4 Model setup for historic volcanic eruptions

A primary aim of this study is the development of a methodology, that automatically integrates all volcanic eruptions significantly impacting the stratosphere in standard simulations using the MESSy framework, and thus reproduces stratospheric $SO_2$ and sulfate mixing ratios. Therefore, we established a default namelist configuration for the EVER submodel spanning the period from 1990 to 2023. This setup encompasses about 800 significant explosive volcanic eruptions, based on the $SO_2$ emission inventory developed by Schallock et al. (2023), which we extended up to the end of 2023, and refined with observations

from the IASI satellite. We evaluated this setup with $SO_2$ observations from the MIPAS satellite and observations of optical properties from OSIRIS.

### 4.1 Namelist setup

The emission inventory from Schallock et al. (2023) provides information regarding the mass of emitted $SO_2$ reaching the stratosphere, the maximum altitude of the plume as observed from satellites, the initial satellite observation date, and the

geographic coordinates of the volcano. This inventory was translated into an EVER namelist, based on the findings from Sect. 3.3. The development of the emission inventory involved satellite observations and the Global Volcanism Program, Smithsonian Institute (https://volcano.si.edu/, last access: 25 March 2024). It is important to note that this inventory may not include all relevant volcanic events, and we encourage the community to contribute additional significant volcanic events to the extendable namelist.

Following the approach detailed in Sect. 3.3, we refine the emission inventory by incorporating information from the IASI satellite if available. For each volcano, we conduct a scan of both temporal and spatial parameters, extending $\pm 10$ days from the emission inventory date and $\pm 10°$ latitude and $\pm 15°$ longitude from the volcano's geographical coordinates. From this analysis, we extract the space-time point exhibiting the maximum stratospheric $SO_2$ mixing ratios as the optimal estimate for both timing and geographical location for injecting the plume into the stratosphere.

The $SO_2$ mass is then distributed vertically in a Gaussian profile centered 1 km below the maximum altitude recorded in the emission inventory, with a total vertical width of 2 km over 6 hours around the identified date and time of peak mixing ratio as default. In reality, eruption duration and plume vertical width may vary, and can be adjusted for the study of specific eruptions. It is important to note that IASI became operational only in 2007 and primarily observes larger volcanoes. For eruptions occurring before 2007 or those not observed, we utilize the geographical location of the volcano and release the $SO_2$

mass from 9:00 to 15:00 UTC on the date provided by the emission inventory from Schallock et al. (2023). Consequently, these emissions are subject to uncertainties as discussed earlier.

We provide the namelist setup as a supplement for direct application in numerical simulations with the EVER submodel. All injections optimized using the IASI observations are marked accordingly.



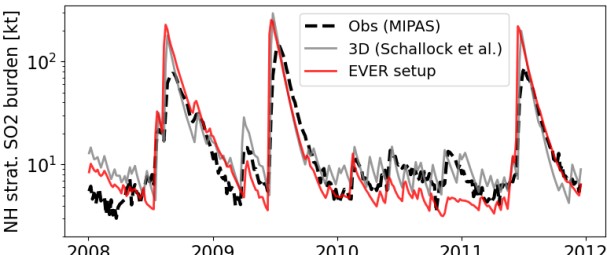

**Figure 10.** Timeline of total stratospheric $SO_2$ mass as observed from the MIPAS instrument (black dashed line) compared with simulations using the EMAC model with the new historic volcanic setup (red) and with 3D emission fields (Schallock et al., 2023). We use the same stratospheric cutoff as in Fig. 7

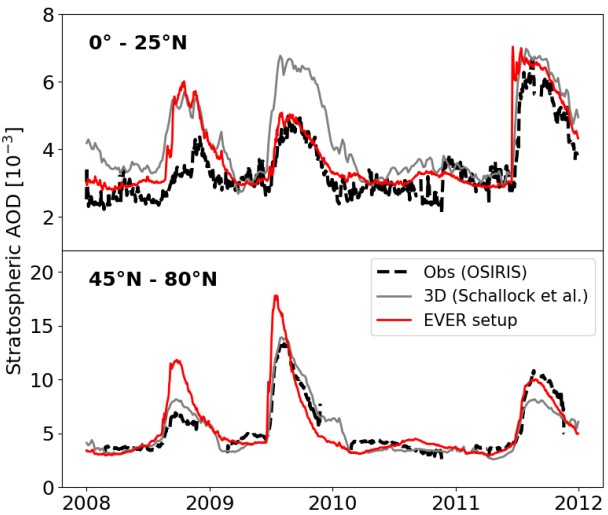

**Figure 11.** Timeline of stratospheric AOD (sAOD) as observed from the OSIRIS instrument (black dashed line) compared with simulations using the EMAC model with the new historic volcanic setup (red) and with 3D emission fields (Schallock et al., 2023). The sAOD is evaluated in the Northern hemispheric tropical latitudinal band from $0°$ to $25°$N (top) with a stratospheric cutoff altitude of 16 km and at higher northern latitudes from $45°$ to $80°$N at a stratospheric cutoff altitude of 12 km (bottom).

## 4.2 Results

We performed a simulation spanning from January 2008 to December 2011 to evaluate the newly developed historic volcanic setup for the EVER submodel, using the model configuration detailed in Sect. 3.1. This timeframe is characterized by high volcanic activity, encompassing three strong eruptions — Kasatochi (August 2008), Sarychev (June 2009) and Nabro (June 2011) — alongside several smaller eruptions. Over the simulated four years, a total of 107 stratospheric injections from volcanic events are documented by Schallock et al. (2023) and considered in the simulation.





Figure 10 illustrates total northern hemispheric stratospheric $SO_2$ burdens, analogously to the upper panel of Fig. 7 for the complete simulated timeframe. We compare MIPAS observations to our simulation using the new EVER setup and the simulation from Schallock et al. (2023) with 3D emissions at a single point in time.

The three primary peaks are largely reproduced similarly in both simulations, as anticipated, given that the injection mass from the emission inventory is derived from the 3D emission fields in the work of Schallock et al. (2023). However, in general, the observed peaks are lower than the simulated ones. This discrepancy can be attributed to the delayed response of satellite observations to stronger eruptions, as discussed previously (see Section 3.3.2).

The EVER simulation exhibits an underestimation of background $SO_2$ mixing ratios, particularly evident during the relatively quiet period from the end of 2010 until the eruption of Nabro in June 2011, whereas the 3D simulation mostly reproduces the background mixing ratios. The discrepancy in the EVER setup is likely attributable to limitations in the general model configuration or the overestimation of the vertical distribution width for smaller eruptions, only reaching the tropopause. Future efforts will concentrate on enhancing the representation of background aerosol concentrations in both, the free and upper troposphere, as well as in the stratosphere. However, it is worth noting that the simulation does reproduce smaller volcanic events, albeit at reduced magnitudes. As previously discussed, these smaller injections are not optimized in terms of horizontal position and timing, as they fall below the detection limit of IASI.

Figure 11 shows the stratospheric Aerosol Optical Depth (sAOD) for the four simulated years, categorized into two regions: the northern hemispheric tropical zone (0° to 25°N) and the mid to higher latitudes in the northern hemisphere (45°N to 80°N). In both regions, the three major peaks corresponding to the volcanic eruptions are evident. Discrepancies between the EVER simulation and observations are expected in regions where the stratospheric injection occurred, attributable to cloud overlap and saturation effects in the observations. This discrepancy is observed for Kasatochi (2008) and Sarychev (2009) at higher latitudes, and for Nabro at lower latitudes. Conversely, the 3D emissions, directly derived from satellite observations and applied with a delay, reproduce these measurement biases.

In the aftermath of the Nabro eruption in June 2011, the stratospheric Aerosol Optical Depth (sAOD) in the EVER simulation exhibits a sharp increase followed by notable fluctuations. The fluctuations can be attributed in part to the movement of the plume center across the 25° latitude band, resulting in variable alignment with the observed latitude window. Moreover, the presence of very high mixing ratios of $SO_2$ and subsequent formation of $H_2SO_4$ may introduce non-linearities in the model, particularly considering that emissions are confined to a single horizontal grid box. Consequently, very localized enhanced mixing ratios may emerge, potentially contributing to the observed sharp increase and the subsequent fluctuations. This phenomenon could potentially be addressed by distributing emissions across multiple horizontal grid boxes and releasing the $SO_2$ over an extended time period.

Another interesting feature is the pronounced overestimation of sAOD in the tropical latitudes following the Kasatochi eruption. This anomaly could be attributed to an overestimation of transport from higher latitudes to the tropical stratosphere, or a general overestimation of the emissions. Notably, this feature is not observed after the Sarychev eruption in the EVER simulation, although it is present in the 3D simulation.



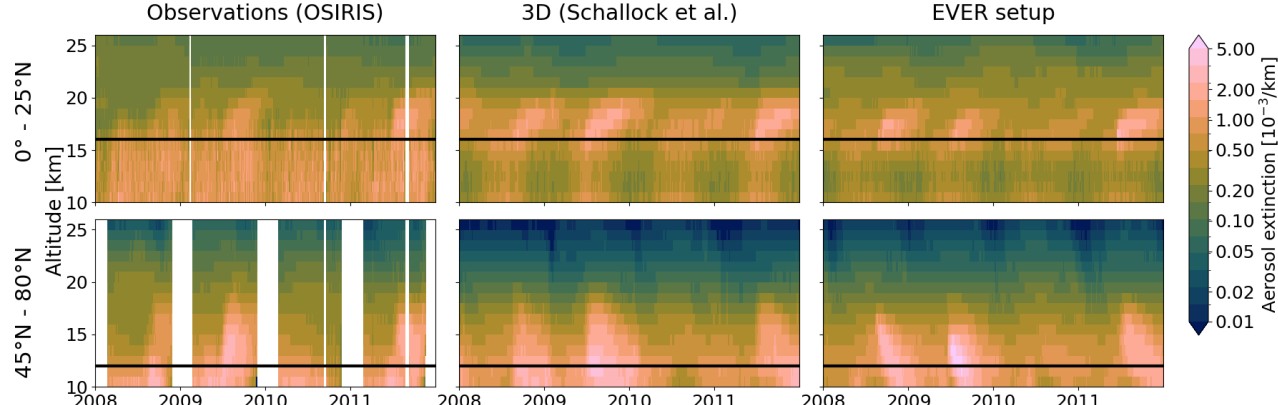

**Figure 12.** Timeline of aerosol extinction as observed from the OSIRIS instrument (left) and simulated by the EMAC model using 3D emissions (middle; Schallock et al., 2023) and using the new EVER setup (right), evaluated in the Northern hemispheric tropical latitudinal band from $0°$ to $25°$N (top), and at higher northern latitudes from $45°$ to $80°$N (bottom).

The differences in the long-term transport of the sAOD after the Sarychev (to the tropics) and Nabro (to high latitudes) eruptions can be attributed to the different timing of the emissions. While emission times in the EVER simulation were optimized to the first detection of stratospheric $SO_2$ from the IASI satellite, 3D emissions were applied on the dates specified in Schallock et al. (2023). The interaction with the South Asian monsoon anticyclone potentially causes differing transport to lower or higher latitudes, respectively.

It is important to note that total sAOD is highly sensitive to the altitude chosen as a lower cutoff. In this study, we adopted a cutoff of 16 km in the tropics and 12 km at higher latitudes, consistent with typical tropopause altitudes.

The aerosol extinction (Fig. 12) exhibits similar patterns in both, the observations and simulations. However, discrepancies are noticeable in the maximum altitude of the plume and below the tropopause. While the 3D simulation largely reproduces the observed maximum altitude, using satellite observations with, in case of MIPAS, the incorporated Aerosol Kernel Matrix (AKM) as input, the EVER simulation may present more realistic maximum altitudes. The differences between the observations and the simulation below the tropopause are strongly driven by the coincidence with clouds which hinder the retrieval of aerosol extinction. The slight differences between 10 and 15 km between the two simulations can most likely be attributed to the differences between the simulation setups. In addition, the 3D setup might better reproduce the altitude of maximum injection and maximum plume altitude due to direct derivation from satellite observations.

## 5 Discussion

We showed that $SO_2$ emissions from explosive volcanic eruptions and the subsequent plume evolution and aerosol formation can be reasonably reproduced in EMAC within the MESSy model system, using either 3D emissions (Schallock et al., 2023)



derived from satellite observations, column emissions with differing vertical profiles or point sources in one single grid box. The different approaches exhibit strengths and weaknesses, and reveal information on the general capabilities of the model.

The usage of 3D emissions, as investigated by Schallock et al. (2023), offers significant advantages for assessing the long-term impact of volcanic eruptions. By directly deriving emissions from 3D satellite observations several days after the initial eruption, this approach ensures an accurate representation of the plume's horizontal and vertical evolution, particularly during the crucial initial phase post-eruption, which is heavily influenced by local meteorological conditions. However, a notable drawback of this method is the inherent limitation in the vertical sensitivity of the satellite observations, leading to an over-estimation of the vertical width of the plume, and thus not reproducing the real distribution. Consequently, this discrepancy impacts the plume's subsequent evolution and results in differing stratospheric lifetimes for the aerosol burden. Additionally, due to the reliance on 3D satellite observations, short-term volcanic effects cannot be adequately examined as reliable data becomes available only days to weeks after the eruption. Another limitation of the approach outlined by Schallock et al. (2023) lies in its technical implementation, which necessitates either manual extraction, integration and correction of the 3D fields and subsequent model restarts after each volcanic event or the use of large import files, posing practical challenges for operational use.

Column or point source approaches rely on several assumptions regarding critical parameters such as plume height and location, emitted mass, and emission profile. With sufficient observational data, these parameters can be effectively constrained, enabling accurate predictions of $SO_2$ mixing ratios. However, this requires also manual work for creating the tables or to rely on existing inventories. As demonstrated with the Nabro volcano, we used data from the volcanic $SO_2$ emission inventory compiled by Schallock et al. (2023) in combination with observations from the IASI satellite to accurately retrieve these parameters. Consequently, our model simulations exhibited strong agreement with both, short- and long-term observations of $SO_2$ mixing ratios and aerosol optical properties obtained from IASI, MIPAS, and OSIRIS satellite instruments, specifically evaluated for strong eruptions.

The sensitivity studies revealed differences in the importance of the emission parameters for adequately simulating the strato-spheric aerosol burden. Primarily, emitting an appropriate quantity of $SO_2$ at the correct altitude appears to be the most critical factor. Variations in the $SO_2$ amount directly influence stratospheric $SO_2$ mixing ratios, with the mixing ratios approximately proportional to the mass emitted. Conversely, discrepancies in the injection altitude substantially impact the stratospheric life-time of the resulting $SO_2$ and sulfate, as well as the meteorological conditions encountered by the plume.

However, column or point source approaches come with inherent limitations as well. First, emissions are constrained to a single horizontal grid box in this study, potentially resulting in localized and exaggerated mixing ratios of $SO_2$ and $H_2SO_4$, leading to non-linearities in the model that diverge from reality, as volcanic plumes typically span several horizontal grid boxes. This effect is exacerbated even further when point sources instead of finite vertical distributions are employed, concentrating emissions within a smaller volume. This concentration can lead to lower $SO_2$ and aerosol mixing ratios in the mid- to long-term, as aerosols grow excessively large and subsequently sediment out of the stratosphere, as observed following the Nabro eruption. Therefore, we recommend using vertical distributions instead of point sources in all simulations of stratospheric volcanic eruptions. This recommendation is reinforced by observations indicating that volcanic plumes typically exhibit significant





vertical extension also in the stratosphere. While distributing emissions across multiple horizontal grid boxes may further mitigate this effect (by defining multiple emission points per eruption), we did not explore this aspect in our analysis. However, in detailed studies of strong eruptions, we additionally recommend exploring the effects of emissions over multiple horizontal grid boxes and an extended time period to avoid non-linearities due to very high concentrations.

Second, volcanic activity typically extends beyond a single day, with $SO_2$ emissions occurring over prolonged periods, occasionally reaching the stratosphere. The identification and appropriate mass allocation across multiple entry points pose challenges for automation. In the case of the Nabro eruption, our observations from IASI revealed two distinct stratospheric entry points. We observed discrepancies in the short-term evolution of $SO_2$ between the reference simulation, which considered only one stratospheric entry point, and the optimized simulation, which distributed emissions across two entry points. While these differences dissipated over the long term, it is important to note that this outcome may not be necessarily applicable to all volcanic eruptions.

Third, it is important to note that detailed data regarding the precise timing and geographical coordinates of the stratospheric entry, as derived from IASI, is not accessible for all volcanic eruptions. Smaller eruptions, in particular, often fall below the detection threshold of IASI, and IASI only became operational in 2007. Our sensitivity analyses, where we varied the timing and geographical positioning of plume entry into the stratosphere, uncovered short- and long-term disparities when such information is lacking.

Based on the insights from the sensitivity studies, we developed a historical namelist configuration for the new EVER submodel spanning the past three decades, employing vertical distributions. Despite the aforementioned limitations, our evaluation simulation demonstrates a satisfactory alignment with observations of both, $SO_2$ mixing ratios and aerosol optical properties. Notably, the historic namelist setup accurately reproduces significant eruptions, thereby representing the primary contribution of volcanic events to the stratospheric $SO_2$ and aerosol load. This aligns with our conclusion that altitude and mass are the most crucial emission parameters, which were directly determined from the emission inventory and do not depend on the availability of IASI observations.

Consequently, we recommend the adoption of the submodel EVER with the proposed historical namelist setup (see Sect. 4 and supplement) in all numerical simulations using the MESSy framework at global or regional scale, particularly those encompassing the stratospheric and upper tropospheric domains. This is the first study to systematically incorporate volcanic eruptions into atmospheric simulations within MESSy using the EMAC model, presenting a more flexible and easy-to-implement alternative to the 3D emission approach. Furthermore, in cases where disparities with observations arise, owing to the aforementioned uncertainties, or when focusing on specific volcanic events, the namelist setup can be adjusted accordingly. Moreover, comparisons with simulations using 3D emission fields may offer additional insights into the evolution of individual volcanic plumes.

In section 3.4, we demonstrated the additional capability of EVER to simulate $SO_2$ from degassing volcanoes. Furthermore, we applied a model-driven optimization approach for the corresponding emission emission rates, reproducing observed $SO_2$ column amounts.



This optimization approach has the potential to be extended to historic degassing volcanoes, facilitating the development of a default setup akin to the stratospheric default setup for explosive volcanoes. This process would include integrating TROPOMI observations with an initial simulation using rough estimates of degassing emissions, which could subsequently be refined through stochastic gradient descent (SGD) optimization. This capability could potentially replace the outdated climatology by

640 Diehl et al. (2012) used in many EMAC simulations.

However, implementing this approach may encounter challenges related to the required high horizontal resolution, the estimation of the injection altitude, and the identification and initial approximate estimation of all significant degassing volcano emissions.

Up to this point, our focus has been primarily on volcanic $SO_2$. However, it is worth mentioning the versatility of the sub-

645 model for a wide range of use cases where gaseous or aerosol tracers are injected into the atmosphere in vertical distributions, with limited horizontal extent. This includes the following use cases:

- **Volcanic ash:** Apart from emitting trace gases, volcanic eruptions also release primary aerosols, such as volcanic ash. The EVER submodel is explicitly designed to simulate the evolution of aerosol species, including volcanic ash, after volcanic eruptions.

650 - **Water vapor:** Eruptions of submarine volcanoes, such as the notable event at Hunga Tonga-Hunga Ha'apai in January 2022 (e.g., Vömel et al., 2022; Sellitto et al., 2022; Schoeberl et al., 2022; Xu et al., 2022), release substantial quantities of water vapor into the atmosphere. The EVER module can be used to investigate the effects of enhanced water vapor concentrations in the stratosphere due to volcanic activity. For the Hunga Tonga eruption we recommend the simultaneous injection of water vapor in multiple horizontal grid boxes to avoid quick removal by ice formation and to be

655 consistent with observations.

- **Wildfires:** Strong wildfires can inject significant amounts of carbonaceous aerosols and various trace gases directly into the stratosphere via pyro-cumulonimbi. EVER can be used to model these emissions from wildfires.

- **Solar geoengineering:** Studies on solar geoengineering, particularly artificial injections of $SO_2$ or other trace gases into the stratosphere to form aerosols that reflect sunlight back into space, can benefit from the capabilities of EVER. These

660 scenarios involve large uncertainties, which can be addressed with studies using EVER.

- **Transport processes:** Transport processes play a crucial role throughout the atmosphere. EVER allows for the emission of active and passive aerosols and trace gases throughout the atmosphere, enabling the study of processes such as the exchange between the troposphere and stratosphere.

- **Sensitivity studies:** Atmospheric properties can be highly sensitive to perturbations in trace gas or aerosol mixing ratios.

665 By injecting the respective atmospheric constituents with EVER, it is possible to estimate the sensitivity of climate, atmospheric dynamics, and the ozone column to these perturbations.



## 6 Conclusions

We presented the new submodel for tracer emissions from Explosive Volcanic ERuptions (EVER v1.1), developed within the Modular Earth Submodel System (MESSy, version 2.55.1).

First, we described the new submodel, designed for the addition of gaseous and aerosol tracer tendencies following volcanic eruptions in columns with user-specified vertical profiles at point or area sources. Size-resolved volcanic particles, such as ash, can be mapped to number and mass of model aerosol size modes. We evaluated the EVER submodel with the simulation of volcanic $SO_2$ emissions in the ECHAM5/MESSy Atmospheric Model (EMAC) for the explosive eruption of Nabro in June 2011 and a degassing event from Kilauea in July 2018, employing satellite observations from IASI, MIPAS, OMI, TROPOMI, and OSIRIS. The EVER submodel is available from MESSy version 2.55.1 and will be continuously developed further.

Second, we performed sensitivity simulations of $SO_2$ emissions from the Nabro eruption with the EMAC model. They revealed the importance of the emission of a reasonable amount of $SO_2$ above the tropopause with an appropriate altitude distribution. Horizontal position and emission timing were found to have a minor impact on the long-term $SO_2$ burden in the stratosphere. Nevertheless, these parameters play a crucial role in detailed process studies during the initial weeks after an eruption. Overall, we conclude that simulations of volcanic eruptions can be effectively performed with the help of 3D-, column, and point emissions. The optimal approach depends on the specific use case, with column emissions excelling in the short-term, and similar performance in the long-term.

Third, we developed a historic submodel setup for EVER, incorporating stratospheric significant volcanic eruptions spanning from 1990 to 2023. It is based on the volcanic $SO_2$ emission inventory by Schallock et al. (2023). We additionally optimized the timing and geographical location of the volcanic plume entering the stratosphere, using $SO_2$ observations from the IASI satellite. However, this information was only available from 2007 on, and for strong volcanic eruptions only. The historic namelist setup is provided as a supplement, and we advocate its inclusion in simulations using the MESSy framework focusing on the upper troposphere and stratosphere. For very strong eruptions, it may be beneficial to distribute the emissions over multiple horizontal grid boxes and an extended time period, or adjust the vertical plume width, if discrepancies with observations occur.

In addition to the extensively discussed application to explosive volcanic eruptions, the versatility of the EVER submodel gives rise to various other research areas. These include investigations into the interplay between $SO_2$ and volcanic ash post-eruption, exploration of solar geoengineering scenarios, modeling of wildfires, and analyses of atmospheric transport processes. Future work could involve the development of a climatology of $SO_2$ emissions from degassing volcanoes employing the new submodel.

*Code availability.* The Modular Earth Submodel System (MESSy, https://zenodo.org/doi/10.5281/zenodo.8360186) is continuously further developed and applied by a consortium of institutions. The usage of MESSy and access to the source code is licenced to all affiliates of institutions which are members of the MESSy Consortium. Institutions can become a member of the MESSy Consortium by signing the MESSy Memorandum of Understanding. More information can be found on the MESSy Consortium Website (http://www.messy-interface.



700 org). The code presented here is available in MESSy version 2.55.1 (https://zenodo.org/records/8367075). The respective namelists, chemical mechanisms and run scripts used are made available via supplement (see Sect. 3.1).

Scientific colour maps (https://doi.org/10.5281/zenodo.5501399, Crameri, 2021) are used in this study to prevent visual distortion of the data and exclusion of readers with colour vision deficiencies (Crameri et al., 2020).

The historic default namelist setup for the new submodel EVER is available as supplement (*ever_historic_stratVolcanoes.nml*).

705 *Data availability.* SO$_2$ data from MIPAS observations is available after registration at http://www.imk-asf.kit.edu/english/308.php (Höpfner et al., 2013, 2015). OMI observations are taken from https://disc.gsfc.nasa.gov/datasets/OMSO2_003/summary (Li et al., 2020). IASI SO$_2$ products are available at https://doi.org/10.25326/41 (Clarisse, 2023). We obtained OSIRIS aerosol products from https://research-groupstest.usask.ca/osiris/data-products.php (Rieger et al., 2019). TROPOMI observations are publicly available at https://doi.org/10.5270/S5P-yr8kdpp (ESA, 2018). Model output and setups are archived at the DKRZ in Hamburg, and are available on request. The historic SO$_2$ emission in-
710 ventory of explosive volcanic eruptions is available at https://doi.org/10.26050/WDCC/SSIRC_3 (Brühl et al., 2021; Schallock et al., 2023).

*Author contributions.* MK and AP planned the research. MK developed the EVER submodel with the help of AP and PJ. JS and CB provided the emission inventory, performed the simulation with 3D emissions, and assisted in the provision of satellite data and information on volcanoes. AJ and SB provided the daily SO$_2$ emission rates for the Kilauea volcano. MH provided the MIPAS observations with the respective averaging kernel matrix (AKM). MK evaluated the simulations, analysed the model results, developed the historic namelist setup,
715 and wrote the manuscript. AP and HT supervised the project. All authors discussed the results and contributed to the review and editing of the manuscript.

*Competing interests.* Two co-authors are member of the editorial board of Geoscientific Model Developments.

*Acknowledgements.* MK acknowledges the financial support of the Max Planck Graduate Center with the Johannes Gutenberg University (Mainz). The model simulations have been performed at the German Climate Computing Centre (DKRZ) through support from the Max
720 Planck Society. We acknowledge the effort of Michael Kiefer to supply the Averaging Kernel Matrices for the SO$_2$ product from MIPAS. HT acknowledges funding support from the Deutsche Forschungsgemeinschaft (DFG, German Research Foundation) – TRR 301 – Project-ID 428312742.



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
