# Peer review of "New submodel for emissions from Explosive Volcanic ERuptions (EVER v1.1) within the Modular Earth Submodel System (MESSy, version 2.55.1)"

_EGUsphere, 2024_

## Referee Comment (RC1)

**Review for «New submodel for emissions from Explosive Volcanic ERuptions (EVER v1.1) within the Modular Earth Submodel System (MESSy, version 2.55.1)» by Kohl et al.**

The submitted manuscript contains the description of a new submodule within MESSy. The submodule is described in detail in section 2. Section 3 provides a description of the model setup for the validation of the submodel. For model validation the 2011 Nabro and 2018 Kilauea $SO_2$ emission plumes were evaluated with different sensitivity simulations and with a fine-resolution simulation, respectively. In chapter 4 the explosive volcanic $SO_2$ injections of the time period 2007 to 2011 was simulated based on a 3-D observational emission dataset which was improved based on the findings of chapter 3.

The submitted manuscript describes a new submodule for emission of volcanic substances and provides some validation for the evolution of $SO_2$ plumes after volcanic eruptions. However, in my point of view, the manuscript has some substantial flaws, which I explain below and which need to be addressed. Even though topic wise the manuscript would be a good fit for publication in GMD, I do not think that quality wise this manuscript deserves publication in this journal.

The main environmental impact of large explosive volcanic eruption on a global scale (heterogeneous chemistry, cooling of the surface, local warming of the stratosphere) are effects from sulfuric acid aerosols and not from $SO_2$. The manuscript almost entirely focuses on the simulation of $SO_2$ plumes and thus, a crucial aspect of volcanic emission plumes is missing. I know that for the accurate simulation of aerosol burden and their effects it is important to accurately simulate the $SO_2$ plume in the first place, however, if this was the goal of this study this needs to be motivated and explained in the introduction.

To me many important aspects of a paper are not sufficiently explained: What is the novelty of this study? What can the model do what other models can't do? Why is it important to simulate the emissions of $SO_2$ accurately? Why do we need yet another model which is capable of simulating volcanic injections? What have other models shown and how does your model compare to other models?

I think the manuscript does not provide/discuss a substantial part of literature relevant for the simulation of volcanic emission plumes. There have been many studies, which highlight important aspects discussed in this manuscript, which are not even cited (e.g. Brodovsky et al. 2021 or Quaglia et al. 2023). Usually, the 1991 Mt. Pinatubo eruption is used as a basis for model validation (e.g., Quaglia et al. 2023, which even includes EMAC simulations). Why did you not perform a simulation of Mt. Pinatubo to validate your model? Probably because this has been done extensively already, but it would allow comparing also with other models.

The introduction could focus more on topics relevant to the rest of the manuscript. E.g. why do you explain ozone chemistry and ash in such detail if it is not relevant to the

manuscript? The manuscript provides a very broad discussion about the general impact of volcanic eruptions on climate and atmospheric chemistry, but it does not discuss the problems and current limitations when modeling volcanic eruptions (e.g., spatial & temporal resolution, model agreement/disagreement of past model studies among each other and with observations, sectional versus modal microphysics modules, previous studies which addressed the vertical/horizontal distribution of volcanic emission plumes). Since you submit to "Geoscientific model development" I think a more technical motivation/introduction would be appropriate. I think the introduction should clearly motivate the research so that it is clear to a reader why the submodule was developed and why there is need for the research provided. In my point of view this is not the case.

Your conclusions just summarize what you have done in your work, including a brief recapitulation of some qualitative results with a brief outlook for further potential applications of the sub-model in the end. However, I think the conclusion section should summarize the key findings of the paper (also some quantitative statements, not only qualitative) and put them into a broader context. The conclusions should demonstrate the importance of the paper and convey the larger implications of the paper to the research field. Additionally, I also find it important to address the limitations of the research/model provided (or more specifically: of the presented sub-module) and highlight aspects identified in your work which require further research and development. I don't think these points are provided in the current version of the conclusion section of the manuscript.

Similarly, the abstract just summarizes what you did in this study, but you do not provide any results or conclusions or broader implications to the research fields in the abstract, which I think is an essential part of the abstract.

An important aspect which is missing in the analysis and discussion of your results is the role of chemical loss of $SO_2$. In addition to dilution and transport, a (maybe even more) important aspect which determines the $SO_2$ lifetime is chemical $SO_2$ loss (i.e., mainly oxidation with OH and O3). How is this represented in MECCA, and how does this influence your modelled results. You also write that for the Kilauea study, you applied "simplified chemistry". What does this mean? Since chemical loss is very important for the $SO_2$ lifetime you need to discuss how this affects the results in your study. How is the reaction with O3 and OH represented in MECCA? Were the O3 and OH fields also nudged to observations like written in the model description (probably not)? How does this influence the modelled results?

Another aspect which is not discussed is the importance of aerosol microphysics. You show an in-depth analysis of the $SO_2$ plume evolution and then in chapter 4 you suddenly come up with AOD analysis. However, there is an important step missing in between: Aerosol Microphysics. You write that you are using GMXE as an aerosol microphysics module and represent the aerosols using a modal approach. However, you

do not provide any information about resulting aerosol burden, aerosol size distribution ect. You only show AOD and extinction (without indicating the wavelengths under consideration). Thus, the whole aerosol microphysics (which is a very important aspects when simulating volcanic emission plumes) is treated as a black box in this manuscript. In my point of view, it is crucial to also show resulting aerosol burden and compare them with observations as well as which other models (e.g. see Brodowsky et al. 2021).

Another aspect which comes too short in the discussion section is the influence of the spatial and temporal resolution of different processes. It is well known that these aspects are very important for realistic representation of volcanic plumes. While for the troposphere the horizontal resolution is more important in the stratosphere the vertical resolution is more important. For aerosol microphysics the microphysical timestep should be set small enough to realistically simulate nucleation and condensation. You could mention these aspects in the discussion of your results.

I also find the manuscript too long. Many aspects which are discussed in the introduction and submodel description are not relevant to the storyline or are not picked up again in the discussion. I suggest shortening substantially and putting part of the text (e.g. description of code and namelists) into the supplement. Also, the structure could be improved. For example, the model description and setup and description of the observations (sect. 3.1 and sect. 3.2) could be a chapter for its own or part of chapter 2. Because the setup described there is also used in chapter 4. The different events simulated here (i.e., Nabro, Kilauea and the 2007-2011 period) seem a little disconnected to each other. Why don't you show a full analysis of only one event (e.g. Nabro), but in more detail including sulfuric acid aerosol burden ect.

More detailed comments can be found below.

Brodowsky, C., Sukhodolov, T., Feinberg, A., Höpfner, M., Peter, T., Stenke, A., & Rozanov, E. (2021). Modeling the sulfate aerosol evolution after recent moderate volcanic activity, 2008–2012. *Journal of Geophysical Research: Atmospheres*, 126, e2021JD035472. https://doi.org/10.1029/2021JD035472

Quaglia, I., Timmreck, C., Niemeier, U., Visioni, D., Pitari, G., Brodowsky, C., Brühl, C., Dhomse, S. S., Franke, H., Laakso, A., Mann, G. W., Rozanov, E., and Sukhodolov, T.: Interactive stratospheric aerosol models' response to different amounts and altitudes of $SO_2$ injection during the 1991 Pinatubo eruption, Atmos. Chem. Phys., 23, 921–948, https://doi.org/10.5194/acp-23-921-2023, 2023.

Timmreck, C., Mann, G. W., Aquila, V., Hommel, R., Lee, L. A., Schmidt, A., Brühl, C., Carn, S., Chin, M., Dhomse, S. S., Diehl, T., English, J. M., Mills, M. J., Neely, R., Sheng, J., Toohey, M., and Weisenstein, D.: The Interactive Stratospheric Aerosol Model Intercomparison Project (ISA-MIP): motivation and experimental design, Geosci. Model Dev., 11, 2581–2608, https://doi.org/10.5194/gmd-11-2581-2018, 2018.  a, b, c, d

Mills, M. J., Schmidt, A., Easter, R., Solomon, S., Kinnison, D. E., Ghan, S. J., Neely, R. R., Marsh, D. R., Conley, A., Bardeen, C. G., and Gettelman, A.: Global volcanic aerosol properties derived from emissions, 1990–2014, using CESM1 (WACCM), J. Geophys. Res.-Atmos., 121, 2332–2348, https://doi.org/10.1002/2015JD024290, 2016.

**Detailed Comments:**

**Line 1:** What is a methodological study? Isn't every scientific study methodological?

**Line 14/15:** Suggesting to change "solar geoengineering" to "solar radiation modification", a more appropriate term.

**General Comment on the abstract:** The abstract mostly reflects what was done in this study, but there is no mentioning on results and conclusions, which I think is a key component of an abstract. Thus, I suggest adding some quantitative results and conclusions/broader impacts.

**Line 19/20:** "On the on hand… on the other hand" is normally used for opposing arguments. The ones mentioned here are more additive. I suggest reformulating. Whether it is "substantial" or not, depends on the magnitude.

**Line 25 and lines 32-34:** Do you have references for that?

**Line 25/26:** Do you have a reference for this?: "The composition of volcanic plumes exhibits considerable variability and depends on the intricate mixture of chemical species in the magma"

**Line 27-31:** I suggest combining the two sentences (i.e., list the example of Hunga Tonga in the first sentence).

**Line 32-34:** a definition of "long-term" would be good. sulfuric acid aerosols and their precursors are usually removed from the stratosphere within 2 years. I don't think this is long-term in terms of climate. Maybe chlorine species would have a long term effect.

**Line 34-36:** This is somehow confusing. In the first paragraph you speak of "the most explosive volcanic eruptions" and now you write of emissions per year. Do you still speak of large explosive volcanic eruptions or do these numbers also account for degassing non-volcanic eruptions? I suggest being more precise here to what exactly these numbers refer to.

**Line 37/38:** I suggest changing the term "sulphate" with "sulfuric acid", since technically speaking, a sulphate is a solid (e.g. CaSO4). Also add references to this statement.

**Line 36:** Maybe add "under volcanic conditions", otherwise other sulfuric acid precursor gases are also important. I know $SO_2$ is poisonous, but isn't the effect on acid rain mainly a result of uptake of sulfuric acid aerosols (and not primarily $SO_2$)?

**Line 46:** "up to" or "over"? But not both.

**Line 81:** What is a "horizontal gid box"? What is the difference between a horizontal or vertical grid box? Do you mean "vertical column of gird boxes"?

**Line 85:** Are you only aiming at providing or are you providing? I suggest reformulating to "We provide …", or reformulate in another way if you don't. Same for the last sentence of the paragraph: "This was achieved through the following three steps of work: ". Don't undersell your research.

**Line 90:** "… of vertical emission distributions … "

**Line 116:** You only use GMXe aerosol microphysics module in this study. Why do you introduce the other two submodules too? This only causes confusion.

**Line 127:** What do you mean by "linear columns"?

**Section 2.1:** I think the description of the new submodel is too technical. It is probably not useful for most readers of this study. I suggest making the description of the new sub model more general and provide a more technical description (e.g. how the name list works, what the different name list parameters are) in the supplement.

**Section 2.2.1:** The title of this section is "Primary emissions", but the subchapter is specifically on direct aerosol emissions. Maybe specify this in the title.
However, why is this subchapter important? In this manuscript, only $SO_2$ injections are evaluated. Maybe you can skip this subchapter or put it into the supplement.

**Section 2.2.2:** This could be picked up later in the paper.

**Section 3.1** There is no mentioning of the microphysical, chemical and dynamical time steps applied in the models used in this work. A recent study has highlighted the need for appropriately setting the microphysical time step when simulating volcanic eruptions.

**Section 3.2:** Maybe this section can be shortened. Are such detailed descriptions of all the different satellite and technical details such as their resolutions required?

**Section 3/4:** I assume the model description provided in section 3.1 also applies to section 4, right? And some of the described observations (satellites) in 3.2 are only used in section 4, right? To avoid confusion, I suggest separating the model description and observations from chapter 3 and create an own chapter for this. Or maybe something similar, just improve the structure of the paper, it is confusing sometimes.

**Line 203-207:** Why is this information important at all? You are not looking at aerosols in this chapter, but only at $SO_2$ plumes. Is $SO_2$ also treated within GMXe?

**Line 200:** Nudged to which variables? Wind and temperature?

**Line 222:** Maybe just write: "Namelist setup, chemical mechanism and runscripts can be found in the supplement." However, I think this belongs in the data availability statement not in the main text.

**Line 216-219:** Why is this information important? In this chapter (chapter 3) you are only focusing on $SO_2$ plumes, but no aerosol optical effects. I suggest skipping.

**Line 284/285:** "Especially the second stratospheric plume on June 16 could comprise remnants of the tropospheric plume, that are uplifted"
This sentence is confusing to me: What do you mean with tropospheric plume? The volcanic plume or the monsoon?

**Line 323:** There is no specification of the emission in Mills et al. 2016 so far. This pups up here a little abruptly, since this was not discussed in the introduction or anywhere prior to here.

**Line 334:** "The column amount estimation assumes that all $SO_2$ of the plume is centered at the respective altitude depicted in Fig. 4 "
To me it is not clear how this explanation should explain the discrepancies between observed and modelled column amounts. Can you explain further?

**Line 338:** "From Fig. 4, it seems that the simulated columns slightly broaden over time compared to the observations, with the plume appearing to sink." I am confused here. Figure 4 shows altitudes not column $SO_2$. It also seems like the simulated plume (reference) is higher up compared to the observed plume. Do you mean the observed plume appears to sink? It is hard to see any broadening of the plume in Figure 4.

**Line 339/340:** The simulated (reference) $SO_2$ column distribution in Figure 5 is broader compared to the observation... not narrower. This is confusion.

**Line 361/362:** "However, IASI faces limitations in capturing the long-term evolution of volcanic plumes due to the dilution of the emitted $SO_2$, leading to column amounts that fall below the instrument's detection limit»
Do you really know dilution is the main process that $SO_2$ concentrations fall below the detection limit? If yes, do you have references for this? Isn't chemical loss ($SO_2$ oxidation via OH and O3) equally or even more important on longer time scales? How is this represented in the model and how does this affect the long-term evolution of the $SO_2$ plume?

**Lines 416-422:** "The overall slightly faster decline observed in the simulation compared to the observations may be a consequence of the absence of primary particles, such as volcanic ash, in the simulations, resulting in a discrepancy between simulated and observed particle size distributions."
What processes should be the reasons for that? I guess you mean that ash could result in self-lofting of airmasses due to absorption of radiation and thus local heating? Or what other processes do you have in mind? It is important to name them since this is not clear from how it is written now.

"Alternatively, the simulated particle sizes may grow excessively large too quickly, leading to an overestimation of sedimentation efficiency»

You do not show any simulated particle sizes. You show and write about $SO_2$ plumes. $SO_2$ is a gas. Gases are mainly subject to diffusion & transport and in the case of $SO_2$ more importantly: chemical loss, ... but definitely not sedimentation. What about chemical loss? How does this affect the dissipation of the plume compared with observations?

"Whether this discrepancy arises from nucleation rates versus condensation efficiency, the overall representation of the size distribution with only four modes, or the limitation to one horizontal grid box will be the topic of upcoming studies."
$SO_2$ concentrations are definitely not affected by nucleation and condensation rates. Chemical loss of $SO_2$ is dominated by reaction with OH and O3. These reaction result in formation of $SO_3$, which then together with H2O forms H2OS4 gas. H2SO4 gas has a very low vapor pressure and immediately forms sulfuric acid aerosols via condensation or nucleation. Have a look at Feinberg et al. 2019 and the stratospheric sulfur cycle presented in there. It should get obvious that nucleation and condensation rates as well as aerosol size distributions do not affect the chemical $SO_2$ lifetime/burden.

Feinberg, A., Sukhodolov, T., Luo, B.-P., Rozanov, E., Winkel, L. H. E., Peter, T., and Stenke, A.: Improved tropospheric and stratospheric sulfur cycle in the aerosol–chemistry–climate model SOCOL-AERv2, Geosci. Model Dev., 12, 3863–3887, https://doi.org/10.5194/gmd-12-3863-2019, 2019.

**Line 441:** What "data"? Simulated or observed?

**Line 341/342:** Ahaa... it only becomes clear that you were talking about the initial plume on June 14 until now. The statements you make in this paragraph are only true for the initial plume on June 14. You really need to be more precise here... The statements of this paragraph are not valid for June 17.

**Figures:** I suggest assigning letters a, b, c, d ... to the subpanels of figures to enable better referencing.

**Figure 444:** It is hard to see any difference in agreement/disagreement with observation in Figure 8 of the June 7 and 10 data compared to for example June 5 and June 15. I suggest plotting the differences compared to the observations in the middle and lower panel. This would highlight the differences.

**Line 448-451:** Most of this can go into the figure caption.

**Line 556:** I disagree with that. The observations and the model does not "exhibit similar patterns" between 0 and 25N. The QBO signal is much more pronounced in the model compared to the observations. And the observed extinction is very different compared to the modelled ones in absolute numbers.

**Line 459:** Why do you think the observations are wrong? It could well be your model which is wrong. Why don't you optimize your model to improve agreement with observations?

**Line 460:** With "implemented emission rates" you mean the observations, right?

**Line 461:** I guess you considered the "emission rates" from the observations, right?

**Line 465:** "The coefficients $a_{d-i}$ and the background $SO_2$ column amount, $SO_{2\,(col,BG)}$, represent the free parameters in the linear predictor and were determined through a least squares fit"
To what is "least square fit" referring to? What is it fitted to? "Least square fit" to the observed total column $SO_2$? If yes, then it should be obvious that the simulations in the end agree with the observed total column $SO_2$.

**Line 466:** What is a "stochastic gradient descent"? It would be helpful to describe this in one sentence, other wise it is just a black box to most readers. The code provided below does not help, since this is rather technical. This can go to a supplement.

**Line 470:** Why would you expect increases in spatial correlation if you only improve the emission rates?

**Line 473:** What do you mean with "effectively". I disagree with this statement. You only get good agreement in total column $SO_2$ when tuning the emissions in your model to fit the observational data. I think most models get better agreement with observations when tuning their emissions.

**Line 474-478:** You did not investigate any sensitivity to spatial resolution. Thus, you cannot make this conclusions. Delete this part, or show evidence for this conclusions.

**Line 477/478:** This is the most critical result which I think you must discuss more. You only get good agreement with total column $SO_2$ observations if you tune the emission rates according to your simulation results. I know that this is a common problem for models simulating volcanic eruptions (e.g. Mt. Pinatubo), but you should highlight this. critically discuss it and derive the right conclusions.
I also think "analysis" is not the right word here. More precise would be "tuning".

**Line 520/521:** The magnitude of the signal would not change if the satellite signals were only delayed compared to observations. Isn't it mainly the sensitivity of the measured satellite signal? Please be more precise here.

**Line 522-529:** Here again: What is the impact of chemical loss of $SO_2$? Also did you compare your model to background sulfur cycle (see Brodovsky et al. 2024)? It might make sense to fist perform same simulation as in this study to compare to other models and observations.
Brodowsky, C. V., Sukhodolov, T., Chiodo, G., Aquila, V., Bekki, S., Dhomse, S. S., Höpfner, M., Laakso, A., Mann, G. W., Niemeier, U., Pitari, G., Quaglia, I., Rozanov, E.,

Schmidt, A., Sekiya, T., Tilmes, S., Timmreck, C., Vattioni, S., Visioni, D., Yu, P., Zhu, Y., and Peter, T.: Analysis of the global atmospheric background sulfur budget in a multi-model framework, Atmos. Chem. Phys., 24, 5513–5548, https://doi.org/10.5194/acp-24-5513-2024, 2024.

**Line 530**: Here you suddenly start talking and comparing AOD resulting from these volcanic eruptions. So far you talked and compared $SO_2$ plumes. It would be great to first see some sulfuric acid aerosol size distribution or how the sulfuric acid aerosol plume/burden evolves in the aftermath of these volcanic eruptions (see Brodowsky 2021). This is what defines the AOD downstream not the $SO_2$ plume. There is an important part missing here when going from $SO_2$ plumes to AOD. Without this intermediate step it is hard to say where the discrepancies between model and observations are coming from. It is just guessing since aerosol formation and distribution in the model appear like a black box to the reader…

Also, crucial information is missing about the wavelengths to which the AOD and extinctions shown in Figure 11 and 12 are referring to.
Why did you only look at 0 ∘ to 25∘N and 45°-80°N? and not other regions?

**Line 537-544:** This paragraph needs references and is somehow handwaving.

**Line 539/541:** I think this reads a little hand wavy here. Please be more specific. A paper which addresses some potential effects is Vattioni et al. 2024.
Vattioni, S., Stenke, A., Luo, B., Chiodo, G., Sukhodolov, T., Wunderlin, E., and Peter, T.: Importance of microphysical settings for climate forcing by stratospheric $SO_2$ injections as modeled by SOCOL-AERv2, Geosci. Model Dev., 17, 4181–4197, https://doi.org/10.5194/gmd-17-4181-2024, 2024.

**Line 542-544:** "This phenomenon could potentially be addressed by distributing emissions across multiple horizontal grid boxes and releasing the $SO_2$ over an extended time period."
Why would you do this? In section 3 you showed good spatial agreement with observations, so why change the spatial distribution? What I think could help might be changing the horizontal resolution. It seems you again are looking for the error in the emission scheme/observations instead of in within the model. Your suggestion would reduce the $SO_2$ concentrations and thus the H2SO4 concentrations downstream. This reduces condensation and especially aerosol nucleation rates. But why should this be justified?

**Line 546/547:** "This anomaly could be attributed to an overestimation of transport from higher latitudes to the tropical stratosphere, or a general overestimation of the emissions."
Why should this be the case? Isn't the transport in this region going exactly into the other direction (from the tropics to higher latitudes)? And there is also a tropical "transport barrier".

**Line 549:** What differences are you talking about? Difference compared to what?

**Line 552/553:** "The interaction with the South Asian monsoon anticyclone potentially causes differing transport to lower or higher latitudes, respectively."
Weren't the simulations nudged towards observed wind fields?

**Line 554/555:** This sensitivity needs to be addressed by showing some plots in the supplement with different cutoff altitudes, since this defines whether the model agrees with observations or not...

**Line 556-561:** I would make it clear in this paragraph (and for the whole discussion of AOD from line 530 onward) that here you are talking about the sulfuric acid aerosol plume and the AOD resulting from these aerosols, whereas so far in the paper you talked about the $SO_2$ plume. The two $SO_2$ and sulfuric acid aerosol plumes likely look different.

**Line 565/566:** The first sentence of the discussion is not true. You do not show anything related to "aerosol formation". You only show comparison with AOD observations, but this does not tell you anything about aerosol formation processes.

**Line 575:** You do not show "aerosol burden" here. Thus, you can not make any conclusions about this.
Do you mean "forecasted" instead of "examined"? They can be examined, but just not immediately.

**Line 589-593:** You did not analyze how to "adequately simulate stratospheric aerosol burden". You cannot make any conclusions about stratospheric aerosol burden, if you do not show aerosol burden in the manuscript. The first sentence is confusing. What do you mean with "differences" in the first sentence of this paragraph? A difference compared to what? Again, what is the importance of chemical loss of $SO_2$ in the whole analysis? This could also be discussed here. You cannot make conclusions about the "sulfate" lifetime with the analysis shown in your manuscript.

**Line 594-605:** I agree that the horizontal extent of the emissions can influence the simulations.
"...emissions are constrained to a single horizontal grid box in this study..." I know what you mean, but this reads wrong. You also applied column emissions and vertically gaussian distributed emissions, which do not inject into "one single grid box". I would change this to "...emissions are constrained to a single grid box or columns of gid boxes in this study..." or make this clearer in a different way (e.g. what is the difference between a horizontal and a vertical gid box? ) To me a grid box is a grid box... whether it is vertical or horizontal.

"...leading to non-linearities in the model that diverge from reality..." This statement needs references. Why is this important? What non-linearities are you talking about? I recommend highlighting the impact on aerosol formation/microphysics from this artefact (e.g. Vattioni et al. 2024).

In this paragraph you should also discuss the effect of the vertical and horizontal resolution of the model, since it is known that this can affect simulations of volcanic plumes.

"This concentration can lead to lower $SO_2$ and aerosol mixing ratios in the mid- to long-term, as aerosols grow excessively large and subsequently sediment out of the stratosphere, as observed following the Nabro eruption.» You provide an explanation for lower aerosol mixing ratios, but what would be the reason for differences in $SO_2$ mixing ratios? Also this sentence (and the whole paragraph) needs references, since you don't show this with your results.

**Line 623-625:** What about the importance of the stratospheric entry point?

**Line 634:** "emission" is written twice.

**Line 691:** Conclusions last paragraph: This paragraph should be put into future tense (and or conjunctive), since like it is written know one could think that this is already provided or underway.

**Figure 1:** It is not very helpful showing code in the main manuscript since this will not be helpful to most readers. If at all I would put this into a supplement.

**Figure 2:** From just looking at the figure caption it is not clear what the "plume" refers to: $SO_2$, Aerosol in general, ash or sulfuric acid aerosol? Maybe specify in the caption what the IASI satellite measures.

**Figure 3:** "amount" is not very specific. I would call the unit by its name ($SO_2$ column).

**Figure 4:** The caption could be clearer. What do you mean by "shortly after"?
If you compare observations to the altitude of the "maximum $SO_2$ mixing ratio", only one altitude should be displayed in your plot, since there is only one maximum in the vertical column, right? I am confused here.
Maybe change the last sentence to: "In the simulations $SO_2$ was only injected into the stratosphere, except for mills_et_al".

**Figure 5:** See comments on Figure 4. Why don't you compare to OMI as well?

**Figure 6:** Maybe replace "zonally" with "zonally averaged". And also "study" with "simulations". I would skip "approximately" or be more specific. Does the date provided refer to the 5-day average or to the date of the eruption? There is no space between the first and the second panel, and the black line covers the "0". Please correct this.

**Figure 7:** The first sentence of the caption can be skipped or integrated into the second one.
What is the unit of the x axis? The format mm/dd is not used universal (in Europe dd/mm is more common). Thus, I suggest writing Jun 15, Jul 1, Jul 15 and so one, to make this clear.
Y-Axis label: It is "$SO_2$", not "SO2".

Why don't you show the spatial correlation for 15/6?

Concerning the "stratospheric cutoff altitude": Do you mean tropopause? If not, why don't you use the tropopause altitude? If yes, I would name the tropopause by its name. Why did you choose these altitudes? Did you check the sensitivity of your assumptions? Looking at the satellite data and your simulations, you can see that 3 days after the eruption a considerable amount of the plume is exactly around 30°N. Thus, slightly changing the "stratospheric cutoff altitude" might have an impact on the results shown here. This could for example be done, by providing plots with 1km higher and lower "stratospheric cutoff altitudes".

**Figure 8:** To me it is very hard to see any difference between the middle row and the lower row. Maybe it makes more sense to show the difference between the middle row columns and the lower row columns to better display the improvement (if there is any).

**Figure 10 and 11:** Same comment as on Figure 7. What does "stratospheric cutoff altitude mean" and how sensitive are results to this definition?

Change to: "... using the EMAC model with the new *EVER* historic volcanic setup (red) and..."

The axis label should read "$SO_2$", not "SO2"

**Figures 11 and 12:** To which wavelengths do the aerosol optical depths and extinctions refer to? This is crucial information which is missing.

---

## Referee Comment (RC2)

**Review comment on "New submodel for emissions from Explosive Volcanic Eruptions (EVER v1.1) within the Modular Earth Submodel System (MESSy, version 2.55.1)"**

The paper describes a new submodel within the MESSy framework for better simulation of mainly SO2 emissions from volcanic eruptions. Other potential applications are mentioned, such as emissions from wildfires, or volcanic degassing. In the study, the submodel is used to assess different distributions of SO2 emission distributions for two case studies. An explosive volcanic eruption (Nabro) and an effusive/surface emitting eruption (Kilauea). A setup for historical simulations from 2007 to 2011 is also presented. The model output is then discussed and compared with satellite retrievals.

The paper presents advances in modelling with the new submodel EVERv1.1 and is therefore fitting for GMD. EVER is a new tool to implement the emissions of SO2 from volcanic eruptions more flexibly and could be the basis for a more unified way to represent volcanic emissions in the future.

However, some restructuring of the text is needed to facilitate understanding. The manuscript is also very long and could be shortened considerably. I find the general structure of the presented manuscript very confusing. The methods used in this paper are generally stated but spread throughout the whole manuscript, which makes it difficult to read. I therefore recommend a more traditional structure with the introduction of the model, the different observations used in the analysis and the experiment description all in one dedicated chapter. Currently it reads more like the methods precede the respective result chapter and sometimes more information e.g. on the datasets is stated within the respective result sections. Also some information on the observational datasets is missing in the methodology. For example what gases are measured by what instrument. Some of this is explained later in the manuscript. There is also a section called "results" which would mean that everything before this is methodology? But this is not the case here. More detailed comments on this are below. This could also help making the paper more concise, since there are currently a few repetitions.

The results show how different set ups of the emissions can influence the SO2 plume. It is discussed thoroughly, how the different SO2 injections influence the plume evolution after an eruption and finally is able to show, how the new set-up improves this with respect to observations. It is mentioned, that the Asian Monsoon influences the plume evolution, "simultaneously probing the dynamics in the model". This is not followed up with the necessary sensitivity simulations or discussion on the implications of dynamics and performance of the model.

Since the code is not currently available, the results are currently not fully reproducible. However, the description in the paper should allow for a similar model to be constructed. But it is stated, that this submodel is portable to other base models, while it was only tested for three very specific modal models. Would this submodel not function with a different type of sectional model?

Some more previously published research on the topic should be considered. E.g. different injection scenarios for the Pinatubo eruption in various models by Quaglia et al. (2023).

While the title sufficiently describes the presented model, the abstract is currently still missing some comment on the models performance but rather only summarizes the methods. Some statement on the performance of the new model should be added here.

Also please check all abbreviations, some of the abbreviations are not explained while others are specified several times. Similarly, some of the sources don't conform to the GMD format.

Throughout the manuscript there are several very technical explanations and formulas, code snippets etc. that would fit better in the appendix or supplementary material. Figure 1 does not fit in this manuscript. This is only an example of a namelist, not a result and should be kept in the supplementary material. Particularly the first figure should represent the main results of the paper.

**Line by line comments**

**L20ff/L55ff:** Both of these paragraphs introduce heterogeneous chemistry on aerosol surfaces and ozone chemistry
**L134:** "Up to 800 volcanic eruptions" why is there a limit? Is it impossible to simulate 801 eruptions?
**L150:** This formula is very general and not new to this paper, it should therefore be moved to the supplements.
**Fig1:** The namelists are already in the supplementary material. This is also just an example and not a result. I suggest removing this figure and referring to supplementary material
**L143:** Unexplained acronym AER
**L156:** Why is this not possible?
**L170ff:** Is there a reference for how this is done in EVER 1.0?
**L186ff:** "and width" I find the term "width" misleading here as I understand that as spreading over several horizontal grid boxes. Or does this refer to the mass emitted? Vertical extent? If there is a fixed definition, please specify.
**L191ff:** There are important microphysical implications for this, see e.g. in Fig. 3 in Tilmes et al. (2023)
**L201:** What variables are nudged? What are the implications of nudging?
**L209:** What does simplified chemistry mean, is OH prescribed?
**L223ff:** The following paragraphs lack some consistency, consider also summarizing some of the key properties of these measurements in a table. E.g. resolution, extent, what gases are measured, time when they were/are in operation. Just from reading this, I am not sure where these measurements are used later. Is it about SO2 or aerosol, tropospheric or stratospheric?
**L313:** "width of 2km" again I find the use of the word "width" a bit confusing, do you mean vertical extent?
**L360ff:** This information on what limitations the different satellite instruments/retrievals have fits better in the methods section with the description of the instruments/datasets.
**L374:** "AKM" was already defined previously, it is again defined several times throughout the manuscript.
**L376:** "Vertical width"
**L473ff:** This reads like it should be in the discussion/conclusion
**L479ff:** Parts of these section introduce more motivation for the study again, which belongs in the introduction. I also recommend combining the methodology with the ones in the other chapters.
**L509:** The title for this subsection is misleading, is everything before this methodology?
**L595:** What about availability of oxidants in these spatially confined plumes? There would also be some differences between simplified chemistry and a more sophisticated chemistry scheme.
**L634:** "emission emission"
**L639:** How are eruptions after 2012 currently simulated?

**L915-925:** Check sources/format

**References**

Tilmes, S., Mills, M. J., Zhu, Y., Bardeen, C. G., Vitt, F., Yu, P., Fillmore, D., Liu, X., Toon, B., and Deshler, T.: Description and performance of a sectional aerosol microphysical model in the Community Earth System Model (CESM2), Geosci. Model Dev., 16, 6087–6125, https://doi.org/10.5194/gmd-16-6087-2023, 2023.

Quaglia, I., Timmreck, C., Niemeier, U., Visioni, D., Pitari, G., Brodowsky, C., Brühl, C., Dhomse, S. S., Franke, H., Laakso,A., Mann, G. W., Rozanov, E., and Sukhodolov, T.: Interactive stratospheric aerosol models' response to different amounts andaltitudes of SO2 injection during the 1991 Pinatubo eruption, Atmos. Chem. Phys., 23, 921–948, https://doi.org/10.5194/acp-23-921-2023, 2023.

---

## Author Comment (AC1)

**Reply to general comments of RC1:**

We thank the reviewer for the time to review our manuscript, and especially the very detailed and constructive feedback. We reply to the general comments before the end of the discussion phase, while we will present a detailed reply on the specific comments once the discussion phase is concluded. We report the comments (grey, bold) along with our replies (blue).

**The submitted manuscript contains the description of a new submodule within MESSy. The submodule is described in detail in section 2. Section 3 provides a description of the model setup for the validation of the submodel. For model validation the 2011 Nabro and 2018 Kilauea $SO_2$ emission plumes were evaluated with different sensitivity simulations and with a fine-resolution simulation, respectively. In chapter 4 the explosive volcanic $SO_2$ injections of the time period 2007 to 2011 was simulated based on a 3-D observational emission dataset which was improved based on the findings of chapter 3.**

**The submitted manuscript describes a new submodule for emission of volcanic substances and provides some validation for the evolution of $SO_2$ plumes after volcanic eruptions. However, in my point of view, the manuscript has some substantial flaws, which I explain below and which need to be addressed. Even though topic wise the manuscript would be a good fit for publication in GMD, I do not think that quality wise this manuscript deserves publication in this journal.**

We thank the reviewer for the assessment of our manuscript. We agree that there are some mentioned shortcomings that have to be addressed as pointed out, however, we disagree on the overall evaluation. We will discuss the general comments in detail below.

**The main environmental impact of large explosive volcanic eruption on a global scale (heterogeneous chemistry, cooling of the surface, local warming of the stratosphere) are effects from sulfuric acid aerosols and not from $SO_2$. The manuscript almost entirely focuses on the simulation of $SO_2$ plumes and thus, a crucial aspect of volcanic emission plumes is missing. I know that for the accurate simulation of aerosol burden and their effects it is important to accurately simulate the SO2 plume in the first place, however, if this was the goal of this study this needs to be motivated and explained in the introduction.**

We agree with the reviewer that the main environmental impact of volcanic eruptions comes from sulfuric acid aerosols. However, as the reviewer pointed out, these sulfuric acid aerosols are a result of volcanic $SO_2$ emissions. We also state this twice in the introduction: "Moreover, $SO_2$ undergoes oxidation to form sulfuric acid $H_2SO_4$, which rapidly converts to the aerosol phase forming sulfate under most atmospheric conditions." and "When volcanic $SO_2$ emissions reach the stratosphere, the subsequently formed sulfate aerosols enhance the stratospheric aerosol burden and are distributed widely across the globe". Therefore, for the evaluation of the EVER submodel, $SO_2$ is the correct tracer that needs to be investigated and evaluated. Nevertheless, the effects of sulfuric acid aerosol are described in the introduction, as they have the most important impact, and they result from volcanic $SO_2$ emissions.

Indeed, the goal of the study is to present a new submodel that handles emissions of gaseous and aerosol species, and the accurate simulation of volcanic $SO_2$. The subsequent formation of sulfuric acid and aerosols is a consequence of chemistry, thermodynamics and microphysics.

This is handled by the other submodels present in the MESSy model system (e.g. MECCA and GMXe), which have been largely evaluated elsewhere and therefore are not subject of this paper. We also state this in the introduction: "Our primary aim is to provide recommendations regarding the implementation of stratospheric $SO_2$ injections from volcanic eruptions within the MESSy framework ...", "We evaluate the EVER submodel using $SO_2$ emissions from the 2011 Nabro explosive volcanic eruption ..." and "Based on the stratospheric volcanic $SO_2$ emission inventory developed by Schallock et al. (2023), we establish a default setup for the EVER submodel ...". We want to make this clearer in the abstract and the introduction.

Regarding the comment "a crucial aspect of volcanic emission plumes is missing": We do not agree on that. The EVER submodel is only responsible for the emission of the initial plume species. Sulfuric acid aerosols are not part of the "volcanic emission plume". They form in the evolution of the plume. Standard aerosol simulations with EMAC (or MESSy in general) already included the chemistry, thermodynamics and microphysics of sulfuric acid aerosol (see for example Brühl et al., 2018; Schallock et al., 2023), however it was missing a standard setup for including the initial $SO_2$ emissions, which we wanted to address.

As we agree, that the long-term $SO_2$ mixing ratios (Sect. 3.3.2) and the resulting aerosol evaluated later depend on the $SO_2$ removal and microphysics, we will talk about their handling in more detail in a revised version (see also comments below), but this is a more technical comment in our opinion. Moreover, the chemical mechanism, including $H_2SO_4$ formation, is already provided as supplement, which we refer to in the manuscript.

**To me many important aspects of a paper are not sufficiently explained: What is the novelty of this study? What can the model do what other models can't do? Why is it important to simulate the emissions of $SO_2$ accurately? Why do we need yet another model which is capable of simulating volcanic injections? What have other models shown and how does your model compare to other models?**

This manuscript is a documentation of a new submodel, along with its evaluation and the development of a default setup. It is not a study, aiming to provide new results on volcanic eruptions. This is well within the scope of GMD papers. From the "Aims and Scope" of GMD papers (`https://www.geoscientific-model-development.net/about/aims_and_scope.html`) amongst others:

- "geoscientific model descriptions, from statistical models to box models to GCMs"

- "development and technical papers, describing developments such as new parameterizations or technical aspects of running models such as the reproducibility of results"

We chose the manuscript type "Development and technical papers" (from `https://www.geoscientific-model-development.net/about/manuscript_types.html`): "These papers describe technical developments relating to model improvements such as the speed or accuracy of numerical integration schemes as well as new parameterisations for processes represented in modules. [...] Development and technical papers usually include a significant amount of evaluation against standard benchmarks, observations, and/or other model output as appropriate."

We think that most of the raised questions are already sufficiently answered within the scope of a GMD paper.

- **"What is the novelty of this study?"** In MESSy, we so far only had a more or

less generic submodel (TREXP, Jöckel et al., 2010) for point or column like emissions in the atmosphere. This submodel, however, was made for different purposes, and missing features which are particularly required for emissions from volcanic eruptions (e.g. the possibility to specify vertical emission profile shapes or primary aerosol emissions as needed for volcanic ash). To avoid increasing the complexity of TREXP, we decided to implement EVER. We describe this in lines 125ff. In previous studies a lot of manual work was necessary to consider explosive volcanoes and other injections into the stratosphere, and there was no standard setup covering stratospheric volcanic eruptions.

- **"What can the model do what other models can't do?"** Indeed, other models can also handle volcanic eruptions; however, in the MESSy model system, it was not possible so far. So, we don't see any contradiction here. We present our new submodel and evaluate it using observations. The implementation differs from other models being specific to the MESSy interface, although we can refer more to other model developments.

- **"Why is it important to simulate the emissions of $SO_2$ accurately?"** Details about the importance of sulfuric acid aerosols resulting from volcanic $SO_2$ emissions are listed in the introduction. This implies the importance of accurately simulating $SO_2$ emissions.

- **"Why do we need yet another model which is capable of simulating volcanic injections?"** As there was none in MESSy, and other available models were not sufficient for our needs, this paper described a further development of the code. We will make it clearer in the introduction, why we chose an own implementation, and thank the reviewer for this suggestion. We will also add a more detailed part in the introduction, where we discuss available volcanic models.

- **"What have other models shown and how does your model compare to other models?"** While we mostly focused on comparison with observations, we will highlight the differences and similarities to other model studies also in a revised version.

To avoid further confusion, we want to provide a clearer description of the nature and the goal of the study in the abstract and the introduction.

I think the manuscript does not provide/discuss a substantial part of literature relevant for the simulation of volcanic emission plumes. There have been many studies, which highlight important aspects discussed in this manuscript, which are not even cited (e.g. Brodovsky et al. 2021 or Quaglia et al. 2023).

We thank the reviewer for the comment. We will include more model literature in a revised version, and agree with the reviewer that this literature is crucial to discuss.

Usually, the 1991 Mt. Pinatubo eruption is used as a basis for model validation (e.g., Quaglia et al. 2023, which even includes EMAC simulations). Why did you not perform a simulation of Mt. Pinatubo to validate your model? Probably because this has been done extensively already, but it would allow comparing also with other models.

We chose the Nabro volcano, as there are extensive $SO_2$ satellite observations available on this eruption, including 3-D information from MIPAS. This is not the case for the Pinatubo eruption, and there are large uncertainties in observations. We will add this explanation on

why we chose Nabro.

We also want to clarify, that this study is not a specific study on the impacts of volcanic eruptions. We want to present the new submodel, evaluate it, and provide a default setup to achieve a realistic $SO_2$ emission plume. We do not concentrate on extreme events, such as the Pinatubo eruption, but we include various eruptions plumes. The goal is to provide a standard setup that includes accurate $SO_2$ emissions also for studies not focussing on volcanic eruptions. The extreme event of the Pinatubo did not appear to be a suitable candidate for this evaluation.

**The introduction could focus more on topics relevant to the rest of the manuscript. E.g. why do you explain ozone chemistry and ash in such detail if it is not relevant to the manuscript? The manuscript provides a very broad discussion about the general impact of volcanic eruptions on climate and atmospheric chemistry, but it does not discuss the problems and current limitations when modeling volcanic eruptions (e.g., spatial & temporal resolution, model agreement/disagreement of past model studies among each other and with observations, sectional versus modal microphysics modules, previous studies which addressed the vertical/horizontal distribution of volcanic emission plumes). Since you submit to "Geoscientific model development" I think a more technical motivation/introduction would be appropriate. I think the introduction should clearly motivate the research so that it is clear to a reader why the submodule was developed and why there is need for the research provided. In my point of view this is not the case.**

We thank the reviewer for this suggestion. We chose to focus more on the general volcanic impacts to motivate why this submodel is needed within MESSy, and not necessarily why it is needed in addition to other available volcanic emission routines in various models. We agree with the reviewer, that a more technical focus would improve the introduction, and we will discuss limitations in modeling volcanic eruptions more detailed in a revised version. We did motivate the research by showing, why it is crucial to have this submodel in MESSy, but we can make it clearer, why we did not merely take an existing routine from other models.

**Your conclusions just summarize what you have done in your work, including a brief recapitulation of some qualitative results with a brief outlook for further potential applications of the sub-model in the end. However, I think the conclusion section should summarize the key findings of the paper (also some quantitative statements, not only qualitative) and put them into a broader context. The conclusions should demonstrate the importance of the paper and convey the larger implications of the paper to the research field. Additionally, I also find it important to address the limitations of the research/model provided (or more specifically: of the presented sub-module) and highlight aspects identified in your work which require further research and development. I don't think these points are provided in the current version of the conclusion section of the manuscript.**

Yes, we mostly summarized our work, but in addition, we also summarize our key findings, e. g. "Horizontal position and emission timing were found to have a minor impact on the long-term $SO_2$ burden in the stratosphere. Nevertheless, these parameters play a crucial role in detailed process studies during the initial weeks after an eruption. Overall, we conclude that simulations of volcanic eruptions can be effectively performed with the help of 3D-, column,

and point emissions. The optimal approach depends on the specific use case, with column emissions excelling in the short-term, and similar performance in the long-term" and " [...] , and we advocate its inclusion in simulations using the MESSy framework focusing on the upper troposphere and stratosphere. For very strong eruptions, it may be beneficial to distribute the emissions over multiple horizontal grid boxes and an extended time period, or adjust the vertical plume width, if discrepancies with observations occur".

We agree, that we can put them into some broader context with previous findings, and potentially point to some limitations and shortcomings, discussed in detail in the Discussion. Additionally, we will outline further necessary research and development in a revised version. However, a technical paper describing a new submodel, evaluating it, and developing a default setup does not need "larger implications of the paper to the research field" but rather vital informations for the people using the submodel. Larger scientific implications to the research field can result from future studies using the submodel.

Similarly, the abstract just summarizes what you did in this study, but you do not provide any results or conclusions or broader implications to the research fields in the abstract, which I think is an essential part of the abstract.

As outlined above, this is a technical paper describing a new submodel, evaluating it, and developing a default setup. The conclusion basically is, that it was developed, it works and it can reproduce observations. There are no real broader implications for the research field, apart from the findings mentioned in the conclusion (see above). However, broader implications are not required in a "model description" manuscript in GMD, as this one is.

An important aspect which is missing in the analysis and discussion of your results is the role of chemical loss of $SO_2$. In addition to dilution and transport, a (maybe even more) important aspect which determines the $SO_2$ lifetime is chemical $SO_2$ loss (i.e., mainly oxidation with OH and $O_3$). How is this represented in MECCA, and how does this influence your modelled results. You also write that for the Kilauea study, you applied "simplified chemistry". What does this mean? Since chemical loss is very important for the $SO_2$ lifetime you need to discuss how this affects the results in your study. How is the reaction with $O_3$ and OH represented in MECCA? Were the $O_3$ and OH fields also nudged to observations like written in the model description (probably not)? How does this influence the modelled results?

We agree with the reviewer that the chemical loss of $SO_2$ is important, and actually is the reason for sulfuric acid aerosols to form. In the supplement, we provide the full chemical mechanism for both simulations, the stratospheric one and the "simplified chemistry" for the Kilauea simulation. In a revised version, we will also cover this in the manuscript.

While we treat $SO_2$ oxidation in both setups as described in the supplement, we do not consider oxidation with $O_3$. This reaction only takes place in the aqueous phase and is not important for the stratosphere (e. g. Kremser et al., 2016). $O_3$ and OH are part of the chemical mechanism (see supplement) and are not nudged towards observations. The described nudging affects only the model dynamics using meteorological reanalysis data.

Another aspect which is not discussed is the importance of aerosol microphysics. You show an in-depth analysis of the SO2 plume evolution and then in chapter 4

you suddenly come up with AOD analysis. However, there is an important step missing in between: Aerosol Microphysics. You write that you are using GMXE as an aerosol microphysics module and represent the aerosols using a modal approach. However, you do not provide any information about resulting aerosol burden, aerosol size distribution ect. You only show AOD and extinction (without indicating the wavelengths under consideration). Thus, the whole aerosol microphysics (which is a very important aspects when simulating volcanic emission plumes) is treated as a black box in this manuscript. In my point of view, it is crucial to also show resulting aerosol burden and compare them with observations as well as which other models (e.g. see Brodowsky et al. 2021).

We agree with the reviewer, that aerosol microphysics is a very important aspect for the modelling of volcanic aerosol, and we can provide more detail on the representation in GMXe. Thank you for pointing this out. However, we want to make clear again, that this manuscript is not aiming to evaluate the aerosol microphysics or study the sensitivity of the results to the aerosol microphysics.

We thank the reviewer for pointing out to Brodowsky et al., 2021, and we apologize for not including it in the initial submission. We will provide a comparison of resulting sulfur and sulfate burden in a revised version, and point to the discussed uncertainties. Nevertheless, in our opinion our study has a different focus, as we want to evaluate the EVER submodel. Thus, we look into the uncertainties in the $SO_2$ emission parameters, while not considering uncertainties in aerosol microphysics in great detail. Our manuscript is a "Development and technical paper" in Geoscientific Model Development, while the Brodowsky et al., 2021 paper is a scientific study in JGR Atmoheres focusing on the evolution of sulfate aerosol, and thus focuses more on the production of $H_2SO_4$ and the aerosol burden.

We apologize for not mentioning the wavelengths under consideration, and will do so in future versions. Additionally, we will compare our results to previous model results. We decided to focus on comparison to observations of $SO_2$, as this is mostly influenced by the EVER submodel, and we wanted to show that we can reproduce observed $SO_2$ mixing ratios. As pointed out, resulting aerosol burden depends on a lot of different factors, the evaluation of which goes beyond the scope of this manuscript. However, we plan to include aerosol burdens as well in a revised version to motivate the evaluation using AOD and extinction.

Another aspect which comes too short in the discussion section is the influence of the spatial and temporal resolution of different processes. It is well known that these aspects are very important for realistic representation of volcanic plumes. While for the troposphere the horizontal resolution is more important in the stratosphere the vertical resolution is more important. For aerosol microphysics the microphysical timestep should be set small enough to realistically simulate nucleation and condensation. You could mention these aspects in the discussion of your results.

Yes, the spatial resolution is very important for the modelling of volcanic eruptions. For that reason, we chose the maximum available horizontal resolution for the tropospheric study of Kilauea (T255), and the maximum available vertical resolution (90 levels) for the stratospheric study as also used as maximum vertical resolution in Brodowsky et al., 2021. The evaluation of different resolutions again goes beyond the scope of the paper, but we can discuss these aspects,

yes. Thank you for the suggestion.

We do not distinguish the chemical and microphysical timestep in our simulation, as the general model timestep length is only 8 minutes (we will mention the timestep length in the manuscript), and chemistry and microphysics are calculated in each timestep. This is different to the recent publication by the reviewer,Vattioni et al., 2024, where chemical and microphysical timestep differ. In Vattioni et al., 2024, the reviewer concluded, that in volcanic simulations, first considering condensation and nucleation results in the smallest numerical error. This is the default in GMXe, and thus we did not discuss it. Further reducing the timestep will of course improve the results, but is not computationally feasible. We additionally argue, that the chemical timestep length of 2 hours (in Vattioni et al., 2024) will lead to different errors in microphysics compared to the 8 minute chemical timestep length in our EMAC simulations. Moreover, the main focus of the manuscript is on the initial $SO_2$ plume, that is not influenced by the microphysical settings.

I also find the manuscript too long. Many aspects which are discussed in the introduction and submodel description are not relevant to the storyline or are not picked up again in the discussion. I suggest shortening substantially and putting part of the text (e.g. description of code and namelists) into the supplement. Also, the structure could be improved. For example, the model description and setup and description of the observations (sect. 3.1 and sect. 3.2) could be a chapter for its own or part of chapter 2. Because the setup described there is also used in chapter 4. The different events simulated here (i.e., Nabro, Kilauea and the 2007-2011 period) seem a little disconnected to each other. Why don't you show a full analysis of only one event (e.g. Nabro), but in more detail including sulfuric acid aerosol burden etc.

Thank you for this suggestion. We agree, that we can shorten the introduction with respect to the discussion of the general impacts of volcanic eruptions. However, we believe that in a technical paper on the implementation of a new submodel, the description of the code and namelist is essential, and is common practice in numerous GMD papers. Moreover, we do not believe that the submodel description in a technical paper has to necessarily follow the storyline of the evaluation, but should instead focus on completeness.

Regarding the restructuring of the manuscript, we agree that we could make 3.1 and 3.2 a separate Section, as the setup is also used in Sect. 4. Nevertheless, we are hesitant to include this in Sect. 2, as we wanted to focus solely on the new submodel in this section.

We understand, that the different events seem to be disconnected. We will motivate the methodology in more detail in a revised version. We chose the different events for the following different reason:

- Single analysis of the Nabro volcano to evaluate the submodel in the stratosphere, and perform sensitivity studies on the emission parameters of the submodel.

- Kilauea study to show the applicability of the submodel for degassing volcanoes

- 2008-2011 period to evaluate the historic namelist setup developed

For this reason, we want to keep the current structure. Nevertheless, we can analyse the Nabro eruption in more detail (including resulting burdens).

**References**

Brodowsky, C., T. Sukhodolov, A. Feinberg, M. Höpfner, T. Peter, A. Stenke, and E. Rozanov (2021). "Modeling the Sulfate Aerosol Evolution After Recent Moderate Volcanic Activity, 2008–2012". In: *Journal of Geophysical Research: Atmospheres* 126.23, e2021JD035472. DOI: `https://doi.org/10.1029/2021JD035472`.

Brühl, C., J. Schallock, K. Klingmüller, C. Robert, C. Bingen, L. Clarisse, A. Heckel, P. North, and L. Rieger (2018). "Stratospheric aerosol radiative forcing simulated by the chemistry climate model EMAC using Aerosol CCI satellite data". In: *Atmospheric Chemistry and Physics* 18.17, pp. 12845–12857. DOI: `10.5194/acp-18-12845-2018`.

Jöckel, P., A. Kerkweg, A. Pozzer, R. Sander, H. Tost, H. Riede, A. Baumgaertner, S. Gromov, and B. Kern (2010). "Development cycle 2 of the Modular Earth Submodel System (MESSy2)". In: *Geoscientific Model Development* 3.2, pp. 717–752. DOI: `10.5194/gmd-3-717-2010`.

Kremser, S., L. W. Thomason, M. von Hobe, M. Hermann, T. Deshler, C. Timmreck, M. Toohey, A. Stenke, J. P. Schwarz, R. Weigel, S. Fueglistaler, F. J. Prata, J.-P. Vernier, H. Schlager, J. E. Barnes, J.-C. Antuña-Marrero, D. Fairlie, M. Palm, E. Mahieu, J. Notholt, M. Rex, C. Bingen, F. Vanhellemont, A. Bourassa, J. M. C. Plane, D. Klocke, S. A. Carn, L. Clarisse, T. Trickl, R. Neely, A. D. James, L. Rieger, J. C. Wilson, and B. Meland (2016). "Stratospheric aerosol—Observations, processes, and impact on climate". In: *Reviews of Geophysics* 54.2, pp. 278–335. DOI: `10.1002/2015RG000511`.

Schallock, J., C. Brühl, C. Bingen, M. Höpfner, L. Rieger, and J. Lelieveld (2023). "Reconstructing volcanic radiative forcing since 1990, using a comprehensive emission inventory and spatially resolved sulfur injections from satellite data in a chemistry-climate model". In: *Atmospheric Chemistry and Physics* 23.2, pp. 1169–1207. DOI: `10.5194/acp-23-1169-2023`.

Vattioni, S., A. Stenke, B. Luo, G. Chiodo, T. Sukhodolov, E. Wunderlin, and T. Peter (2024). "Importance of microphysical settings for climate forcing by stratospheric $SO_2$ injections as modeled by SOCOL-AERv2". In: *Geoscientific Model Development* 17.10, pp. 4181–4197. DOI: `10.5194/gmd-17-4181-2024`.

---

## Author Response (AR1)

Dear Editor,

we thank you very much for editing our manuscript. We highly valued the reviewer comments, and we believe that we could strongly improve the manuscript based on them. We provide the updated manuscript along with the track-changes. The detailed point-to-point replies can be found below.

The largest change was the restructuring of the manuscript, clearer distinguishing between the submodel description, the methods and the results. Unfortunately, this leads to a slightly confusing track-changes file. The major according structural changes are provided in the replies to the general comments below. All further major changes are described below, in addition to the track-changes file.

Note, that we decided to show the sensitivity on the lower integration limit, and the information on the optimization for the degassing volcanic event in the appendix instead of the supplement (as we had stated in the replies).

Thank you again and best regards,

*Matthias Kohl (on behalf of all co-authors)*

**Reply to comments of RC1:**

We thank the reviewer for the time to review our manuscript, and especially the very detailed and constructive feedback. After the initial reply on the general comments, we now want to address all comments in detail. Here we also repeat and extend the answers to the general comments already posted in the open discussion for completeness, now also considering the comments of the second reviewer and the revised manuscript. We report the comments (grey, bold) along with our replies (blue).

**The submitted manuscript contains the description of a new submodule within MESSy. The submodule is described in detail in section 2. Section 3 provides a description of the model setup for the validation of the submodel. For model validation the 2011 Nabro and 2018 Kilauea $SO_2$ emission plumes were evaluated with different sensitivity simulations and with a fine-resolution simulation, respectively. In chapter 4 the explosive volcanic $SO_2$ injections of the time period 2007 to 2011 was simulated based on a 3-D observational emission dataset which was improved based on the findings of chapter 3.**

**The submitted manuscript describes a new submodule for emission of volcanic substances and provides some validation for the evolution of $SO_2$ plumes after volcanic eruptions. However, in my point of view, the manuscript has some substantial flaws, which I explain below and which need to be addressed. Even though topic wise the manuscript would be a good fit for publication in GMD, I do not think that quality wise this manuscript deserves publication in this journal.**

We thank the reviewer for the assessment of our manuscript. We agree that there are some shortcomings that need to be addressed, however, we disagree on the overall evaluation. We will discuss the comments in detail below.

**The main environmental impact of large explosive volcanic eruption on a global scale (heterogeneous chemistry, cooling of the surface, local warming of the stratosphere) are effects from sulfuric acid aerosols and not from $SO_2$. The manuscript almost entirely focuses on the simulation of $SO_2$ plumes and thus, a crucial aspect of volcanic emission plumes is missing. I know that for the accurate simulation of aerosol burden and their effects it is important to accurately simulate the SO2 plume in the first place, however, if this was the goal of this study this needs to be motivated and explained in the introduction.**

We agree with the reviewer that the main environmental impact of volcanic eruptions comes from sulfur aerosol (we prefer to talk about "sulfur aerosol" in contrast to "sulfuric acid aerosol", as it also includes sulfate and bi-sulfate species such as ammonium (bi-)sulfate; see also later specific comment). However, as the reviewer pointed out, these sulfur aerosols are a result of volcanic $SO_2$ emissions. We also stated this twice in the introduction of the submitted manuscript: "Moreover, $SO_2$ undergoes oxidation to form sulfuric acid ($H_2SO_4$), which rapidly converts to the aerosol phase forming sulfate under most atmospheric conditions." and "When volcanic $SO_2$ emissions reach the stratosphere, the subsequently formed sulfate aerosols enhance the stratospheric aerosol burden and are distributed widely across the globe". Therefore, for the evaluation of the EVER submodel (handling the initial volcanic emissions), $SO_2$ is the correct tracer that needs to be investigated and evaluated. Nevertheless, the effects of sulfur aerosol are described in the introduction, as they have the most important impact, and they

result from volcanic $SO_2$ emissions.

Indeed, the goal of the study is to present a new submodel that handles emissions of gaseous and aerosol species, and the accurate simulation of volcanic $SO_2$. The subsequent formation of sulfuric acid and aerosols is a consequence of chemistry, thermodynamics and microphysics. This is handled by the other submodels present in the MESSy model system (e.g. MECCA (Sander et al., 2019) and GMXe (Pringle et al., 2010)), which have been evaluated elsewhere and therefore are not subject of this paper. We also state this in the introduction: "Our primary aim is to provide recommendations regarding the implementation of stratospheric $SO_2$ injections from volcanic eruptions within the MESSy framework ...", "We evaluate the EVER submodel using $SO_2$ emissions from the 2011 Nabro explosive volcanic eruption ..." and "Based on the stratospheric volcanic $SO_2$ emission inventory developed by Schallock et al. (2023), we establish a default setup for the EVER submodel ...". In the revised version, we make the main goal of the study clearer in the introduction, and distinguish it from previously published studies.

Regarding the comment "a crucial aspect of volcanic emission plumes is missing": We disagree on that. The EVER submodel is only responsible for the emission of the initial plume species. Sulfur aerosols are not part of the "volcanic emission plume". They form in the evolution of the plume. Standard aerosol simulations with EMAC (or MESSy in general) already included the chemistry, thermodynamics and microphysics leading to the formation of sulfur aerosol from $SO_2$ (see for example Brühl et al. (2018) and Schallock et al. (2023)), however it was missing a standard setup for including the initial $SO_2$ emissions, which we wanted to address.

As we agree, that the mid-term $SO_2$ mixing ratios (Sect. 3.3.2 of the submitted manuscript) and the resulting aerosol evaluated later depend on the $SO_2$ removal and microphysics, we talk about their handling in more detail in the revised version (see also comments below), but this is a more technical comment in our opinion. Moreover, the chemical mechanism, including $H_2SO_4$ formation, is already provided as supplement, which we refer to in the manuscript. We also want to add, that the investigation of the resulting AOD is just an additional validation, and should not be seen as a volcanic aerosol study. We apologize, if we implied otherwise, and we clarified this in the revised version.

**To me many important aspects of a paper are not sufficiently explained: What is the novelty of this study? What can the model do what other models can't do? Why is it important to simulate the emissions of $SO_2$ accurately? Why do we need yet another model which is capable of simulating volcanic injections? What have other models shown and how does your model compare to other models?**

This manuscript is a documentation of a new submodel, along with its evaluation and the development of a default setup. It is not a study, aiming to provide new results on volcanic eruptions. This is well within the scope of GMD papers. From the "Aims and Scope" of GMD papers (`https://www.geoscientific-model-development.net/about/aims_and_scope.html`) amongst others:

- "geoscientific model descriptions, from statistical models to box models to GCMs"

- "development and technical papers, describing developments such as new parameterizations or technical aspects of running models such as the reproducibility of results"

We chose the manuscript type "Development and technical papers" (from `https://www.geoscientific-`

model-development.net/about/manuscript_types.html): "These papers describe technical developments relating to model improvements such as the speed or accuracy of numerical integration schemes as well as new parameterisations for processes represented in modules. [...] Development and technical papers usually include a significant amount of evaluation against standard benchmarks, observations, and/or other model output as appropriate."

We believe that most of the raised questions are already sufficiently answered within the scope of a GMD paper.

- **"What is the novelty of this study?"** In MESSy, we so far only had a more or less generic submodel (TREXP, Jöckel et al. (2010)) for point or column like emissions in the atmosphere. This submodel, however, was made for different purposes, and missing features which are particularly required for emissions from volcanic eruptions (e.g. the possibility to specify vertical emission profile shapes or primary aerosol emissions as needed for volcanic ash). To avoid increasing the complexity of TREXP, we decided to implement EVER. We describe this in lines 125ff. In previous studies a lot of manual work was necessary to consider explosive volcanoes and other injections into the stratosphere, and there was no standard setup covering stratospheric volcanic eruptions. In addition, we provide recommendations on the implementation of the initial $SO_2$ emissions of explosive volcanic eruptions, based on the available emission inventories, as Brodowsky et al. (2021) and Quaglia et al. (2023) found, that the different emission databases lead to vastly differing results. This can be mostly attributed to the unclear fraction of the volcanic plume in the stratosphere. We now provide more information on that in the abstract, introduction, discussion and conclusions in the revised manuscript.

- **"What can the model do what other models can't do?"** Indeed, other models can also handle volcanic eruptions; however, in the MESSy model system, it was not possible so far. So, we don't see any contradiction here. We present our new submodel and evaluate it using observations. The implementation differs from other models being specific to the MESSy interface. Additionally, we studied the correct emission location, timing and stratospheric mass that has to be implemented, while previous studies (e. g. Brodowsky et al., 2021) only used some assumptions, on which mass fraction from the emission inventory of Brühl et al. (2018) has to be emitted in the stratosphere, leading to vastly differing stratospheric $SO_2$ burdens. We refer to other model developments in more detail in the introduction and the discussion of the revised version.

- **"Why is it important to simulate the emissions of $SO_2$ accurately?"** Details about the importance of sulfur aerosol resulting from volcanic $SO_2$ emissions are listed in the introduction. This implies the importance of accurately simulating $SO_2$ emissions. The results also show, how differing $SO_2$ emission parameters lead to differing results. We include these findings in the abstract and in the conclusion in the revised version.

- **"Why do we need yet another model which is capable of simulating volcanic injections?"** As there was none in MESSy, and other available models were not sufficient for our needs, this paper described a further development of the code. We made it clearer in the introduction, why we chose an own implementation, and thank the reviewer for this suggestion. The most important reason is the handling of the position, timing and stratospheric mass of the initial $SO_2$ emission, that was not investigated in previous volcanic timeline studies. We added more detail in the introduction, where we discuss

available volcanic models and emission inventories.

- **"What have other models shown and how does your model compare to other models?"** The main goal of the paper is to achieve reasonable agreement with observations to provide a standard setup. For that reason, we mostly focused on the observational side as the validation by observations is more important than comparisons to other model simulations in our opinion. However, we agree with the reviewer, that it is important to discuss other models, and we highlighted the differences and similarities to other model studies in the revised version in the introduction (see also next comment).

To avoid further confusion, we generally provide a clearer description of the nature and the goal of the study in the introduction, as mentioned in the points above.

**I think the manuscript does not provide/discuss a substantial part of literature relevant for the simulation of volcanic emission plumes. There have been many studies, which highlight important aspects discussed in this manuscript, which are not even cited (e.g. Brodovsky et al. 2021 or Quaglia et al. 2023).**

We thank the reviewer for the comment. We included and discussed more model literature (including the recommended ones) in the introduction of the revised version, and we agree with the reviewer that this literature is crucial to discuss:

"Volcanic $SO_2$ emission databases are the basis for the correct implementation of volcanic eruptions in atmospheric models. Timmreck et al. (2018) recommend four different emission inventories. Neely III and Schmidt (2016) and Mills et al. (2016) provide an inventory for tropospheric and stratospheric volcanoes, covering daily emissions and providing plume top and minimal height. Carn et al. (2017) provide a very detailed list of tropospheric and stratospheric volcanoes, however not distinguishing between the tropospheric and stratospheric part of the plume. Brühl et al. (2018) developed a volcanic $SO_2$ emission database from 1990 to 2021 (updated by Schallock et al., 2023), focusing only on the stratospheric part of the plume, also including smaller eruptions. Finally, the emission database from Diehl et al. (2012) only covers volcanoes up to 2010.

The treatment of $SO_2$ from volcanic eruptions based on the available emission inventories in global atmospheric models varies widely (e. g. Quaglia et al., 2023; Timmreck et al., 2018). Quaglia et al. (2023) performed a model intercomparison study focusing on the Pinatubo eruption, finding that inter- and intra-model differences in the response of $SO_2$ and sulfur aerosol on the Pinatubo eruption are large for a range of sensitivity experiments. The differences were mostly attributed to differing stratospheric transport, emission databases, aerosol microphysics and stratospheric chemistry. Vattioni et al. (2024) investigated the impact of microphysical settings within the SOCOL-AERv2 aerosol chemistry-climate model and found that the microphysical timestep, as well as the order of the microphysical processes can lead to vastly differing results in stratospheric aerosol burden.

Brodowsky et al. (2021) studied small- and medium-sized volcanic eruptions from 2008 to 2012, investigating the sensitivity of the resulting aerosol burdens to the different emission databases, internal model variability, dynamic nudging and the vertical resolution. The largest uncertainties resulted from the emission databases and their application. Most volcanic model studies inject $SO_2$ in columns at the geographical location of the volcano (e. g. Brodowsky et al., 2021; Mills et al., 2016; Quaglia et al., 2023; Schmidt et al., 2018), however the vertical extent

of the column is mostly unknown and depends on a number of assumptions. For instance, Brodowsky et al. (2021) emitted the recommended $SO_2$ burdens for the emission inventories of Brühl et al., 2018; Carn et al., 2017; Diehl et al., 2012 from the prescribed plume top height down a third of the distance to the Earth's surface. However, this approach highly depends on the emission inventory derivation, i. e. if only the stratospheric plume is considered for the derived $SO_2$ burden (as in the database from Brühl et al. (2018)) or the tropospheric part of the plume as well (as in the databases from Carn et al. (2017) and Diehl et al. (2012)). "

Usually, the 1991 Mt. Pinatubo eruption is used as a basis for model validation (e.g., Quaglia et al. 2023, which even includes EMAC simulations). Why did you not perform a simulation of Mt. Pinatubo to validate your model? Probably because this has been done extensively already, but it would allow comparing also with other models.

We chose the Nabro volcano, as there are extensive $SO_2$ satellite observations available on this eruption, including 3-D information from MIPAS. This is not the case for the Pinatubo eruption, and there are large uncertainties in observations. We added an explanation on why we chose Nabro: "As it was observed by a number of satellite instruments, it offers a perfect case study to investigate the spatio-temporal evolution of volcanic $SO_2$ in the stratosphere."

We also want to clarify, that this study is not a specific study on the impacts of volcanic eruptions. We want to present the new submodel, evaluate it, and provide a default setup to achieve realistic $SO_2$ emission plumes for the last three decades. We do not concentrate on case studies and their atmospheric consequences, such as the Pinatubo eruption; instead we include a large number of eruption plumes reaching the stratosphere. The goal is to provide a standard setup that includes reliable $SO_2$ emissions also for studies not focusing on volcanic eruptions, especially in the stratosphere region. The extreme event of the Pinatubo did not appear to be a suitable candidate for this evaluation.

The introduction could focus more on topics relevant to the rest of the manuscript. E.g. why do you explain ozone chemistry and ash in such detail if it is not relevant to the manuscript? The manuscript provides a very broad discussion about the general impact of volcanic eruptions on climate and atmospheric chemistry, but it does not discuss the problems and current limitations when modelling volcanic eruptions (e.g., spatial & temporal resolution, model agreement/disagreement of past model studies among each other and with observations, sectional versus modal microphysics modules, previous studies which addressed the vertical/horizontal distribution of volcanic emission plumes). Since you submit to "Geoscientific model development" I think a more technical motivation/introduction would be appropriate. I think the introduction should clearly motivate the research so that it is clear to a reader why the submodule was developed and why there is need for the research provided. In my point of view this is not the case.

We thank the reviewer for this suggestion. Indeed, we talked too detailed about ash and ozone chemistry. We shortened this considerably. However, we wanted to include the discussion on the importance of ash, as EVER was explicitly designed, such that volcanic ash can be simulated.

We chose to focus more on the general volcanic impacts to motivate why this submodel is needed within MESSy, and not necessarily why it is needed in addition to other available

volcanic emission routines in various models. However, we agree with the reviewer, that a more technical focus would improve the introduction. Thus, we discussed limitations in modelling volcanic eruptions more detailed in the introduction of the revised version. We did motivate the research in the original version by showing, why it is crucial to have this submodel in MESSy, but we now made it clearer, why we did not merely take an existing routine from other models. The technical extension to the introduction can be found as a reply to the comment on the additional literature above.

**Your conclusions just summarize what you have done in your work, including a brief recapitulation of some qualitative results with a brief outlook for further potential applications of the sub-model in the end. However, I think the conclusion section should summarize the key findings of the paper (also some quantitative statements, not only qualitative) and put them into a broader context. The conclusions should demonstrate the importance of the paper and convey the larger implications of the paper to the research field. Additionally, I also find it important to address the limitations of the research/model provided (or more specifically: of the presented sub-module) and highlight aspects identified in your work which require further research and development. I don't think these points are provided in the current version of the conclusion section of the manuscript.**

Yes, we mostly summarized our work, but in addition, we also summarized our key findings in the submitted manuscript, e. g. "Horizontal position and emission timing were found to have a minor impact on the long-term $SO_2$ burden in the stratosphere. Nevertheless, these parameters play a crucial role in detailed process studies during the initial weeks after an eruption. Overall, we conclude that simulations of volcanic eruptions can be effectively performed with the help of 3D- and column emissions. The optimal approach depends on the specific use case, with column emissions excelling in the short-term, and similar performance in the long-term" and " [...] , and we advocate its inclusion in simulations using the MESSy framework focusing on the upper troposphere and stratosphere. For very strong eruptions, it may be beneficial to distribute the emissions over multiple horizontal grid boxes and an extended time period, or adjust the vertical plume width, if discrepancies with observations occur".

We agree, that we can put them into some broader context with previous findings and point to some limitations and shortcomings, and did so in detail in the revised Discussion section. Additionally, we outlined further necessary research and development in the revised version. However, a technical paper describing a new submodel, evaluating it, and developing a default setup does not necessarily need "larger implications of the paper to the research field" but rather vital information for the people using the submodel. Larger scientific implications to the research field can result from future studies using the submodel. However, as we indeed produce findings, that are important for the research field, we discuss them in more detail in the conclusion and abstract in the revised version (see also abstract below).

**Similarly, the abstract just summarizes what you did in this study, but you do not provide any results or conclusions or broader implications to the research fields in the abstract, which I think is an essential part of the abstract.**

As outlined above, this is a technical paper describing a new submodel, evaluating it, and developing a default setup. The conclusion basically is, that it was developed, it works and it can reproduce observations. We added additional findings and implications (see comment

above) in the revised abstract:

" This work documents the operation of a new submodel for tracer emissions from Explosive Volcanic ERuptions (EVER v1.1), developed within the Modular Earth Submodel System (MESSy, version 2.55.1). EVER calculates additional tendencies of gaseous and aerosol tracers based on emission source parameters, aligned to specific sequences of volcanic eruptions or other atmospheric emission sources, allowing for the employment of various vertical emission profiles. We show that volcanic $SO_2$ plumes can be reasonably reproduced through EVER emissions in numerical simulations with the ECHAM/MESSy Atmospheric Chemistry Model (EMAC), using satellite observations of $SO_2$ columns and mixing ratios following the explosive eruption of the Nabro volcano (Eritrea) in 2011 and a degassing event of the Kilauea volcano (2018) in Kilauea. Previous volcanic studies showed large variability in stratospheric $SO_2$ burdens depending on the chosen volcanic emission databases and parameters. Sensitivity studies on $SO_2$ emissions from the Nabro volcano explore perturbations of the emission source parameters, revealing that emission altitude and the emitted mass above the tropopause are most important for the mid- to long-term evolution of stratospheric $SO_2$ plumes, while the correct timing and geographical location of the stratospheric entrance is crucial for the short-term plume evolution. We integrate information from a volcanic $SO_2$ emission inventory, additional satellite observations, and our findings from the sensitivity studies to establish a historical standard setup for volcanic eruptions impacting stratospheric $SO_2$ from 1990 to 2023, successfully evaluated with satellite observations of stratospheric $SO_2$ burden and aerosol optical properties. We advocate for this to be a standardized setup in all simulations within the MESSy framework concentrating on the upper troposphere and stratosphere in this period. Further potential applications of EVER involve studies on volcanic ash, wildfires, solar radiation modification, and atmospheric transport processes. "

An important aspect which is missing in the analysis and discussion of your results is the role of chemical loss of $SO_2$. In addition to dilution and transport, a (maybe even more) important aspect which determines the $SO_2$ lifetime is chemical $SO_2$ loss (i.e., mainly oxidation with OH and $O_3$). How is this represented in MECCA, and how does this influence your modelled results. You also write that for the Kilauea study, you applied "simplified chemistry". What does this mean? Since chemical loss is very important for the $SO_2$ lifetime you need to discuss how this affects the results in your study. How is the reaction with $O_3$ and OH represented in MECCA? Were the $O_3$ and OH fields also nudged to observations like written in the model description (probably not)? How does this influence the modelled results?

We agree with the reviewer that the chemical loss of $SO_2$ is important, and actually is the reason for sulfur aerosols to form. In the supplement, we provide the full chemical mechanism for both simulations, the stratospheric one and the "simplified chemistry" for the Kilauea simulation. We have also shortly added these information in the revised manuscript.

For the stratospheric setup:

"We employed the Mainz Isoprene Mechanism MIM1; Jöckel et al., 2006; Pöschl et al., 2000 within MECCA (see supplement for detailed mechanism), considering oxidation of $SO_2$ with OH to form $SO_3$, subsequently reacting with $H_2O$ to form $H_2SO_4$."

For the Kilauea setup:

"Oxidation of $SO_2$ to $H_2SO_4$ is directly realized via reaction with OH, without producing any intermediates. We do not consider DMS and OCS here (see supplement), leading to a potential underestimation of background maritime $SO_2$ concentrations."

While we treat $SO_2$ oxidation with OH in both setups as described in the supplement, we do not consider its oxidation with $O_3$. This reaction only takes place in the aqueous phase and is not important for the stratosphere (e. g. Kremser et al. (2016)). $O_3$ and OH are part of the chemical mechanism (see supplement) and are not nudged towards observations. The described nudging affects only the model dynamics using meteorological reanalysis data, which is more clearly described in the revised version (see also specific comment on nudging below).

Another aspect which is not discussed is the importance of aerosol microphysics. You show an in-depth analysis of the SO2 plume evolution and then in chapter 4 you suddenly come up with AOD analysis. However, there is an important step missing in between: Aerosol Microphysics. You write that you are using GMXE as an aerosol microphysics module and represent the aerosols using a modal approach. However, you do not provide any information about resulting aerosol burden, aerosol size distribution ect. You only show AOD and extinction (without indicating the wavelengths under consideration). Thus, the whole aerosol microphysics (which is a very important aspects when simulating volcanic emission plumes) is treated as a black box in this manuscript. In my point of view, it is crucial to also show resulting aerosol burden and compare them with observations as well as which other models (e.g. see Brodowsky et al. 2021).

We agree with the reviewer, that aerosol microphysics is a very important aspect for the modelling of volcanic aerosol, and we provide more detail in the revised version (e. g. the microphysical timestep of 8 minutes). Thank you for pointing this out. However, we want to make clear again, that this manuscript is not aiming to evaluate the aerosol microphysics or study the sensitivity of the results to the aerosol microphysics. Instead, we focused on the initial $SO_2$ plume, and only used the comparison to OSIRIS for an additional validation, and to compare our results to Schallock et al. (2023).

We thank the reviewer for pointing to Brodowsky et al. (2021), and we apologize for not including it in the initial submission. However, the only decadal satellite observations of (sulfur) aerosol burden available we could find is the MIPAS dataset from Günther et al. (2018), that was also used by Brodowsky et al. (2021). We considered using this dataset. However, a number of assumptions have to be made for a model-observation comparison, including the density of the $H_2SO_4$-$H_2O$ system, which fraction of the sulfur aerosol is liquid or solid (not directly distinguishable in our model), and how much of the aerosol water to include. The uncertainties in these assumptions lead to large variations in our model. Brodowsky et al. (2021) simplified this by normalizing the comparison to background levels, or adding a baseline correction to MIPAS, not showing the absolute differences. However, we believe that this would not add much benefit to our work, as the results would look very good, but only due to the normalization. However, if it is necessary, we could add this comparison to the supplement.

Moreover, in our opinion our study has a different focus, as we want to evaluate the EVER submodel. Thus, we look into the uncertainties in the $SO_2$ emission parameters, while not considering uncertainties in aerosol microphysics in great detail. Our manuscript is a "Development and technical paper" in Geoscientific Model Development, while the Brodowsky et al. (2021)

paper is a scientific study in JGR Atmopheres focusing on the evolution of sulfate aerosol, and thus focuses more on the production of $H_2SO_4$ and the resulting aerosol burden.

We apologize for not mentioning the wavelengths under consideration, and we have added this information in the revised version. The extinction and AOD are considered at a wavelength of 750 nm. Additionally, we will compare our results to previous model results. We decided to focus on comparison to observations of $SO_2$, as this is mostly influenced by the EVER submodel, and we wanted to show that we can reproduce observed $SO_2$ mixing ratios. As pointed out, the resulting aerosol burden depends on a lot of different factors, the evaluation of which goes beyond the scope of this manuscript. We clarify in the revised version, that the comparison of simulated AOD to observations is only a validation of the changes compared to the previous study by Schallock et al. (2023), and not a separate study on aerosol formation.

**Another aspect which comes too short in the discussion section is the influence of the spatial and temporal resolution of different processes. It is well known that these aspects are very important for realistic representation of volcanic plumes. While for the troposphere the horizontal resolution is more important in the stratosphere the vertical resolution is more important. For aerosol microphysics the microphysical timestep should be set small enough to realistically simulate nucleation and condensation. You could mention these aspects in the discussion of your results.**

Yes, the spatial resolution is very important for the modelling of volcanic eruptions. For that reason, we chose the maximum available horizontal resolution for the tropospheric study of Kilauea (T255), and the maximum available vertical resolution (90 levels) for the stratospheric study, similarly to the work of Brodowsky et al. (2021). The evaluation of different resolutions again goes beyond the scope of the paper.

We do not distinguish the chemical and microphysical timestep in our simulation, as the general model timestep length is only 8 minutes (we have added this information to the revised manuscript), and chemistry and microphysics are calculated in each timestep. This is different to the recent publication by the reviewer (Vattioni et al., 2024) where chemical and microphysical timestep differ. Vattioni et al. (2024) showed that in volcanic simulations, first considering condensation and nucleation results in the smallest numerical error. This is the default in GMXe, and thus we did not discuss it. Further reducing the timestep will of course improve the results, but is not computationally feasible. We further argue, that the chemical timestep length of 2 hours in the work from Vattioni et al. (2024) will lead to different errors in microphysics compared to the 8 minute chemical timestep length in our EMAC simulations. Moreover, the main focus of the manuscript is on the initial $SO_2$ plume, that is not influenced by the microphysical settings. For that reason, we do not want to engage in this discussion.

**I also find the manuscript too long. Many aspects which are discussed in the introduction and submodel description are not relevant to the storyline or are not picked up again in the discussion. I suggest shortening substantially and putting part of the text (e.g. description of code and namelists) into the supplement. Also, the structure could be improved. For example, the model description and setup and description of the observations (sect. 3.1 and sect. 3.2) could be a chapter for its own or part of chapter 2. Because the setup described there is also used in chapter 4. The different events simulated here (i.e., Nabro, Kilauea and the**

2007-2011 period) seem a little disconnected to each other. Why don't you show a full analysis of only one event (e.g. Nabro), but in more detail including sulfuric acid aerosol burden etc.

We thank the reviewer for the very helpful suggestions, that we applied to our manuscript. First, we considerably shortened the manuscript. The main affected Sections are described below:

- The description of general impacts of volcanic eruptions were shortened in the introduction.

- We removed the namelist example for the new submodel (Fig. 1) and some of the detailed explanations, and moved it to the supplement.

- We shortened the introduction of the Nabro and Kilauea volcano.

- We restructured the methods, and thereby avoided repetitions.

In addition, we generally restructured the methods and results section. Sect. 2, including the presentation of the EVER submodel mostly remained in the same structure, only adding the description of the default namelist setup as an additional subsection. Sect. 3 now only focuses on the methods and the experimental setup, while Sect. 4 only shows the results of the three experiments. This results in the following new structure for Sect. 2-4:

2. New MESSy submodel for Explosive Volcanic ERuptions (EVER)

   2.1. Submodel description

      2.1.1. Primary aerosol emissions

      2.1.2. Vertical distributions

   2.2. Historic default namelist setup

3. Methods

   3.1. Observations

   3.2. Model setup and numerical experiments

      3.2.1. $SO_2$ in explosive volcanic eruptions — Nabro (2011)

      3.2.2. Multi-year simulation with the historic namelist setup

      3.2.3. $SO_2$ from degassing volcanoes — Kilauea (2018)

4. Results

   4.1. $SO_2$ in explosive volcanic eruptions — Nabro (2011)

   4.2. Evaluation of the historic default namelist setup in a multi-year simulation from 2008 to 2011 (stratospheric $SO_2$ burden and sAOD)

   4.3. $SO_2$ from degassing volcanoes — Kilauea (2018)

However, we wanted to keep the submodel description in a separate section, as this is a very important part of the manuscript. Moreover, we do not believe that the submodel description

in a technical paper has to necessarily follow the storyline of the evaluation, but should instead focus on completeness.

We understand, that the different events seem to be disconnected. We motivated the methodology in more detail in the experimental setup section in the revised version. We chose the different events for the following different reason:

- Single analysis of the Nabro volcano to evaluate the submodel in the stratosphere, and perform sensitivity studies on the emission parameters for stratospheric volcanoes.

- Kilauea study to show the applicability of the submodel for degassing volcanoes

- 2008-2011 period to evaluate the historic namelist setup developed

For that reason, we want to keep the different evaluations, with an improved structure, in the revised version.

**Detailed comments**

Line1: What is a methodological study? Isn't every scientific study methodological?

We agree and removed the word "methodological".

Line 14/15: Suggesting to change "solar geoengineering" to "solar radiation modification", a more appropriate term.

We replaced "solar geoengineering" by "solar radiation modification" in the abstract, and all later occurrences.

General Comment on the abstract: The abstract mostly reflects what was done in this study, but there is no mentioning on results and conclusions, which I think is a key component of an abstract. Thus, I suggest adding some quantitative results and conclusions/broader impacts.

We slightly extended the abstract, including the findings of our study that are important for the community. The full abstract of the revised version can be found above.

Line 19/20: "On the on hand... on the other hand" is normally used for opposing arguments. The ones mentioned here are more additive. I suggest reformulating. Whether it is "substantial" or not, depends on the magnitude.

We reformulated that, thank you for the remark: "The resulting additional stratospheric aerosol loading may exert a negative radiative forcing (Schallock et al., 2023; Schmidt et al., 2018), and serves as surfaces for heterogeneous reactions, thus, impacting stratospheric composition in general and ozone in particular (Klobas et al., 2017; Tie and Brasseur, 1995)."

Line 25 and lines 32-34: Do you have references for that?

The source for line 25 was also Textor et al. (2004) as the following sentence. However, we removed the sentence anyways, as it is was not important for the introduction, and we aim to shorten it, following the general comments above.

**Line 25/26: Do you have a reference for this?: "The composition of volcanic plumes exhibits considerable variability and depends on the intricate mixture of chemical species in the magma"**

The source was also Textor et al. (2004) as the following sentence. However, we removed the sentence anyways, as it is was not important for the introduction, and we aim to shorten it.

**Line 27-31: I suggest combining the two sentences (i.e., list the example of Hunga Tonga in the first sentence).**

Thank you for the suggestion. We reformulated it accordingly.

**Line 32-34: a definition of "long-term" would be good. sulfuric acid aerosols and their precursors are usually removed from the stratosphere within 2 years. I don't think this is long-term in terms of climate. Maybe chlorine species would have a long term effect.**

We removed "long-term".

**Line 34-36: This is somehow confusing. In the first paragraph you speak of "the most explosive volcanic eruptions" and now you write of emissions per year. Do you still speak of large explosive volcanic eruptions or do these numbers also account for degassing non-volcanic eruptions? I suggest being more precise here to what exactly these numbers refer to.**

Thank you for the remark. We removed this part as it was not important for the introduction.

**Line 37/38: I suggest changing the term "sulphate" with "sulfuric acid", since technically speaking, a sulphate is a solid (e.g. CaSO4). Also add references to this statement.**

We used the term sulfate, as it is common practice in the literature, but we agree with the reviewer, that it is imprecise. However, we prefer to use the term "sulfur aerosol" over "sulfuric acid aerosol", as it also includes (bi-)sulfates, such as ammonium (bi-)sulfate, that can form in the presence of ammonia. We adjusted it everywhere. We also added a reference (Kremser et al., 2016) to the mentioned statement.

**Line 36: Maybe add "under volcanic conditions", otherwise other sulfuric acid precursor gases are also important. I know $SO_2$ is poisonous, but isn't the effect on acid rain mainly a result of uptake of sulfuric acid aerosols (and not primarily $SO_2$)?**

We removed this passage in the process of shortening the introduction. We agree with the reviewer, that acid rain is a result of uptake of sulfur aerosol, and not $SO_2$. We implied that "$SO_2$ plays a pivotal role in air pollution and the formation of acid rain" via the formation of sulfur aerosol and subsequent uptake, but did not explicitly write it.

**Line 46: "up to" or "over"? But not both.**

Removed "up to", and reformulated to "The lifetime of stratospheric aerosols can exceed 2 years when injected in the tropics".

**Line 81:** What is a "horizontal gid box"? What is the difference between a horizontal or vertical grid box? Do you mean "vertical column of gird boxes"?

Yes, that was the intended meaning. This part was majorly reworked, when giving a more technical introduction, and this formulation does not appear any more in the revised version (see also the replies to the general comments on the more technical introduction).

**Line 85:** Are you only aiming at providing or are you providing? I suggest reformulating to "We provide ...", or reformulate in another way if you don't. Same for the last sentence of the paragraph: "This was achieved through the following three steps of work: ". Don't undersell your research.

Thank you for the suggestion, we majorly reformulated the respective paragraphs including the suggestions.

**Line 90:** "... of vertical emission distributions ... "

As we specified, that we talk about emissions in the sentence already, we thought it was not necessary here. However, we added "emission" to make it clear.

**Line 116:** You only use GMXe aerosol microphysics module in this study. Why do you introduce the other two submodules too? This only causes confusion.

The new submodel EVER was developed, such that it can be coupled to the different aerosol submodels in the MESSy environment, and the technical functionality was tested for all three submodels. We wanted to make that clear at that point, as this is an essential part of the submodel. We added an explanation, that primary emissions were not explicitly evaluated, and only GMXe was used in all studies to process the resulting sulfur aerosol. Additionally, we moved this paragraph to the Subsection on "Primary aerosol emissions"

**Line 127:** What do you mean by "linear columns"?

We reformulated to "... capable of emitting point sources and uniformly distributed columns of trace gases". We replaced all occurrences of "linear columns".

**Section 2.1:** I think the description of the new submodel is too technical. It is probably not useful for most readers of this study. I suggest making the description of the new sub model more general and provide a more technical description (e.g. how the name list works, what the different name list parameters are) in the supplement.

We thank the reviewer for the suggestion. We moved the namelist to the supplement, however kept the explanation for the different parameters, that need to be provided, as we believe this is essential for a documentation of a new submodel.

**Section 2.2.1:** The title of this section is "Primary emissions", but the subchapter is specifically on direct aerosol emissions. Maybe specify this in the title. However, why is this subchapter important? In this manuscript, only $SO_2$ injections are evaluated. Maybe you can skip this subchapter or put it into the supplement.

We changed the title of the section to "Primary aerosol emissions". This work documents the new submodel, which is capable of directly emitting aerosols. We only performed the sensitivity studies and evaluation for $SO_2$. However, we believe that it is important to keep this paragraph

for completeness, as the capability for primary aerosol emissions is essential for the submodel, and its documentation for the people using it.

**Section 2.2.2: This could be picked up later in the paper.**

We believe that the reviewer refers to Section 2.1.2, as there is no Section 2.2.2. We pick this up later in the experimental setup (Gaussian distribution) and in the discussion (distribution of emissions on a larger area).

**Section 3.1 There is no mentioning of the microphysical, chemical and dynamical time steps applied in the models used in this work. A recent study has highlighted the need for appropriately setting the microphysical time step when simulating volcanic eruptions.**

We thank the reviewer for this comment, and provided the model timestep (8 minutes) in the revised version (identical for microphysics, chemistry and dynamics), and also refer to Vattioni et al. (2024) in the introduction.

**Section 3.2: Maybe this section can be shortened. Are such detailed descriptions of all the different satellite and technical details such as their resolutions required?**

We thank the reviewer for the suggestion. We considerably shortened this paragraph, by providing a table on the used satellites, and refer to the references for more detail. In the new Section on the Experimental Setup (3.2), we give more information on which satellite observations we use for which evaluation. The new table is provided in the reply to Reviewer 2.

**Section 3/4: I assume the model description provided in section 3.1 also applies to section 4, right? And some of the described observations (satellites) in 3.2 are only used in section 4, right? To avoid confusion, I suggest separating the model description and observations from chapter 3 and create an own chapter for this. Or maybe something similar, just improve the structure of the paper, it is confusing sometimes.**

We agree with the referee, that the chosen structure can be confusing, and are thankful for the suggestion. Indeed, the model description and observations of Sections 3.1 and 3.2 also applied for Section 4. We rearranged the structure, providing a separate Section for the methods, where we introduce the satellites, the model setup and outline the experimental setup. We also made it clear, which model setup and which observations contributed to which experiment. See also the reply to the last general comment for more details on the applied major changes to the manuscript structure.

**Line 203-207: Why is this information important at all? You are not looking at aerosols in this chapter, but only at $SO_2$ plumes. Is $SO_2$ also treated within GMXe?**

We agree, that GMXe is not important for Section 3, as $SO_2$ is not treated with GMXe. We wanted to give an overview on the general model setup, that was used throughout the manuscript, but understand that the structure was confusing and adjusted it (see previous comment). The microphysics are now only introduced for the multi-year simulation.

**Line 200: Nudged to which variables? Wind and temperature?**

We reformulated to:

"Temperature, the logarithm of the surface pressure, divergence and vorticity are "nudged" (more details in Jöckel et al., 2006) towards meteorological reanalysis data (ERA5, Hersbach et al., 2020) from the European Centre for Medium-Range weather forecasts (ECMWF). In addition, we employ the QBO submodel to weakly nudge the simulations to QBO zonal wind observations between 10 and 90 hPa (Giorgetta et al., 2002) to avoid a phase drift."

**Line 222: Maybe just write: "Namelist setup, chemical mechanism and runscripts can be found in the supplement." However, I think this belongs in the data availability statement not in the main text.**

We rephrased it accordingly. However, we want to provide the information here, so the reader can refer to the supplement, in case of any question to the chemical mechanism.

**Line 216-219: Why is this information important? In this chapter (chapter 3) you are only focusing on $SO_2$ plumes, but no aerosol optical effects. I suggest skipping.**

We agree with the reviewer. As mentioned above, we rearranged the manuscript, and the aerosol optical properties are now part of the experimental setup section, and only introduced for the outline of the evaluation of the default namelist setup.

**Line 284/285: "Especially the second stratospheric plume on June 16 could comprise remnants of the tropospheric plume, that are uplifted" This sentence is confusing to me: What do you mean with tropospheric plume? The volcanic plume or the monsoon?**

In the preceding paragraph (Lines 276-278), we discuss the separation of the tropospheric and stratospheric plume: "Concurrently, a wind shear within the AMA around the tropical tropopause induced a separation between tropospheric and stratospheric $SO_2$, with the stratospheric component further north." So we speculate (or previous studies speculated) if the second stratospheric emission plume actually results from the uplift of the tropospheric plume from June 13. We clarify in the revised manuscript, that we refer to the previous plume.

**Line 323: There is no specification of the emission in Mills et al. 2016 so far. This pups up here a little abruptly, since this was not discussed in the introduction or anywhere prior to here.**

We agree, and we discuss different emission inventories in the introduction and Mills et al. (2016) in particular in the revised version (see above in the replies to the general comments). In addition, we set the different emission inventories into perspective, and clarify, why we decided for Schallock et al. (2023) in the Discussion section of the revised manuscript. Moreover, we give more information about the Nabro emissions from Mills et al. (2016) in the revised manuscript: "Emissions as described by Mills et al. (2016). The emissions are injected over several days and in columns covering the free troposphere and the UTLS region, ranging from 2.5 to 17 km. The only partly stratospheric emission takes place on June 13, emitting 1500 kt, uniformly distributed over the altitude range from 9.7 to 17 km."

**Line 334: "The column amount estimation assumes that all $SO_2$ of the plume is**

centered at the respective altitude depicted in Fig. 4 " To me it is not clear how this explanation should explain the discrepancies between observed and modelled column amounts. Can you explain further?

We agree, that this statement does not sufficiently explain the discrepancies, as it was not formulated precisely. In fact, the retrieval considers that the plume is centered at these altitudes above 14 km. However, it takes into account the whole column for this retrieval, also including the tropospheric part of the plume, thereby not only representing the stratospheric part. We added the following sentence after the cited sentence: "However, it takes into account the complete column, also including the tropospheric part of the plume."

Line 338: "From Fig. 4, it seems that the simulated columns slightly broaden over time compared to the observations, with the plume appearing to sink." I am confused here. Figure 4 shows altitudes not column $SO_2$. It also seems like the simulated plume (reference) is higher up compared to the observed plume. Do you mean the observed plume appears to sink? It is hard to see any broadening of the plume in Figure 4.

We want to shortly clarify the two statements in the following:

- "... it seems that the simulated columns slightly broaden over time compared to the observations ...": We wanted to explain here, why in the bottom row of Fig. 4 (June 17) the simulated altitude distribution is broader than in the observations. The reason is that we see maximum $SO_2$ mixing ratios in the model at these altitudes, however at the edges of the plume these values are very small, below the detection limit of IASI, which can be seen from Fig. 5 bottom row, that shows a much narrower plume.

- "... with the plume appearing to sink.": This refers again to the bottom column of Fig. 4 (June 17). At longitudes higher than 90° E the reference simulation shows lower altitudes compared to the observations. This is, what we wanted to comment on.

To make the first statement clearer, we reformulated as follows: "While the simulated altitude distribution broadens over time (Fig. 4, June 17 **reference**), the total column analysis (Fig. 5, June 17 **reference**) shows that the columns at the edges of the plume are very low, falling below the detection limit of IASI." We omitted the comment on the sinking to avoid confusion.

Line 339/340: The simulated (reference) $SO_2$ column distribution in Figure 5 is broader compared to the observation... not narrower. This is confusion.

See reply above.

Line 361/362: "However, IASI faces limitations in capturing the long-term evolution of volcanic plumes due to the dilution of the emitted $SO_2$, leading to column amounts that fall below the instrument's detection limit" Do you really know dilution is the main process that $SO_2$ concentrations fall below the detection limit? If yes, do you have references for this? Isn't chemical loss ($SO_2$ oxidation via OH and O3) equally or even more important on longer time scales? How is this represented in the model and how does this affect the long-term evolution of the $SO_2$ plume?

We moved this statement to the Methods section, and reformulated to. "This can likely be

attributed to the dilution and chemical removal of $SO_2$ in the plume." We talk about the removal in the Experimental setup in more detail (see general comments).

Lines 416-422: "The overall slightly faster decline observed in the simulation compared to the observations may be a consequence of the absence of primary particles, such as volcanic ash, in the simulations, resulting in a discrepancy between simulated and observed particle size distributions."
What processes should be the reasons for that? I guess you mean that ash could result in self-lofting of airmasses due to absorption of radiation and thus local heating? Or what other processes do you have in mind? It is important to name them since this is not clear from how it is written now.

"Alternatively, the simulated particle sizes may grow excessively large too quickly, leading to an overestimation of sedimentation efficiency"
You do not show any simulated particle sizes. You show and write about $SO_2$ plumes. $SO_2$ is a gas. Gases are mainly subject to diffusion & transport and in the case of $SO_2$ more importantly: chemical loss, ... but definitely not sedimentation. What about chemical loss? How does this affect the dissipation of the plume compared with observations?

"Whether this discrepancy arises from nucleation rates versus condensation efficiency, the overall representation of the size distribution with only four modes, or the limitation to one horizontal grid box will be the topic of upcoming studies."
$SO_2$ concentrations are definitely not affected by nucleation and condensation rates. Chemical loss of $SO_2$ is dominated by reaction with OH and O3. These reaction result in formation of SO3, which then together with H2O forms H2OS4 gas. H2SO4 gas has a very low vapor pressure and immediately forms sulfuric acid aerosols via condensation or nucleation. Have a look at Feinberg et al. 2019 and the stratospheric sulfur cycle presented in there. It should get obvious that nucleation and condensation rates as well as aerosol size distributions do not affect the chemical $SO_2$ lifetime/burden.

Feinberg, A., Sukhodolov, T., Luo, B.-P., Rozanov, E., Winkel, L. H. E., Peter, T., and Stenke, A.: Improved tropospheric and stratospheric sulfur cycle in the aerosol– chemistry–climate model SOCOL-AERv2, Geosci. Model Dev., 12, 3863–3887, https://doi.org/10.5194/gmd-12-3863-2019, 2019.

We agree, that the discussion in this paragraph was not correct, as we attributed the decline of the stratospheric $SO_2$ burden to aerosol effects, while they are actually not affected by these. We replaced the paragraph by the following in the revised version:

" The overall slightly faster decline of the stratospheric $SO_2$ burden in the simulation compared to the observations appears consistent across all simulations. It can either be attributed to an overestimation of the chemical removal, i. e. the oxidation with OH, or too efficient transport from the stratosphere to the troposphere, that also depends on the injection altitude. "

Line 441: What "data"? Simulated or observed?

[Figure]

Figure 1: Observed (top) and simulated SO$_2$ columns resulting from the degassing of the Kilauea volcano at selected days in June 2018 at a model resolution of T255. **a\*,** Observations from TROPOMI, regridded to T255 resolution. **b\*,** Simulation with emission rates derived by Jost (2021, scaled by a factor of 4.3 – see text for details). **c\*,** Simulation with optimized emission rates, based on a comparison between model and observations (refer to the text for more details). **d\*,** Relative change from the simulation with derived emission rates by Jost (2021) to the optimized simulation.

We believe that the reviewer refers to line 341. We refer to the simulation here, and state this in the revised version. We apologize for the incorrect formulation

Line 341/342: Ahaa... it only becomes clear that you were talking about the initial plume on June 14 until now. The statements you make in this paragraph are only true for the initial plume on June 14. You really need to be more precise here... The statements of this paragraph are not valid for June 17.

We actually refer to June 17, where the altitude distribution is broader in the simulation compared to observations. We reformulated as mentioned in the reply to comments above.

Figures: I suggest assigning letters a, b, c, d ... to the subpanels of figures to enable better referencing.

We thank the reviewer for the suggestions, and applied letters to all manuscript figures with more than one panel (see also updated figures in this reply).

Figure 444: It is hard to see any difference in agreement/disagreement with observation in Figure 8 of the June 7 and 10 data compared to for example June 5 and June 15. I suggest plotting the differences compared to the observations in the middle and lower panel. This would highlight the differences.

We believe that the reviewer referred to Figure 8. We adjusted the Figure and also added the relative change between the simulations (see Fig. 1d here).

**Line 448-451: Most of this can go into the figure caption.**

We thank the reviewer for the suggestion. We shortened it here, as the information was already present in the caption. The shortened paragraph is (note that the figure numbers changed as the manuscript was restructured):

" Figure 10 provides a quantitative assessment of the horizontal extent depicted in Figure 9 (spanning from $-168°$ to $-152°$ W, $15°$ to $25°$ N), showing spatially averaged $SO_2$ column amount within this horizontal window, $\overline{SO_{2(col,d)}}$ as observed and simulated at each day in June 2018 $d$ (bottom), and spatial correlation between simulated and observed logarithmic $SO_2$ column amounts within this window (top). "

**Line 556: I disagree with that. The observations and the model does not "exhibit similar patterns" between 0 and 25N. The QBO signal is much more pronounced in the model compared to the observations. And the observed extinction is very different compared to the modelled ones in absolute numbers.**

We thank the reviewer for this comment. Indeed, the QBO signal is more enhanced above 20 km, however we mostly focused on the lower stratosphere. Also, the absolute mixing ratios differ, but this is mostly in line with the finding from the sAOD study, or due to the larger vertical extent of the observed volcanic plumes, that are typically represented in AKMs, that were not available for the OSIRIS observations. We reformulated the paragraph to be more precise:

" Figure 8 displays the aerosol extinction at 750 nm, as observed from the OSIRIS instrument and simulated. The effect of the three major volcanic eruptions is mostly evident between 16 and 20 km in the tropics, and between 12 and 18 km at higher latitudes. Analogously to Fig. 7, the magnitude of extinction is differing between the observations and the two simulations. In addition, discrepancies are noticeable in the maximum altitude of the plume and below the tropopause. While the 3D simulation largely reproduces the observed maximum altitude, using satellite observations with, in case of MIPAS, the incorporated AKM as input, the EVER simulation may present more realistic maximum altitudes. The differences between the observations and the simulation below the tropopause are strongly driven by the coincidence with clouds which hinder the retrieval of aerosol extinction. The slight differences between the two simulations from 10 to 15 km in the tropics can most likely be attributed to the differences between the simulation setups. In addition, the QBO signal with differing aerosol concentrations above 20 km, depending on the QBO phase (e. g. Hommel et al., 2015), is more pronounced in the simulation, potentially subject to future studies. "

**Line 459: Why do you think the observations are wrong? It could well be your model which is wrong. Why don't you optimize your model to improve agreement with observations?**

We do not state, that the observations are wrong. We optimize the emission rates, such that the model can reproduce the observations. So we do not "optimize" the observations, but the input to our model.

**Line 460: With "implemented emission rates" you mean the observations, right?**

We mean the emission rates, that were derived by Jost (2021) based on observed $SO_2$ columns.

We do not argue, that these emission rates are wrong, but that they produce wrong $SO_2$ columns in the model. Thus, we want to optimize them, such that the model produces observed $SO_2$ columns.

**Line 461: I guess you considered the "emission rates" from the observations, right?**

We state in lines 433ff how the emission rates were calculated based on observations. These emission rates were used here.

**Line 465: "The coefficients $a_{d-i}$ and the background $SO_2$ column amount, $SO_{2(col,d)}$, represent the free parameters in the linear predictor and were determined through a least squares fit". To what is "least square fit" referring to? What is it fitted to? "Least square fit" to the observed total column $SO_2$? If yes, then it should be obvious that the simulations in the end agree with the observed total column $SO_2$.**

No, the fit is done as described in the text. $\overline{SO_{2(col,d)}}$ is the simulated spatially averaged $SO_2$ column amount within the depicted horizontal window. Thus, we get a relationship between emission rates of the preceding three days and the resulting simulated average column. Using this relationship, we then use an SGD algorithm to optimize the emission rates, so that we match the observed spatially averaged $SO_2$ column amount. So yes, it is more or less obvious, that we match the observations in the end. However, that was the goal to show a method, how observations can be reproduced with the model.

**Line 466: What is a "stochastic gradient descent"? It would be helpful to describe this in one sentence, other wise it is just a black box to most readers. The code provided below does not help, since this is rather technical. This can go to a supplement.**

We explain it in more detail and give a reference in the Methods section:

" In addition, we perform a simulation with tuned emission rates. For that purpose, we first fit the emission rates from Jost (2021) to the resulting simulated $SO_2$ columns. Subsequently, we optimize the emission rates gradually by using a stochastic gradient descent algorithm (e. g. Ruder, 2016) to match the observed columns. More details on the optimization can be found in the supplement. "

The algorithm was moved to the supplement.

**Line 470: Why would you expect increases in spatial correlation if you only improve the emission rates?**

We wanted to test, if we see improvements in the spatial correlation, as the "correct" representation of the temporal emission over several days may result in the right spatial pattern. However, this was not the case, as the observed $SO_2$ columns were mostly driven by the emissions of the previous 24 hours.

**Line 473: What do you mean with "effectively". I disagree with this statement. You only get good agreement in total column $SO_2$ when tuning the emissions in your model to fit the observational data. I think most models get better agreement with observations when tuning their emissions.**

We only stated, that the model can "effectively" simulate this. We remove the word "effectively". However, we do not claim to do it better than other (regional) models (with higher resolution).

**Line 474-478: You did not investigate any sensitivity to spatial resolution. Thus, you cannot make this conclusions. Delete this part, or show evidence for this conclusions.**

We did this sensitivity study by performing the simulation in different horizontal resolutions, but did not show it here to avoid extending the manuscript even further. We slightly reformulated the sentence to make it clearer, that we performed these studies:

" Simulations with equal emissions performed at more standard horizontal resolutions, such as T63 and T106, failed to reproduce the observations adequately (not shown), whereas these resolutions are mostly sufficient for stratospheric simulations. "

**Line 477/478: This is the most critical result which I think you must discuss more. You only get good agreement with total column $SO_2$ observations if you tune the emission rates according to your simulation results. I know that this is a common problem for models simulating volcanic eruptions (e.g. Mt. Pinatubo), but you should highlight this. critically discuss it and derive the right conclusions. I also think "analysis" is not the right word here. More precise would be "tuning".**

We moved this part to the discussion, used the word "tuning", and clarified that we explicitly use the observations for that. However, we do not see a problem with this approach. We even make clear in Line 636-639, that this could be a process how to incorporate $SO_2$ from degassing volcanoes into the model, if observed values should be reproduced. We do not state, that the optimized emission rates are determined independent of observations or modelling.

**Line 520/521: The magnitude of the signal would not change if the satellite signals were only delayed compared to observations. Isn't it mainly the sensitivity of the measured satellite signal? Please be more precise here.**

We wanted to state here, that the overestimation of the model can be attributed to the limited reliability of MIPAS observations directly after the eruption. We agree, that the reason for that is the sensitivity of the measured satellite signal. We slightly reformulated this to: "This discrepancy can be attributed to the limited reliability of MIPAS observations shortly after strong eruptions (see Sect. ..."

**Line 522-529: Here again: What is the impact of chemical loss of $SO_2$? Also did you compare your model to background sulfur cycle (see Brodovsky et al. 2024)? It might make sense to fist perform same simulation as in this study to compare to other models and observations.**

**Brodowsky, C. V., Sukhodolov, T., Chiodo, G., Aquila, V., Bekki, S., Dhomse, S. S., Höpfner, M., Laakso, A., Mann, G. W., Niemeier, U., Pitari, G., Quaglia, I., Rozanov, E., Schmidt, A., Sekiya, T., Tilmes, S., Timmreck, C., Vattioni, S., Visioni, D., Yu, P., Zhu, Y., and Peter, T.: Analysis of the global atmospheric background sulfur budget in a multi- model framework, Atmos. Chem. Phys., 24, 5513–5548, https://doi.org/10.5194/acp-24- 5513-2024, 2024.**

The model was evaluated for background sulfur before in Brühl et al. (2012). The suggested

comparison would be a new study, that could be performed in the future. We additionally now show, that this underestimation could additionally depend on the lower integration limit for the derivation of the stratospheric $SO_2$ burden (see Fig. 2), and is mostly related to smaller volcanic eruptions, where we did not optimize the stratospheric entry point from IASI, as observations were not available. We adjusted this paragraph in the revised manuscript.

Line 530: Here you suddenly start talking and comparing AOD resulting from these volcanic eruptions. So far you talked and compared $SO_2$ plumes. It would be great to first see some sulfuric acid aerosol size distribution or how the sulfuric acid aerosol plume/burden evolves in the aftermath of these volcanic eruptions (see Brodowsky 2021). This is what defines the AOD downstream not the $SO_2$ plume. There is an important part missing here when going from $SO_2$ plumes to AOD. Without this intermediate step it is hard to say where the discrepancies between model and observations are coming from. It is just guessing since aerosol formation and distribution in the model appear like a black box to the reader... Also, crucial information is missing about the wavelengths to which the AOD and extinctions shown in Figure 11 and 12 are referring to. Why did you only look at 0° to 25°N and 45°-80°N? and not other regions?

We agree that it would be interesting to study the size distributions or the aerosol plume evolution. However, this was not the focus of this manuscript. We still mostly focus on $SO_2$ plume evolution. We just additionally added the aerosol effects for a short validation and comparison to Schallock et al. (2023). We agree that it is hard to say where the discrepancies are coming from. However, we believe that a more detailed study on this would be outside of the scope of the manuscript. We clarified in the manuscript, why we only show AOD and not all aerosol properties, and provide reasons for not showing comparisons to observations of aerosol burdens in the replies to the general comments. Additionally, we clarify our approach in the experimental setup section revised manuscript.

We apologize for not providing the wavelengths for extinction and AOD, but will add it in the revised version (750 nm). We only looked into the mentioned regions, as (1) all major volcanic eruptions happened in the Northern Hemisphere, and (2) the tropopause has a steep altitude gradient in the region between 25° and 45° making a comparison of sAOD to observations more challenging and less reliable, as it is difficult to diagnose the troposphere in the observations.

Line 537-544: This paragraph needs references and is somehow handwaving.

See comments below.

Line 539/541: I think this reads a little hand wavy here. Please be more specific. A paper which addresses some potential effects is Vattioni et al. 2024. Vattioni, S., Stenke, A., Luo, B., Chiodo, G., Sukhodolov, T., Wunderlin, E., and Peter, T.: Importance of microphysical settings for climate forcing by stratospheric $SO_2$ injections as modeled by SOCOL-AERv2, Geosci. Model Dev., 17, 4181–4197, https://doi.org/10.5194/gmd-17-4181-2024, 2024.

We agree, that this speculation is not well justified. As the reviewer correctly pointed out, that we cannot really make assumptions about the microphysics, because we do not explicitly

investigate the microphysics, we removed this part, and replaced it (see above).

Line 542-544: "This phenomenon could potentially be addressed by distributing emissions across multiple horizontal grid boxes and releasing the $SO_2$ over an extended time period." Why would you do this? In section 3 you showed good spatial agreement with observations, so why change the spatial distribution? What I think could help might be changing the horizontal resolution. It seems you again are looking for the error in the emission scheme/observations instead of in within the model. Your suggestion would reduce the $SO_2$ concentrations and thus the H2SO4 concentrations downstream. This reduces condensation and especially aerosol nucleation rates. But why should this be justified?

See above.

Line 546/547: "This anomaly could be attributed to an overestimation of transport from higher latitudes to the tropical stratosphere, or a general overestimation of the emissions." Why should this be the case? Isn't the transport in this region going exactly into the other direction (from the tropics to higher latitudes)? And there is also a tropical "transport barrier".

Indeed, the transport in the stratosphere follows the Brewer-Dobson circulation and transports air from lower to higher latitudes. However, in the two cases of Sarychev and Kasatochi, the injections happened at higher latitudes (47°N and 50°N, respectively), yet they had an effect on the tropical stratosphere (also in the observations). Thus, there has to be some transport from the higher to the lower latitudes. As both volcanoes erupted in the Asian monsoon period, a possible transport pathway would be the monsoon anticyclone.

Line 549: What differences are you talking about? Difference compared to what?

We mean the discrepancies between the two simulations, and clarified it in the revised version: "The discrepancies between the two simulations in the mid-term transport after ..."

Line 552/553: "The interaction with the South Asian monsoon anticyclone potentially causes differing transport to lower or higher latitudes, respectively." Weren't the simulations nudged towards observed wind fields?

The simulations were nudged to reanalysis data (see above). However, the two simulations differ in altitude, geographical location and timing of the emission. Thus, they are differently affected by the monsoon anticyclone. We extended the statement as follows: "The interaction with the South Asian monsoon anticyclone potentially causes differing transport to lower or higher latitudes, respectively, if altitude, timing and geographical location of the emission do not coincide."

Line 554/555: This sensitivity needs to be addressed by showing some plots in the supplement with different cutoff altitudes, since this defines whether the model agrees with observations or not. . .

We agree that it is important to show the sensitivity to the cutoff altitude. As the term "stratospheric cutoff altitude" might be misleading, we generally replaced it with "lower integration limit" and made it clear everywhere. Note that this "lower integration limit" refers to the derivation of the stratospheric $SO_2$ burden and the sAOD, and not to the injection range of

[Figure]

Figure 2: Sensitivity of the evaluation results to the lower integration limit (applied to observations and simulations) for the derivation of the stratospheric $SO_2$ burden. **a,** Lower integration limit as applied in the manuscript (16 km from 0-30°N, 14 km from 30-60°N and 12 km from 60-90°N. Variations of plus 1 km (**b**) and minus 1 km (**c**) show similar agreement at different absolute values. Slight differences can be seen at background levels with increased agreement in 2010 for the increased lower integration limit (**b**). *This figure is added to the supplement of the revised version.*

the emission. We also clarify this in the revised version in the Methods section. We now provide a study in the supplement comparing the results for stratospheric $SO_2$ burden and sAOD, varying the lower integration limit by ± 1 km. We provide the figures here in the responses as Fig. 2 and 3, and additionally add them to the supplement. In the cited paragraph of the submitted manuscript, we state "... that total sAOD is highly sensitive to the altitude chosen as a lower cutoff". However, that does not mean that the agreement with observations is highly sensitive to the lower integration limit. We now clarify this in the manuscript, and refer to the supplement figure.

Line 556-561: I would make it clear in this paragraph (and for the whole discussion of AOD from Line 530 onward) that here you are talking about the sulfuric acid aerosol plume and the AOD resulting from these aerosols, whereas so far in the paper you talked about the $SO_2$ plume. The two $SO_2$ and sulfuric acid aerosol plumes likely look different.

We thank the reviewer for the suggestion, and made the transition from $SO_2$ to AOD and extinction clearer in the revised version, separating it into two different subsubsections, and defining the difference in the Methods.

Line 565/566: The first sentence of the discussion is not true. You do not show anything related to "aerosol formation". You only show comparison with AOD observations, but this does not tell you anything about aerosol formation processes.

The reviewer is correct, that we do not show aerosol formation here, and we removed it from the statement: "We showed that $SO_2$ emissions from explosive volcanic eruptions and the subsequent plume evolution can be reasonably reproduced in EMAC ..."

Line 575: You do not show "aerosol burden" here. Thus, you can not make any conclusions about this. Do you mean "forecasted" instead of "examined"? They can be examined, but just not immediately.

We reformulated to: "... and results in differing stratospheric lifetimes for the volcanic plume".

[Figure]

Figure 3: Sensitivity of the evaluation result to the lower integration limit (applied to observations and simulations) for the derivation of sAOD. **a,d** Lower integration limit as applied in the manuscript (16 km from 0-25°N and 12 km from 45-80°N). Variations of plus 1 km (**b,e**) and minus 1 km (**c,f**) show similar agreement at different absolute values. The most significant differences are observed for the Sarychev volcano, that shows strongly decreased simulated sAOD at increased lower integration limit at higher latitudes (**e**). *This figure is added to the supplement of the revised version.*

Regarding the second question: We refer to the 3D simulation here, that only injects 3D perturbations of stratospheric SO$_2$ a couple of days after the initial eruption. Thus, the short-term evolution cannot be examined. We clarified this in the revised version.

Line 589-593: You did not analyze how to "adequately simulate stratospheric aerosol burden". You cannot make any conclusions about stratospheric aerosol burden, if you do not show aerosol burden in the manuscript. The first sentence is confusing. What do you mean with "differences" in the first sentence of this paragraph? A difference compared to what? Again, what is the importance of chemical loss of SO$_2$ in the whole analysis? This could also be discussed here. You cannot make conclusions about the "sulfate" lifetime with the analysis shown in your manuscript.

We removed the term "aerosol burden" in the mentioned paragraph. We meant that the importance of each parameter is differing. We reformulated the sentence to: "The sensitivity studies revealed that the importance of accurately constraining the emission parameters for adequately simulating the volcanic SO$_2$ plume differs for each parameter."

Line 594-605: I agree that the horizontal extent of the emissions can influence the simulations. "...emissions are constrained to a single horizontal grid box in this study..." I know what you mean, but this reads wrong. You also applied column emissions and vertically gaussian distributed emissions, which do not inject into "one single grid box". I would change this to "...emissions are constrained to a single grid box or columns of gid boxes in this study..." or make this clearer in a different way (e.g. what is the difference between a horizontal and a vertical gid box?

) To me a grid box is a grid box... whether it is vertical or horizontal. "...leading to non-linearities in the model that diverge from reality..." This statement needs references. Why is this important? What non-linearities are you talking about? I recommend highlighting the impact on aerosol formation/microphysics from this artefact (e.g. Vattioni et al. 2024). In this paragraph you should also discuss the effect of the vertical and horizontal resolution of the model, since it is known that this can affect simulations of volcanic plumes. "This concentration can lead to lower $SO_2$ and aerosol mixing ratios in the mid- to long- term, as aerosols grow excessively large and subsequently sediment out of the stratosphere, as observed following the Nabro eruption." You provide an explanation for lower aerosol mixing ratios, but what would be the reason for differences in $SO_2$ mixing ratios? Also this sentence (and the whole paragraph) needs references, since you don't show this with your results.

We thank the reviewer for this remark, and apologize for being imprecise. We reformulated the paragraph according to the reviewer's suggestions, removed the part on the aerosols, and only concentrated on $SO_2$:

" However, column or point emissions come with inherent limitations as well. First, emissions are constrained to a single grid box or columns of grid boxes in this study, potentially resulting in localized and exaggerated mixing ratios of $SO_2$, when volcanic plumes span areas exceeding one grid box (Tilmes et al., 2023). This may lead to a depletion of the oxidants, and subsequently to a slower oxidation to $H_2SO_4$. While distributing emissions on multiple columns horizontally may mitigate this effect by defining multiple emission points per eruption (Tilmes et al., 2023), we did not explore this aspect in our analysis. However, in detailed studies of strong eruptions, we additionally recommend exploring the effects of emissions over multiple columns and an extended time period to avoid non-linearities due to very high concentrations (see also Sect. ...). "

Line 623-625: What about the importance of the stratospheric entry point?

We stated before in line 590: "Primarily, emitting an appropriate quantity of $SO_2$ at the correct altitude appears to be the most critical factor." We agree that we show, that the stratospheric entry point is also important in the evolution. However, especially from Fig. 7 of the submitted manuscript, it is obvious that the altitude and emitted mass have the largest impact. We extended the paragraph in the revised version as follows:

" This aligns with our conclusion that altitude and mass are the most crucial emission parameters, which were directly determined from the emission inventory and do not depend on the availability of IASI observations. However, we showed that the timing and geographical location of the stratospheric entry point can lead to additional uncertainties, that remain when no IASI observations are available. "

Line 634: "emission" is written twice.

We corrected that.

Line 691: Conclusions last paragraph: This paragraph should be put into future tense (and or conjunctive), since like it is written know one could think that this is already provided or underway.

[Figure]

Figure 4: Observations of the Nabro plume on selected dates in the first week after the Nabro eruption. **a***, $SO_2$ plume height retrievals from IASI satellite observations (Clarisse et al., 2014) during ascending (ASC) and descending (DESC) orbits **b***, Derived $SO_2$ columns from IASI (Clarisse et al., 2014) observations. **c***, Derived $SO_2$ columns from OMI (Li et al., 2020) observations. IASI $SO_2$ columns are calculated assuming that all $SO_2$ is centered at the retrieved plume height (top row), and we only display pixels where the retrieved $SO_2$ plume is detected in the stratosphere (above 14 km), whereas OMI displays the total column. Note that the timing of the observations does not coincide in general.

Changed to "These may include ..."

**Figure 1: It is not very helpful showing code in the main manuscript since this will not be helpful to most readers. If at all I would put this into a supplement.**

We moved the namelist setup to the supplement.

**Figure 2: From just looking at the figure caption it is not clear what the "plume" refers to: $SO_2$, Aerosol in general, ash or sulfuric acid aerosol? Maybe specify in the caption what the IASI satellite measures.**

Thank you for the remark. We included this information in the caption. We additionally merged Figures 2 and 3 of the manuscript (see Fig. 4 here for the new version)

**Figure 3: "amount" is not very specific. I would call the unit by its name ($SO_2$ column).**

Replaced here and also in other occurrences.

**Figure 4: The caption could be clearer. What do you mean by "shortly after"? If you compare observations to the altitude of the "maximum $SO_2$ mixing ratio", only one altitude should be displayed in your plot, since there is only one maximum in the vertical column, right? I am confused here. Maybe change the last sentence**

[Figure]

Figure 5: Zonally and 5-day averaged (July 14 - July 18, 2011) profile of northern hemispheric stratospheric $SO_2$ mixing ratios derived from the MIPAS satellite (**a**) is compared to the respective $SO_2$ mixing ratios simulated in the reference (**c**) and sensitivity simulations (**b,d-j**) for the same period (one month after the eruption of the Nabro volcano).

to: "In the simulations $SO_2$ was only injected into the stratosphere, except for mills_et_al".

We reformulated "shortly after the eruption (top) and three days later (bottom)" to "one day (top) and four days (bottom) after the initial eruption".
For each horizontal coordinate, there is one altitude referenced in the IASI dataset. From the model results, we also retrieve the altitude of maximum $SO_2$ per column. We reformulated to "... is compared with the altitude of maximum $SO_2$ mixing ratios for each vertical column from the sensitivity simulations ..." to clarify this.
For the last sentence, we used the suggestion from the reviewer.

**Figure 5: See comments on Figure 4. Why don't you compare to OMI as well?**

We adjusted the caption according to the previous comments. OMI includes observations from the troposphere and the stratosphere as it observes the total column, and the contributions cannot be distinguished. For this reason, we only compare to IASI, as we did not inject into the troposphere, and the focus of the study is on the stratosphere.

**Figure 6: Maybe replace "zonally" with "zonally averaged". And also "study" with "simulations". I would skip "approximately" or be more specific. Does the date provided refer to the 5-day average or to the date of the eruption? There is no space between the first and the second panel, and the black line covers the "0". Please correct this.**

We adapted the suggested adjustments, and clarified the time range. Additionally, we corrected the plot (see Fig. 5 here).

**Figure 7: The first sentence of the caption can be skipped or integrated into the second one. What is the unit of the x axis? The format mm/dd is not used universal (in Europe dd/mm is more common). Thus, I suggest writing Jun 15, Jul 1, Jul 15 and so one, to make this clear. Y-Axis label: It is "$SO_2$", not "SO2". Why don't you show the spatial correlation for 15/6? Concerning the "stratospheric cutoff altitude": Do you mean tropopause? If not, why don't you use the tropopause altitude? If yes, I would name the tropopause by its name. Why did you choose these altitudes? Did you check the sensitivity of your assumptions? Looking at the satellite data and your simulations, you can see that 3 days after the eruption a considerable amount of the plume is exactly around 30°N. Thus, slightly changing the "stratospheric cutoff altitude" might have an impact on the results shown**

[Figure]

Figure 6: **a,** Total northern hemispheric stratospheric SO$_2$ burden as observed from the MIPAS satellite (black dots) and from the sensitivity simulations (5-day averages). **b,** Spatial correlation in the latitude-altitude plane between the simulations and MIPAS observations, i. e. the spatial correlation between the first and all other panels in Fig. 5.

here. This could for example be done, by providing plots with 1km higher and lower "stratospheric cutoff altitudes".

We removed the first sentence, changed the format of the x axis label, and replaced "SO2" by "SO$_2$". We do not show the spatial correlation for the first two datapoints for two reasons. First, MIPAS only gives reliable data some time after the initial eruption, as described earlier in the manuscript (Line 355-359). Second, the 3D emissions are only applied ten days after the initial eruption. We could show the spatial correlation before, however, it would not tell us anything about the simulations. The updated figure can be found in Fig. 6.

We renamed "stratospheric cutoff altitude" to "lower integration limit" to avoid confusion in the revised manuscript. We did not use the tropopause here as the lower integration limit, because it is not necessarily coinciding in model and observations. To enable the comparison with observations, we used a universal lower integration limit for the calculation of the stratospheric SO$_2$ burden and the sAOD as defined here. This universal definition is now defined in the experimental setup section. To show that total sAOD and stratospheric SO$_2$ burden are sensitive to the lower integration limit, however the agreement is not, we now provide a study on the sensitivity to the lower integration limits in the supplement of the revised manuscript and here in Fig. 2 and 3.

Figure 8: To me it is very hard to see any difference between the middle row and the lower row. Maybe it makes more sense to show the difference between the middle row columns and the lower row columns to better display the improvement (if there is any).

We added the relative difference between the two simulations (see earlier reply and Fig. 1 here).

Figure 10 and 11: Same comment as on Figure 7. What does "stratospheric cutoff altitude mean" and how sensitive are results to this definition? Change to: "... using the EMAC model with the new *EVER* historic volcanic setup (red) and..." The axis label should read "SO2", not "SO$_2$"

We changed the caption according to the suggestions, and replaced "SO2" by "SO$_2$". We addressed the stratospheric cutoff altitude (lower integration limit) as outlined above. (see also Fig. 2 and 3)

Figures 11 and 12: To which wavelengths do the aerosol optical depths and extinctions refer to? This is crucial information which is missing.

We apologize for not providing the considered wavelength (750 nm), and provide it in the revised version.

**Reply to comments of RC2:**

The paper describes a new submodel within the MESSy framework for better simulation of mainly SO2 emissions from volcanic eruptions. Other potential applications are mentioned, such as emissions from wildfires, or volcanic degassing. In the study, the submodel is used to assess different distributions of SO2 emission distributions for two case studies. An explosive volcanic eruption (Nabro) and an effusive/surface emitting eruption (Kilauea). A setup for historical simulations from 2007 to 2011 is also presented. The model output is then discussed and compared with satellite retrievals.

The paper presents advances in modelling with the new submodel EVERv1.1 and is therefore fitting for GMD. EVER is a new tool to implement the emissions of SO2 from volcanic eruptions more flexibly and could be the basis for a more unified way to represent volcanic emissions in the future.

We thank the reviewer for the time to review our manuscript, the very constructive feedback, and overall positive evaluation. We report the comments (grey, bold) along with our replies (blue).

However, some restructuring of the text is needed to facilitate understanding. The manuscript is also very long and could be shortened considerably. I find the general structure of the presented manuscript very confusing. The methods used in this paper are generally stated but spread throughout the whole manuscript, which makes it difficult to read. I therefore recommend a more traditional structure with the introduction of the model, the different observations used in the analysis and the experiment description all in one dedicated chapter. Currently it reads more like the methods precede the respective result chapter and sometimes more information e.g. on the datasets is stated within the respective result sections. Also some information on the observational datasets is missing in the methodology. For example what gases are measured by what instrument. Some of this is explained later in the manuscript. There is also a section called "results" which would mean that everything before this is methodology? But this is not the case here. More detailed comments on this are below. This could also help making the paper more concise, since there are currently a few repetitions.

We applied the very helpful suggestions from the reviewer to our manuscript. First, we considerably shortened the manuscript. The main affected Sections are described below:

- The description of general impacts of volcanic eruptions were shortened in the introduction.

- We moved the namelist example for the new submodel (Fig. 1) and some of the detailed explanations to the supplement.

- We shortened the introduction of the Nabro and Kilauea volcano.

- We restructured the methods section, thereby avoiding repetitions.

- We summarized the satellite instruments in a structured table.

In addition, we generally restructured the methods and results section. Sect. 2, including the

presentation of the EVER submodel mostly remained in the same structure, only adding the description of the default namelist setup as an additional subsection. Sect. 3 now only focuses on the methods and the experimental setup, while Sect. 4 only shows the results of the three experiments. This results in the following new structure for Sect. 2-4:

2. New MESSy submodel for Explosive Volcanic ERuptions (EVER)

    2.1. Submodel description

        2.1.1. Primary aerosol emissions

        2.1.2. Vertical distributions

    2.2. Historic default namelist setup

3. Methods

    3.1. Observations

    3.2. Model setup and numerical experiments

        3.2.1. $SO_2$ in explosive volcanic eruptions — Nabro (2011)

        3.2.2. Multi-year simulation with the historic namelist setup

        3.2.3. $SO_2$ from degassing volcanoes — Kilauea (2018)

4. Results

    4.1. $SO_2$ in explosive volcanic eruptions — Nabro (2011)

    4.2. Evaluation of the historic default namelist setup in a multi-year simulation from 2008 to 2011 (stratospheric $SO_2$ burden and sAOD)

    4.3. $SO_2$ from degassing volcanoes — Kilauea (2018)

Moreover, we provided the different observational instruments in a table, covering the most important aspects, and provide information which information is used in which results section in the experimental setup section 3.2 (see later specific comment).

The results show how different set ups of the emissions can influence the SO2 plume. It is discussed thoroughly, how the different SO2 injections influence the plume evolution after an eruption and finally is able to show, how the new set-up improves this with respect to observations. It is mentioned, that the Asian Monsoon influences the plume evolution, "simultaneously probing the dynamics in the model". This is not followed up with the necessary sensitivity simulations or discussion on the implications of dynamics and performance of the model.

We agree with the reviewer, that we did not address this topic. Although we initially planned to study this in detail, we concluded it to be out of scope of the manuscript, and did not include it. Sadly, the original sentence was kept in the submitted manuscript. We have removed it in the revised manuscript.

Since the code is not currently available, the results are currently not fully reproducible. However, the description in the paper should allow for a similar model to be constructed. But it is stated, that this submodel is portable to other base

models, while it was only tested for three very specific modal models. Would this submodel not function with a different type of sectional model?

Indeed, we can unfortunately not make the code public at the moment. We are working on changing this situation in the future (see also reply from Patrick Jöckel to the Chief Editor Comment).
We tested the submodel with the three mentioned modal aerosol models available in the MESSy environment. In general, there are no restrictions for the coupled aerosol model, but there is currently no sectional aerosol submodel available within the MESSy framework. For the coupling to other model systems, an interface has to be constructed. We clarified this in the manuscript.

Some more previously published research on the topic should be considered. E.g. different injection scenarios for the Pinatubo eruption in various models by Quaglia et al. (2023).

We added a more technical review on previous model studies on volcanic eruptions in the introduction, including more references to volcanic emission inventories (e. g. Carn et al., 2017; Diehl et al., 2012; Mills et al., 2016; Neely III and Schmidt, 2016; Schallock et al., 2023) and specific model studies (e. g. Brodowsky et al., 2021; Quaglia et al., 2023; Schallock et al., 2023; Vattioni et al., 2024).

While the title sufficiently describes the presented model, the abstract is currently still missing some comment on the models performance but rather only summarizes the methods. Some statement on the performance of the new model should be added here.

We have included information about the performance and the results of the sensitivity studies in the abstract of the revised version:

"This work documents the operation of a new submodel for tracer emissions from Explosive Volcanic ERuptions (EVER v1.1), developed within the Modular Earth Submodel System (MESSy, version 2.55.1). EVER calculates additional tendencies of gaseous and aerosol tracers based on emission source parameters, aligned to specific sequences of volcanic eruptions or other atmospheric emission sources, allowing for the employment of various vertical emission profiles. We show that volcanic $SO_2$ plumes can be reasonably reproduced through EVER emissions in numerical simulations with the ECHAM/MESSy Atmospheric Chemistry Model (EMAC), using satellite observations of $SO_2$ columns and mixing ratios following the explosive eruption of the Nabro volcano (Eritrea) in 2011 and a degassing event of the Kilauea volcano (2018) in Kilauea. Previous volcanic studies showed large variability in stratospheric $SO_2$ burdens depending on the chosen volcanic emission databases and parameters. Sensitivity studies on $SO_2$ emissions from the Nabro volcano explore perturbations of the emission source parameters, revealing that emission altitude and the emitted mass above the tropopause are most important for the mid- to long-term evolution of stratospheric $SO_2$ plumes, while the correct timing and geographical location of the stratospheric entrance is crucial for the short-term plume evolution. We integrate information from a volcanic $SO_2$ emission inventory, additional satellite observations, and our findings from the sensitivity studies to establish a historical standard setup for volcanic eruptions impacting stratospheric $SO_2$ from 1990 to 2023, successfully evaluated with satellite observations of stratospheric $SO_2$ burden and aerosol optical properties. We advocate for this to be a standardized setup in all simulations within the MESSy framework concentrating on the upper troposphere and stratosphere in this period. Further potential applications of EVER involve studies on volcanic ash, wildfires, solar radiation modification, and atmospheric transport processes."

Also please check all abbreviations, some of the abbreviations are not explained while others are specified several times. Similarly, some of the sources don't conform to the GMD format.

Thank you for this remark. We checked all abbreviations and sources.

Throughout the manuscript there are several very technical explanations and formulas, code snippets etc. that would fit better in the appendix or supplementary material. Figure 1 does not fit in this manuscript. This is only an example of a namelist, not a result and should be kept in the supplementary material. Particularly the first figure should represent the main results of the paper.

As this is a technical paper, we wanted to provide all information, that is necessary to work with the submodel. Example namelists are commonly presented in other GMD papers. However, as both reviewers suggest to remove the namelist from the manuscript, we moved it to the supplement in the revised version. Additionally, we moved the optimization algorithm for the Kilauea study to the supplement.

**Line by line comments**

L20ff/L55ff: Both of these paragraphs introduce heterogeneous chemistry on aerosol surfaces and ozone chemistry

We removed the second paragraph (L55ff), shortening the introduction.

"Up to 800 volcanic eruptions" why is there a limit? Is it impossible to simulate 801 eruptions?

This is just a technical limitation (as a maximum number has to be defined for memory usage), that can be adjusted. We agree, that this was not properly formulated, and we removed this statement.

L150: This formula is very general and not new to this paper, it should therefore be moved to the supplements.

We agree and removed the formula.

Fig1: The namelists are already in the supplementary material. This is also just an example and not a result. I suggest removing this figure and referring to supplementary material

We moved the namelist to the supplement (see general comments). We want to keep the example namelist in the supplement, as it is more concise than the complete historic namelist setup.

L143: Unexplained acronym AER

We removed this in the main text as it was specific to the namelist, that was now moved to the supplement.

**L156: Why is this not possible?**

One of the purposes of the TREXP submodel were artificial, mostly passive tracer experiments. For that reason, new tracer initialisation was possible in TREXP. In EVER we focus on volcanic species, that are chemically active and already treated in our chemical mechanism. Thus, we believed that it is not necessary for this submodel to include the definition of new tracers. As this information is not very important for the manuscript, however, we removed it.

**L170ff: Is there a reference for how this is done in EVER 1.0?**

Unfortunately, there is no reference for EVER 1.0. EVER 1.1 was developed while preparing the manuscript, as we implemented some improvements and adjustments for more flexibility. For that reason, we wanted to provide information on this change here.

**L186ff: "and width" I find the term "width" misleading here as I understand that as spreading over several horizontal grid boxes. Or does this refer to the mass emitted? Vertical extent? If there is a fixed definition, please specify.**

We apologize if the description was unclear. The term width referred to the sigma of the vertical Gaussian distribution. The mass is vertically distributed following a Gaussian distribution with the given sigma. It is only emitted in one horizontal grid box. We reformulated the whole paragraph to:

"In this distribution, the mass follows a Gaussian-shaped vertical profile with mean altitude and sigma defined in the namelist emission entry. It can be confined to a vertical extent as for the **Uniform distribution**, truncating the tail and scaling accordingly. The emission amount in each grid cell is calculated by considering the fraction of the error function integrated from the bottom to the top of the grid cell (for the lowermost and uppermost grid cells, the minimum and maximum altitudes are used, respectively) relative to the integral of the error function across the entire confined vertical extent."

**L191ff: There are important microphysical implications for this, see e.g. in Fig. 3 in Tilmes et al. (2023)**

We thank the reviewer for this suggestion. We included the reference (Tilmes et al., 2023) and adjusted the paragraph as follows:

"Although each emission point emits within a single grid box or a column of grid boxes, multiple emission points can be defined to accurately reproduce the horizontal distribution of a single eruption. This is especially important for large volcanic plumes entering the stratosphere, exceeding the area of one grid box (Schallock et al., 2023). Tilmes et al. (2023) showed, that the emission of the Pinatubo $SO_2$ plume over a horizontal region, rather than in one single column, leads to a significantly improved agreement with observations. However, in this study we only used one emission point for each eruption."

**L201: What variables are nudged? What are the implications of nudging?**

We clarify this in the revised version: "Temperature, the logarithm of the surface pressure, divergence and vorticity are "nudged" (more details in Jöckel et al., 2006) towards meteorological reanalysis data (ERA5, Hersbach et al., 2020) from the European Centre for Medium-Range weather forecasts (ECMWF). In addition, we employ the QBO submodel to weakly nudge the simulations to QBO zonal wind observations between 10 and 90 hPa (Giorgetta et al., 2002)

Table 1: Summary of satellite instruments and the respective observed quantities used for the evaluation of model simulations with the EVER submodel. More technical details on the observations and the applied retrievals can be found in the provided references.

| Instrument | Observation method | Retrieved quant. | Opera-tion | Reference |
|---|---|---|---|---|
| Infrared Atmospheric Sounding Interferometer (IASI) | Thermal infrared emission from Earth (nadir) | $SO_2$ – column + altitude retrieval | 2006–now | Clarisse et al. (2014, 2012) |
| Ozone Monitoring Instrument (OMI) | UV & vis. solar backscatter radiation from Earth's surface (nadir) | $SO_2$ – total column | 2004–now | Li et al. (2020) |
| Michelson Interferometer for Passive Atmospheric Sounding (MIPAS) | Mid-infrared emission spectrometer (limb) | $SO_2$ – 3D (stratosphere) | 2002–2012 | Höpfner et al. (2015, 2013) |
| TROPOspheric Monitoring Instrument (TROPOMI) | As OMI, extended to UV, visible, near-IR & shortwave | $SO_2$ – column (troposphere) | 2017–now | Theys et al. (2017) |
| Optical, Spectroscopic and Infrared Remote Imaging System (OSIRIS) | Spectrally dispersed, scattered sunlight (limb) | **Aerosol extinction** – 3D (stratosph.) | 2001–now | Rieger et al. (2019) |

to avoid a phase drift."

**L209: What does simplified chemistry mean, is OH prescribed?**

The full mechanism for the simplified chemistry is provided in the supplement. The main difference for $SO_2$ is the direct conversion to $H_2SO_4$ on reaction with OH without intermediate products, such as $SO_3$. OH is not prescribed in any of the presented simulations. We discuss this in more detail in the revised manuscript:

"We use a simplified chemistry covering the basic tropospheric chemistry, including $O_3$, OH, NOx, NOy and basic sulfur chemistry (see supplement for details). Oxidation of $SO_2$ to $H_2SO_4$ is directly realized via reaction with OH, without producing any intermediates. We do not consider DMS and OCS here (see supplement), leading to a potential underestimation of background maritime $SO_2$ concentrations."

**L223ff: The following paragraphs lack some consistency, consider also summarizing some of the key properties of these measurements in a table. E.g. resolution, extent, what gases are measured, time when they were/are in operation. Just from reading this, I am not sure where these measurements are used later. Is it about SO2 or aerosol, tropospheric or stratospheric?**

Thank you for this suggestion. We summarised the key information in a table (see Table 1) in the revised manuscript, and give more information on which satellites are used for which purpose in the experimental setup.

**L313: "width of 2km" again I find the use of the word "width" a bit confusing, do you mean vertical extent?**

We reformulated to: "Column emission centred at altitude from the emission inventory minus 1 km (17 km); Gaussian vertical distribution with a sigma of 2 km, confined to the altitude range from 16 to 18 km; ..."

**L360ff: This information on what limitations the different satellite instruments/ retrievals have fits better in the methods section with the description of the in-**

struments/ datasets.

We agree. This part was moved to the experimental setup section (3.2) within the methods, and we only refer to it here.

L374: "AKM" was already defined previously, it is again defined several times throughout the manuscript.

We checked the manuscript for all occurrences, and made sure, that it is only defined once.

L376: "Vertical width"

We reformulated to: "The comparison between simulated and observed $SO_2$ distributions reveals some discrepancies, particularly in the vertical extent of the elevated $SO_2$ mixing ratios. After applying the AKM to the simulations, the $SO_2$ distributions exhibit a slightly larger vertical extent compared to the observations."

L473ff: This reads like it should be in the discussion/conclusion

We agree on that, and moved it (slightly adjusted) to the discussion.

L479ff: Parts of these section introduce more motivation for the study again, which belongs in the introduction. I also recommend combining the methodology with the ones in the other chapters.

We agree with the reviewer, and moved it to the new general methods section. We also generally restructured the methodology and the results (see reply to the general comments for the new structure).

L509: The title for this subsection is misleading, is everything before this methodology?

We summarized all results in one section, and separate them from the methods in the revised version to make the structure clearer (see reply to the general comments).

L595: What about availability of oxidants in these spatially confined plumes? There would also be some differences between simplified chemistry and a more sophisticated chemistry scheme

We agree, that there will be differences between the simplified and more sophisticated chemistry. However, in the discussed simulations, we only used the more sophisticated chemistry scheme. Regarding the availability of oxidants: If the plume is constrained to a smaller volume in the model, the volcanic $SO_2$ will also encounter relatively less OH overall, and the oxidation to $H_2SO_4$ will become slower. We shortly discuss this in the revised version:

"First, emissions are constrained to a single grid box or columns of grid boxes in this study, potentially resulting in localized and exaggerated mixing ratios of $SO_2$, when volcanic plumes span areas exceeding one grid box (Tilmes et al., 2023). This may lead to a depletion of the oxidants, and subsequently to a slower oxidation to $H_2SO_4$."

L634: "emission emission"

Fixed this repetition.

L639: How are eruptions after 2012 currently simulated?

After 2010 only climatological volcanoes of Diehl et al. (2012) are used for tropospheric degassing.

**L915-925: Check sources/format**

Thank you for the remark. We checked the format of the sources.

**References**

Brodowsky, C., T. Sukhodolov, A. Feinberg, M. Höpfner, T. Peter, A. Stenke, and E. Rozanov (2021). "Modeling the Sulfate Aerosol Evolution After Recent Moderate Volcanic Activity, 2008–2012". *Journal of Geophysical Research: Atmospheres* 126.23, e2021JD035472. DOI: `https://doi.org/10.1029/2021JD035472`.

Brühl, C., J. Lelieveld, P. J. Crutzen, and H. Tost (2012). "The role of carbonyl sulphide as a source of stratospheric sulphate aerosol and its impact on climate". *Atmospheric Chemistry and Physics* 12.3, pp. 1239–1253. DOI: `10.5194/acp-12-1239-2012`.

Brühl, C., J. Schallock, K. Klingmüller, C. Robert, C. Bingen, L. Clarisse, A. Heckel, P. North, and L. Rieger (2018). "Stratospheric aerosol radiative forcing simulated by the chemistry climate model EMAC using Aerosol CCI satellite data". *Atmospheric Chemistry and Physics* 18.17, pp. 12845–12857. DOI: `10.5194/acp-18-12845-2018`.

Carn, S., V. Fioletov, C. McLinden, C. Li, and N. Krotkov (2017). "A decade of global volcanic SO2 emissions measured from space". *Scientific reports* 7.1, p. 44095. DOI: `10.1038/srep44095`.

Clarisse, L., P.-F. Coheur, N. Theys, D. Hurtmans, and C. Clerbaux (2014). "The 2011 Nabro eruption, a SO$_2$ plume height analysis using IASI measurements". *Atmospheric Chemistry and Physics* 14.6, pp. 3095–3111. DOI: `10.5194/acp-14-3095-2014`.

Clarisse, L., D. Hurtmans, C. Clerbaux, J. Hadji-Lazaro, Y. Ngadi, and P.-F. Coheur (2012). "Retrieval of sulphur dioxide from the infrared atmospheric sounding interferometer (IASI)". *Atmospheric Measurement Techniques* 5.3, pp. 581–594. DOI: `10.5194/amt-5-581-2012`.

Diehl, T., A. Heil, M. Chin, X. Pan, D. Streets, M. Schultz, and S. Kinne (2012). "Anthropogenic, biomass burning, and volcanic emissions of black carbon, organic carbon, and SO$_2$ from 1980 to 2010 for hindcast model experiments". *Atmospheric Chemistry and Physics Discussions* 12, pp. 24895–24954. DOI: `10.5194/acpd-12-24895-2012`.

Giorgetta, M. A., E. Manzini, and E. Roeckner (2002). "Forcing of the quasi-biennial oscillation from a broad spectrum of atmospheric waves". *Geophysical Research Letters* 29.8, pp. 86-1-86–4. DOI: `https://doi.org/10.1029/2002GL014756`.

Günther, A., M. Höpfner, B.-M. Sinnhuber, S. Griessbach, T. Deshler, T. von Clarmann, and G. Stiller (2018). "MIPAS observations of volcanic sulfate aerosol and sulfur dioxide in the stratosphere". *Atmospheric Chemistry and Physics* 18.2, pp. 1217–1239. DOI: `10.5194/acp-18-1217-2018`.

Hersbach, H., B. Bell, P. Berrisford, S. Hirahara, A. Horányi, J. Muñoz-Sabater, J. Nicolas, C. Peubey, R. Radu, D. Schepers, A. Simmons, C. Soci, S. Abdalla, X. Abellan, G. Balsamo, P. Bechtold, G. Biavati, J. Bidlot, M. Bonavita, G. De Chiara, P. Dahlgren, D. Dee, M. Diamantakis, R. Dragani, J. Flemming, R. Forbes, M. Fuentes, A. Geer, L. Haimberger, S. Healy, R. J. Hogan, E. Hólm, M. Janisková, S. Keeley, P. Laloyaux, P. Lopez, C. Lupu, G. Radnoti, P. de Rosnay, I. Rozum, F. Vamborg, S. Villaume, and J.-N. Thépaut (2020). "The ERA5 global reanalysis". *Quarterly Journal of the Royal Meteorological Society* 146.730, pp. 1999–2049. DOI: `https://doi.org/10.1002/qj.3803`.

Hommel, R., C. Timmreck, M. A. Giorgetta, and H. F. Graf (2015). "Quasi-biennial oscillation of the tropical stratospheric aerosol layer". *Atmospheric Chemistry and Physics* 15.10, pp. 5557–5584. DOI: `10.5194/acp-15-5557-2015`.

Höpfner, M., C. D. Boone, B. Funke, N. Glatthor, U. Grabowski, A. Günther, S. Kellmann, M. Kiefer, A. Linden, S. Lossow, H. C. Pumphrey, W. G. Read, A. Roiger, G. Stiller, H. Schlager, T. von Clarmann, and K. Wissmüller (2015). "Sulfur dioxide (SO$_2$) from MIPAS

in the upper troposphere and lower stratosphere 2002–2012". *Atmospheric Chemistry and Physics* 15.12, pp. 7017–7037. DOI: `10.5194/acp-15-7017-2015`.

Höpfner, M., N. Glatthor, U. Grabowski, S. Kellmann, M. Kiefer, A. Linden, J. Orphal, G. Stiller, T. von Clarmann, B. Funke, and C. D. Boone (2013). "Sulfur dioxide ($SO_2$) as observed by MIPAS/Envisat: temporal development and spatial distribution at 15–45 km altitude". *Atmospheric Chemistry and Physics* 13.20, pp. 10405–10423. DOI: `10.5194/acp-13-10405-2013`.

Jöckel, P., A. Kerkweg, A. Pozzer, R. Sander, H. Tost, H. Riede, A. Baumgaertner, S. Gromov, and B. Kern (2010). "Development cycle 2 of the Modular Earth Submodel System (MESSy2)". *Geoscientific Model Development* 3.2, pp. 717–752. DOI: `10.5194/gmd-3-717-2010`.

Jöckel, P., H. Tost, A. Pozzer, C. Brühl, J. Buchholz, L. Ganzeveld, P. Hoor, A. Kerkweg, M. G. Lawrence, R. Sander, B. Steil, G. Stiller, M. Tanarhte, D. Taraborrelli, J. van Aardenne, and J. Lelieveld (2006). "The atmospheric chemistry general circulation model ECHAM5/MESSy1: consistent simulation of ozone from the surface to the mesosphere". *Atmospheric Chemistry and Physics* 6.12, pp. 5067–5104. DOI: `10.5194/acp-6-5067-2006`.

Jost, A. (2021). "Determining the $SO_2$ emission rates of the Kilauea volcano in Hawaii using S5P-TROPOMI satellite measurements". Bachelor's Thesis. Johannes Gutenberg Universität Mainz.

Klobas, J. E., D. M. Wilmouth, D. K. Weisenstein, J. G. Anderson, and R. J. Salawitch (2017). "Ozone depletion following future volcanic eruptions". *Geophysical Research Letters* 44.14, pp. 7490–7499. DOI: `10.1002/2017GL073972`.

Kremser, S., L. W. Thomason, M. von Hobe, M. Hermann, T. Deshler, C. Timmreck, M. Toohey, A. Stenke, J. P. Schwarz, R. Weigel, S. Fueglistaler, F. J. Prata, J.-P. Vernier, H. Schlager, J. E. Barnes, J.-C. Antuña-Marrero, D. Fairlie, M. Palm, E. Mahieu, J. Notholt, M. Rex, C. Bingen, F. Vanhellemont, A. Bourassa, J. M. C. Plane, D. Klocke, S. A. Carn, L. Clarisse, T. Trickl, R. Neely, A. D. James, L. Rieger, J. C. Wilson, and B. Meland (2016). "Stratospheric aerosol—Observations, processes, and impact on climate". *Reviews of Geophysics* 54.2, pp. 278–335. DOI: `10.1002/2015RG000511`.

Li, C., N. A. Krotkov, P. Leonard, and J. Joiner (2020). *OMI/Aura Sulphur Dioxide (SO2) Total Column 1-orbit L2 Swath 13x24 km V003, Goddard Earth Sciences Data and Information Services Center (GES DISC) [data set]*. DOI: `10.5067/Aura/OMI/DATA2022`.

Mills, M. J., A. Schmidt, R. Easter, S. Solomon, D. E. Kinnison, S. J. Ghan, R. R. Neely III, D. R. Marsh, A. Conley, C. G. Bardeen, and A. Gettelman (2016). "Global volcanic aerosol properties derived from emissions, 1990–2014, using CESM1(WACCM)". *Journal of Geophysical Research: Atmospheres* 121.5, pp. 2332–2348. DOI: `10.1002/2015JD024290`.

Neely III, R. and A. Schmidt (2016). *VolcanEESM: Global volcanic sulphur dioxide (SO2) emissions database from 1850 to present - Version 1.0 [Data set]*. DOI: `10.5285/76ebdc0b-0eed-4f70-b89e-55e606bcd568`.

Pöschl, U., R. von Kuhlmann, N. Poisson, and P. J. Crutzen (2000). "Development and intercomparison of condensed isoprene oxidation mechanisms for global atmospheric modeling". *Journal of Atmospheric Chemistry* 37.1, pp. 29–52. DOI: `10.1023/A:1006391009798`.

Pringle, K. J., H. Tost, S. Message, B. Steil, D. Giannadaki, A. Nenes, C. Fountoukis, P. Stier, E. Vignati, and J. Lelieveld (2010). "Description and evaluation of GMXe: a new aerosol submodel for global simulations (v1)". *Geoscientific Model Development* 3.2, pp. 391–412. DOI: `10.5194/gmd-3-391-2010`.

Quaglia, I., C. Timmreck, U. Niemeier, D. Visioni, G. Pitari, C. Brodowsky, C. Brühl, S. S. Dhomse, H. Franke, A. Laakso, G. W. Mann, E. Rozanov, and T. Sukhodolov (2023). "Interactive stratospheric aerosol models' response to different amounts and altitudes of $SO_2$ injection during the 1991 Pinatubo eruption". *Atmospheric Chemistry and Physics* 23.2, pp. 921–948. DOI: 10.5194/acp-23-921-2023.

Rieger, L. A., D. J. Zawada, A. E. Bourassa, and D. A. Degenstein (2019). "A Multiwavelength Retrieval Approach for Improved OSIRIS Aerosol Extinction Retrievals". *Journal of Geophysical Research: Atmospheres* 124.13, pp. 7286–7307. DOI: 10.1029/2018JD029897.

Ruder, S. (2016). "An overview of gradient descent optimization algorithms". *arXiv preprint arXiv:1609.04747*.

Sander, R., A. Baumgaertner, D. Cabrera-Perez, F. Frank, S. Gromov, J.-U. Grooß, H. Harder, V. Huijnen, P. Jöckel, V. A. Karydis, K. E. Niemeyer, A. Pozzer, H. Riede, M. G. Schultz, D. Taraborrelli, and S. Tauer (2019). "The community atmospheric chemistry box model CAABA/MECCA-4.0". *Geoscientific Model Development* 12.4, pp. 1365–1385. DOI: 10.5194/gmd-12-1365-2019.

Schallock, J., C. Brühl, C. Bingen, M. Höpfner, L. Rieger, and J. Lelieveld (2023). "Reconstructing volcanic radiative forcing since 1990, using a comprehensive emission inventory and spatially resolved sulfur injections from satellite data in a chemistry-climate model". *Atmospheric Chemistry and Physics* 23.2, pp. 1169–1207. DOI: 10.5194/acp-23-1169-2023.

Schmidt, A., M. J. Mills, S. Ghan, J. M. Gregory, R. P. Allan, T. Andrews, C. G. Bardeen, A. Conley, P. M. Forster, A. Gettelman, R. W. Portmann, S. Solomon, and O. B. Toon (2018). "Volcanic Radiative Forcing From 1979 to 2015". *Journal of Geophysical Research: Atmospheres* 123.22, pp. 12491–12508. DOI: 10.1029/2018JD028776.

Textor, C., H.-F. Graf, C. Timmreck, and A. Robock (2004). "Emissions from volcanoes". *Emissions of Atmospheric Trace Compounds*. Ed. by C. Granier, P. Artaxo, and C. E. Reeves. Dordrecht: Springer Netherlands, pp. 269–303. ISBN: 978-1-4020-2167-1. DOI: 10.1007/978-1-4020-2167-1_7.

Theys, N., I. De Smedt, H. Yu, T. Danckaert, J. van Gent, C. Hörmann, T. Wagner, P. Hedelt, H. Bauer, F. Romahn, M. Pedergnana, D. Loyola, and M. Van Roozendael (2017). "Sulfur dioxide retrievals from TROPOMI onboard Sentinel-5 Precursor: algorithm theoretical basis". *Atmospheric Measurement Techniques* 10.1, pp. 119–153. DOI: 10.5194/amt-10-119-2017.

Tie, X. and G. Brasseur (1995). "The response of stratospheric ozone to volcanic eruptions: Sensitivity to atmospheric chlorine loading". *Geophysical Research Letters* 22.22, pp. 3035–3038. DOI: 10.1029/95GL03057.

Tilmes, S., M. J. Mills, Y. Zhu, C. G. Bardeen, F. Vitt, P. Yu, D. Fillmore, X. Liu, B. Toon, and T. Deshler (2023). "Description and performance of a sectional aerosol microphysical model in the Community Earth System Model (CESM2)". *Geoscientific Model Development* 16.21, pp. 6087–6125. DOI: 10.5194/gmd-16-6087-2023.

Timmreck, C., G. W. Mann, V. Aquila, R. Hommel, L. A. Lee, A. Schmidt, C. Brühl, S. Carn, M. Chin, S. S. Dhomse, T. Diehl, J. M. English, M. J. Mills, R. Neely, J. Sheng, M. Toohey, and D. Weisenstein (2018). "The Interactive Stratospheric Aerosol Model Intercomparison Project (ISA-MIP): motivation and experimental design". *Geoscientific Model Development* 11.7, pp. 2581–2608. DOI: 10.5194/gmd-11-2581-2018.

Vattioni, S., A. Stenke, B. Luo, G. Chiodo, T. Sukhodolov, E. Wunderlin, and T. Peter (2024). "Importance of microphysical settings for climate forcing by stratospheric $SO_2$ injections as

modeled by SOCOL-AERv2". *Geoscientific Model Development* 17.10, pp. 4181–4197. DOI: 10.5194/gmd-17-4181-2024.

---

## Author Response (AR2)

**Topic editor decision: Publish subject to minor revisions (review by editor)**

We thank the editor for working through the manuscript again, and appreciate the decision for publication after minor revisions. We went through the comments carefully, and applied them in the manuscript. Below, we report the editor's comments (grey, bold) along with our replies (blue). We only report comments, where we add further explanation or (slightly) deviate from the editor's recommendations. All comments not listed below were implemented precisely as suggested.

**List of minor revisions (n.b. line numbers refer to the Track-Changes manuscript)**

1) Introduction, line 93: This is one of the new segments of text added, from reviewer 1's comments, and although this now refers to several of the papers documenting other volcanic SO2 emission inventories, the wording needs to be amended slightly, to first introduce the concept. A suggested change is to replace:

"Volcanic SO2 emission databases are the basis for the correct implementation of volcanic eruptions in atmospheric models…"

instead with

"Several volcanic SO2 emission databases have been developed, each providing a basis for implementing best-estimate source parameters for individual eruptions into global atmospheric models.…"

or similar text to better introduce the concept. It's quite common to colloquially refer to the individual volcanic emissions as "simulating the eruption" or similar, which in a sense it is, but better to be clear it's the emission not the eruption that's being simulated.

We included the slightly adjusted sentence:

"Several volcanic $SO_2$ emission databases have been developed, each providing a basis for implementing best-estimate [emission] parameters for individual eruptions into global atmospheric [chemistry] models."

2) Introduction, line 94: Please re-word this short sentence "Timmreck et al. (2018) recommend four different emission inventories" to be clear you mean within this intercomparison of interactive stratospheric aerosol models. Remember it's a relatively small proportion of the global atmospheric models that simulate stratospheric aerosol interactively, so this needs to be clear you mean for this co-ordinated intercomparison of global interactive stratospheric aerosol models.

We adjusted the sentence to:

"Timmreck et al. (2018) recommend four different emission inventories within the Interactive Stratospheric Aerosol Model Intercomparison Project (ISA-MIP)."

6) Introduction, line 128: Although GMD papers can include a good depth of

technical information, please change "a newly developed historic default namelist setup" to be clear you mean "historic eruptions" and to explain this re: automation of the previous manual approach that established the magnitude of each individual tracer adjustment. Suggested minor re-wording is to delete the word "default", which I think is meant re: this automated approach, to instead mention that automated functionality later in the sentence. I mean "...this newly developed historic eruption namelist setup then automates this functionality" or similar.

We thank the editor for the suggestion, and we rephrased it. As "for stratospheric volcanic eruptions from 1990 to 2023 ..." directly follows, we omitted "historic eruption" here:

"The new submodel along with a newly developed namelist setup for stratospheric volcanic eruptions from 1990 to 2023 (based on the emission inventory from Schallock et al. (2023) and Brühl et al. (2018)), that automates this functionality, is presented ... "

7) Introduction, line 151: I am not sure re: this sentence "and refined using observations from", is referring to something that was done within this study's activity, or if the present study only evaluates this. If the former please change "and refined using ..." to "and we further refine it here using...".

We actually refined it in this study, so we changed it accordingly.

8) Title of section 2 – "New MESSy submodel for Explosive Volcanic ERuptions (EVER)"

Although I get that the MESSy framework provides the basis for implementing the EVER sub-model, the wording of this section, putting the "MESSy" at the start of the title ("New MESSy submodel...") gives too much emphasis to this coupling, when the primarily functionality being documented is the EVER model for processing the individual eruptions to tracer increments.

Suggest to change the section 2 title to "New emissions submodel for Explosive Volcanic ERuptions (EVER)"

We changed it to:

"New submodel for emissions from Explosive Volcanic ERuptions (EVER)"

9) Section 2, lines 242 to 245 – The 2nd of these two sentences here seems the main information to begin this paragraph, or I am suggesting to integrate the point about the tracer tendencies into a re-worded version of the 2nd sentence here (then deleting the first). A suggested wording for this could be "The EVER module code converts the individual volcanic emissions into tracer increments..." (the text says "tendencies per second", but multiplying "tendency" by a unit of time gives an increment".

We reformulated to: "The EVER module code converts the individual emitted tracer masses into tracer increments, that are added to the total mixing ratios during runtime ..."

11) Section 2.2, line 312 – "...about 800 significant explosive volcanic eruptions", the wording "about 800" here seems too colloquial, for peer-reviewed journal paper. From the context of this para I wonder if could re-word also with combining with the preceding sentence, changing the start of the "Therefore, we established a

default namelist configuration...." to (rather than "default", say something like "The inventory we have established, we advocate to become a standard namelist configuration...", and then give the actual number of explosive injections included there.

We thank the editor for this suggestion. We agree that the word "default" is not optimal here. Consequently, we also changed it in the title of the section. However, we already advocate for the inclusion of the submodel three times (Abstract, Discussion and Conclusion), and we first want to present the setup here and evaluate it. Thus, we combined the sentences as follows:

"Therefore, we established a historic volcanic eruption namelist configuration for the EVER submodel based on the $SO_2$ emission inventory developed by Schallock et al. (2023), which we extended to the period from 1990 to 2023 (now encompassing 774 significant explosive volcanic eruptions), and refined with observations from the IASI satellite."

12) Section 2.2, lines 348-350 – ".. For eruptions occurring before 2007, or those not observed..." Please re-word "not observed", you mean within the "not observed by IASI", right? I think you mean in these cases it's done fully interactively, (i.e. via an explicit emission flux of SO2), rather than making a tracer mixing ratio adjustment. Then suggest to change "we utilize the" instead to "an explicit emission of SO2 is carried out at the geographical location, with the source parameters for the corresponding eruption...." or similar.

Indeed, we mean not observed by IASI. However, we treat these volcanic eruptions similar as the others, with the difference, that we do not update the timing and geographical location of the injection. To clarify this, we reformulated it as follows:

"$SO_2$ injections from volcanic eruptions occurring before 2007 or those not observed by IASI are introduced with the same vertical distribution, however, at the geographical location of the volcano and from 9:00 to 15:00 UTC on the date provided by the emission inventory from Schallock et al. (2023)."

13) Section 3.2.1, line 496 – "IASI faces limitations capturing the mid-term evolution of volcanic plumes with SO2 columns falling below the detection limit." I'm not sure I'm understanding the point being made here, but I guess you mean a limitation with the satellite-nudged tracer adjustment approach, right? I mean any satellite instrument would have that limitation (not only IASI), and its more a limitation of the approach to adjust the tracers, rather than the instrument itself.

Please re-word accordingly, but in fact since the requirement of the method is to realistically capture the enhancement to SO2 very-soon after the eruption, I don't see what the issue is here. I mean the model predicts the progression of the tracer concentrations, after the initial adjustment, and then doesn't require to continually adjust the tracers as the plume disperses.

I'm not sure if maybe the authors mean in the case of two eruptions occurring within a few days of each other (e.g. from two neighbouring volcanoes) or from an explosive eruption occuring near to a continuing significant high-altitude volcanic emission source?

But even in that case I'm not sure this "mid-term evolution" would be an issue, because there presumably are relatively few cases where that is the case, within this particular inventory's time-period anyway.

I'd suggest to potentially delete the lines 496 and 497 (including the ". Conversely "), and just have the paragraph begin "Observations from the ..." I'm not 100% sure of the distinction being pointed out here between IASI and MIPAS, the current wording is not so clear about that. Suggest to re-word into a combined sentence that then makes clearer the difference seeking to be communicated here.

We actually refer to the evaluation here. We use IASI for the evaluation of the short-term dispersion and development of the plume, however, this evaluation is not possible any more after the $SO_2$ columns fall below the detection limit of IASI. Thus, we use the MIPAS observations to qualitatively and quantitatively investigate the mid-term evolution of the plume. We agree, that this can be slightly confusing here. We rephrased and considerably shortened the text in combination with the preceding paragraph, including the changes from the following comment 14, as we agree that the explanations are too comprehensive and unclear:

" IASI observations of $SO_2$ columns and plume altitude are used for the qualitative evaluation of the short-term plume evolution (Sect. ...), by sampling simulated $SO_2$ columns at the time of the satellite's overpass using the SORBIT submodel (Jöckel et al., 2010). For the mid-term plume evolution and quantitative assessment of simulated $SO_2$ mixing ratios, we use three-dimensional observations from the MIPAS instrument[1] (Sect. ...). "

14) Section 3.2.1, line 501-502 – "Indeed, MIPAS observations become more reliable approximately three weeks after the initial eruption". There is no citation given here, and although the scientific writing style is generally good, this is an isolated instance of poor writing. Suggest to delete this or cite the specific case being referred to here (with citing a study that found this to be the case), as this is presumably relating to a study or report that analysed the observations through a specific post-eruption period? If the sentence is deleted, note the follow-on sentence "Its ability to provide..." then needs to be adapted accordingly, to state the instrument name again.

We agree, that this was poorly written and justified, and removed this part, as it is not necessary (see previous reply).

15) Section 3.2.1, line 504 – I'm not sure if a grammar-tool such as "grammarly" or similar has recommended to split the text into 2 sentences, but the 2nd of the two sentences here seems to be the 2nd half of a logical sentence that's then been split into a separate short sentence, that no longer makes sense. Although shorter sentences sometimes make for easier reading, within an academic peer-reviewed journal paper, that rule may not always apply, and the information being explained may require a longer sentence than might be seen to be better practice in a more informal communication, such as a webpage or similar primarily online-accessed info-resource.

Indeed, combining the two sentences improves the readability:
* * *
[1]MIPAS observations are not well-suited for the short-term evaluation due to potential rejection of plume air masses and saturation of spectral lines (Höpfner et al., 2015).

"For the comparison with MIPAS observations, we calculate the stratospheric $SO_2$ burden, applying fixed lower integration limits depending on latitude (16 km for 0-30°, 14 km for 30-60° and 12 km for 60-90°), as the model tropopause does not necessarily coincide with the observed tropopause."

16) Section 3.2.2, lines 511-521 — "Second, we performed a numerical simulation spanning from...." For a journal paper, paragraphs needs to be stand-alone (grammatically) and suggest to simply replace "Second, we performed a numerical simulation spanning January 2008 to December 2011 to evaluate..." with "To evaluate the implementation (within EMAC) of the new EVER historic eruption module ...." or similar.

We formulated to:

To evaluate the newly developed historic volcanic eruption setup for the EVER submodel (see Sect. ...), we performed an EMAC simulation spanning from January 2008 to December 2011, using the same model configuration as detailed in Sect. ... .

As the next section faces the same issue, we reformulated:

"Third, we evaluate the submodel's capability to simulate ..."

to:

"In addition to the evaluation of explosive volcanic eruptions, we evaluate the new submodel's capability to simulate ..."

17) Section 3.2.2, line 517 — "To additionally validate the simulation, we evaluate the extinction.." Further to the general comments re: Referee 1's rejection being primarily from requiring more aerosol evaluation, suggest here to explain the strategy of the study being to assessing the operation of the EVER module, via evaluating simulated SO2, but to show also this Figure for context within the climate impacts originating from simulated aerosol.

Suggest to replace the sentence with something like "Whilst this study focuses primarily to analyse simulated SO2, that being the emitted species handled by EVER, we present here also the EMAC simulated extinction and sAOD at 750nm.... ".

We reformulated the paragraph, and included the disclaimer, mentioned in the next comment:

" We use 3-dimensional observations from the MIPAS instrument to evaluate the total stratospheric $SO_2$ burden (Sect. ...) with the same lower integration limit as described above. Whilst this study primarily focuses on the evaluation of simulated $SO_2$, being the emitted species handled by EVER, we present here also the EMAC simulated extinction and sAOD at 750 nm, evaluated with observations from the OSIRIS instrument in the tropics (0-25°N, lower integration limit 16 km) and at higher northern latitudes (45-80°N, lower integration limit 12 km), as the tropopause altitude is fairly constant in these regions (Sect. ...). The additional evaluation of the simulated aerosol optical properties shall not evaluate the model's microphysics described below, but serve as an additional validation of the approach, for context within the climate impacts of the resulting stratospheric aerosol, and as a comparison to the simulation done by

Schallock et al. (2023), who used a similar setup. "

18) Section 3.2.2, lines 527-529 – further to comment 17) this one-sentence para I'm not sure if maybe is then explained in the suggested edit above. And then this comment could potentially be deleted.

This paragraph was integrated into the one mentioned in the previous comment (see above).

19) Section 3.2.3 title (line 530) and lines 531-544 – further to the differences in the chemistry regime with the Kilauea case, re: the short-lived duration of the volcanic SO2 oxidation product, suggest to hint at this with adding the word "tropospheric" before degassing, and maybe replace "volcanoes" with "case" – i.e. "SO2 from tropospheric degassing case – Kilauea (2018)"

I leave it to the authors whether to make that change, but the test here needs to give some recognition of the issue raised by reviewer 1, re: chemical sinks for SO2 also being important. The text added now mentions a difference between the SO2 oxidation products in the stratospheric chemistry and tropospheric chemistry schemes available within EMAC.

However, I'm not sure if the model actually switches between these 2 schemes, within the operation of the model integration here, i.e. do only gridboxes above the tropopause "see" this extra intermediate oxidation product in the stratospheric SO2 oxidation approach (sulphur trioxide). Are there two different chemistries being integrated within the same model run, or is the scheme applying the same chemical integration across the troposphere and stratosphere.

Obviously air is exchanged between the stratosphere and troposphere, and for eruptions emitting near the tropopause, and then if two chemistries are integrated, is the cut-off for the change at the tropopause, or some distance below the tropopause (e.g. considering the upper tropospheric air likely has more commonalities with the stratosphere, albeit considering the dehydration etc. re; the water vapour affecting the sulphur trioxide chemical-sink timescale.

Whichever approach is used, this needs to be set clearly within this section, and with the re-factoring of the text there is some space to set out a good depth of information to clarify the operation of the two SO2 oxidation schemes now mentioned in the approach (and where the intermediate product is resolved).

I must admit it wasn't 100% clear to me whether the Kilauea case used the tropospheric chemistry scheme because this operates only over the lower-atmosphere domain or so, as presumably this is one integration through the entire period with the same chemistry scheme, albeit presumably operating a hybrid approach of both schemes being integrated.

And whilst the explanation of the chemistry schemes added does partly address reviewer 1's concerns, it should also be clarified here, within this re-structured section 3, the methodological specifics of how the two chemistry schemes mentioned are combined and operate for this hindcast EMAC integration.

We thank the editor for the suggestion, and adjusted the title of the section accordingly.

Regarding the $SO_2$ chemistry: In the simulation for the degassing case, the simplified chemistry is applied globally (i. e. in the troposphere and the stratosphere), due to the high computational cost to run with the increased horizontal resolution. We clarify this in the revised version. We do not perform simulations with a hybrid chemistry, combining the two chemistry schemes, but either use one or the other.

" Simulations for the degassing case are performed at a horizontal resolution of T255 (approximately $50 \times 50$ km at the equator) and 31 model levels (up to an altitude of about 30 km) to capture the tropospheric transport. The timestep has to be decreased to 2.5 minutes to account for the increased horizontal resolution. We use a simplified chemistry globally in this simulation covering the basic tropospheric chemistry (as we do not focus on the stratosphere here), including $O_3$, OH, NOx, NOy and basic sulfur chemistry (see supplement for details), reducing the high computational cost due to the increased horizontal resolution and decreased timestep. Oxidation of $SO_2$ to $H_2SO_4$ is directly realized via reaction with OH in this simplified chemistry, without producing any intermediates. We do not consider DMS and OCS here (see supplement), leading to a potential underestimation of background maritime $SO_2$ concentrations. "

20) Section 3.2.3, lines 550-551 – Please re-word these 2 sentences, and again I think it might be easier to explain what's being communicated here within 1 longer sentence here. The 2nd sentence seems to be the 2nd half of the same point being explained, but has been cut-off part-way through making the point. Please also mention the specific satellite instrument here (S5P-TROPOMI), and clarify the specifics of the approach to fit to these observations. OK this is explained in Appendix B, but some summary-specifics should be given here so the reader can get the general gist of the issue being summarised..

We reformulated the paragraph, and summarised the optimization process more clearly:

" In addition, we perform a simulation with tuned emission rates, based on the results of the reference simulation with the emission rates from Jost (2021). For that purpose, we establish a linear relationship between $SO_2$ emissions and simulated $SO_2$ by fitting the implemented emission rates of the three preceding days to the resulting simulated $SO_2$ columns for each day in June 2018, and subsequently use this linear relationship to derive optimized emission rates resulting in the observed columns from TROPOMI, applying a stochastic gradient descent algorithm (e. g. Ruder, 2016). More details on the optimization can be found in Appendix B. "

21) Section 4.1.2, lines 788-790 – The approach to begin the para "So far, we focused on..." is to colloquial, suggest to re-word again to be clear the SO2 is the main focus of the manuscript's observational evaluation (that being the emitted species handled by the EVER module). That's what the 1st sentence says, but please expand the sentence slightly to communicate that more formally (less colloquially).

We reformulated to:

"The main focus of the observational evaluation of EVER is on the spatio-temporal evolution of volcanic $SO_2$ as the emitted species. However, once in the atmosphere, ..."

22) Section 4.1.2, line 837 – "Interestingly, in the sensitivity simulations..." Similar

to comment 21, please re-word for more formalised and objective language (delete
"Interestingly" – it's a subjective point, different readers will find different aspects
more or less interesting).

We removed "Interestingly, ..." here, and also at two other occurences

**References**

Brühl, C., J. Schallock, K. Klingmüller, C. Robert, C. Bingen, L. Clarisse, A. Heckel, P. North, and L. Rieger (2018). "Stratospheric aerosol radiative forcing simulated by the chemistry climate model EMAC using Aerosol CCI satellite data". *Atmospheric Chemistry and Physics* 18.17, pp. 12845–12857. DOI: 10.5194/acp-18-12845-2018.

Höpfner, M., C. D. Boone, B. Funke, N. Glatthor, U. Grabowski, A. Günther, S. Kellmann, M. Kiefer, A. Linden, S. Lossow, H. C. Pumphrey, W. G. Read, A. Roiger, G. Stiller, H. Schlager, T. von Clarmann, and K. Wissmüller (2015). "Sulfur dioxide ($SO_2$) from MIPAS in the upper troposphere and lower stratosphere 2002–2012". *Atmospheric Chemistry and Physics* 15.12, pp. 7017–7037. DOI: 10.5194/acp-15-7017-2015.

Jöckel, P., A. Kerkweg, A. Pozzer, R. Sander, H. Tost, H. Riede, A. Baumgaertner, S. Gromov, and B. Kern (2010). "Development cycle 2 of the Modular Earth Submodel System (MESSy2)". *Geoscientific Model Development* 3.2, pp. 717–752. DOI: 10.5194/gmd-3-717-2010.

Jost, A. (2021). "Determining the $SO_2$ emission rates of the Kilauea volcano in Hawaii using S5P-TROPOMI satellite measurements". Bachelor's Thesis. Johannes Gutenberg Universität Mainz.

Ruder, S. (2016). "An overview of gradient descent optimization algorithms". *arXiv preprint arXiv:1609.04747*.

Schallock, J., C. Brühl, C. Bingen, M. Höpfner, L. Rieger, and J. Lelieveld (2023). "Reconstructing volcanic radiative forcing since 1990, using a comprehensive emission inventory and spatially resolved sulfur injections from satellite data in a chemistry-climate model". *Atmospheric Chemistry and Physics* 23.2, pp. 1169–1207. DOI: 10.5194/acp-23-1169-2023.

Timmreck, C., G. W. Mann, V. Aquila, R. Hommel, L. A. Lee, A. Schmidt, C. Brühl, S. Carn, M. Chin, S. S. Dhomse, T. Diehl, J. M. English, M. J. Mills, R. Neely, J. Sheng, M. Toohey, and D. Weisenstein (2018). "The Interactive Stratospheric Aerosol Model Intercomparison Project (ISA-MIP): motivation and experimental design". *Geoscientific Model Development* 11.7, pp. 2581–2608. DOI: 10.5194/gmd-11-2581-2018.

---

## Author Response (AR3)

**GMD 2nd post-review TE review of "Simulating volcanic emissions in MESSy – New submodel for Explosive Volcanic ERuptions (EVER v1.1)" by Matthias Kohl, Christoph Bruehl, Jennifer Shallock et al.)**

We thank the editor for the additional review. We went through the comments carefully, and applied them in the manuscript. Below, we report the editor's comments (grey, bold) along with our replies (blue). We only report comments, where we add further explanation or (slightly) deviate from the editor's recommendations. All comments not listed below were implemented precisely as suggested.

**Further minor revisions (line numbers from Track Changes MS)**

**1) Lines 582-583, and follow-on paragraphs of discussion on lines 584-600.**

**I don't get the point being made on lines 582-583, the manuscript arguing there that emissions constrained to a single gridbox tend to "potentially result in localized and exaggerated mixing ratios of SO2".**

**I can see there is a follow-on point referring to "when volcanic plumes span areas exceeding one gridbox", but the logic here was unclear to me.**

**I'd argue that point sources tend more to more likely be under-representing the local (to the volcano) SO2 mixing ratio, within grid-box average values.**

**For coarse-spatial-scale global models, there is the issue of whether the model can represent regimes where a depletion of oxidants can affect predicted sulphate production (for example), and more generally variations between the concentrated emitted species (at the plume-scale) viz-a-viz the larger-scale volcanic enhancement in the model, that determining the magnitude predicted impacts.**

**There is quite a good depth of Discussion already within section 5, and suggest to re-word the main points here into just 1 paragraph, rather than the 3 paragraphs here on lines 584 to 600.**

We wanted to refer to the following scenario here: Stratospheric plumes of strong volcanic eruptions can cover several horizontal columns in reality, e.g. Hunga Tonga (2022). When confined to one column, this may lead to overestimated local $SO_2$ mixing ratios and non-linearities. We agree, that there is also the effect of underestimating local $SO_2$ mixing ratios when averaged over grid boxes, however, this cannot be solved within fixed horizontal resolution. We combined everything into one paragraph, and omitted the addressed formulation to avoid misunderstandings:

" However, column or point emissions come with inherent limitations as well. First, emissions are constrained to a single grid box or columns of grid boxes in this study. In detailed studies of strong eruptions, we additionally recommend exploring the effects of emissions over multiple columns and an extended time period to avoid non-linearities due to very high local concentrations (see also Sect. ...). Second, volcanic activity typically extends beyond a single day, with $SO_2$ emissions occurring over prolonged periods, occasionally reaching the stratosphere

(see Sect. ... for the Nabro eruption). While related discrepancies dissipated in the mid-term for the Nabro volcano, this may not be necessarily the case for other volcanic eruptions. Third, the exact timing and geographical location of the stratospheric entry point cannot always be accurately estimated. Our sensitivity analyses uncovered short- and mid-term disparities when such information is lacking. "

2) Lines 322-325 – Please revise this new sentence added here "shall not evaluate the microphysics". Further to the main comments above, since the topic of this manuscript is re: the emissions sub-model, there should be no expectation of requiring to validate the simulated aerosol. What I suggest here, is to refer to a "self-consistency check" for the model, in the sense that global models may tend to focus mainly on evaluating predicted aerosol, and may be less concerned with accurately predicting volcanic SO2. I realise that's perhaps what you mean by "additional evaluation", but since this study is not a model evaluation paper (in that sense) better to be clear this is more a case of analysing the aerosol predicted from the validated SO2, than it is validating the predicted aerosol.

Perhaps that is what you meant by "evaluation", but again, given reviewer 2's comments, suggest to word this more towards a "self-consistency check" for predicted aerosol (or similar alternative label than evaluation or validation).

We added this disclaimer as a response to reviewer 1, who was expecting a more proper evaluation and study of the model's microphysics. We agree with the editor, that this should not be expected here. We reformulated to:

" The additional evaluation of the simulated aerosol optical properties serves as a self-consistency check for the model, for context within the climate impacts of the resulting stratospheric aerosol, and as a comparison to the simulation done by Schallock et al. (2023), who used a similar setup. "

3) Lines 204 to 209 – Need to explain re: the default emissions-altitude, where IASI heights are used

The revised manuscript now refers to "with the same vertical distribution", but it's actually not been explained what reference-altitude is given in these cases where IASI can't observe the plume-top height.

From checking the file in the Supplement, I see the default emissions-altitude is 15km, but this is not stated currently within the manuscript (to my reading). Please add "at 15km" at the end of line 205, and also re-wording the earlier part of the revised sentence (re: IASI and "or those not observed") as this is still slightly unclear, and should be worded more concisely.

Suggest "For eruptions before 2007, or other volcanic SO2 plumes not observable from IASI, we distribute the column SO2 at 15km, with the Gaussian x-km vertical dispersion." (or similar).

Re: the latter, the text "all injections optimized using the IASI observations are marked accordingly" needs to give an indication of how many eruptions are where this further optimization from IASI. You can also delete "on the date provided by the emissions inventory from Schallock et al. (2023)", as that's implicit.

Also, again from checking the "ever_historic_stratVolcanoes" namelist, I saw the text-flag "IASI optimised" is given for eruptions 544 to 548 (2019 Raikoke), 550 (2019 Ulawun) and 568 (2020 Taal), and then please also add to note "for example 2019 Raikoke, 2019 Ulawun, 2020 Taal" where a revised observed plume-height has been found from the original inventory (Schallock et al., 2023).

We do not use the IASI top plume altitude here, and we also do not use a default altitude of 15 km. In line 200-202, we specify the vertical distribution and altitude we use:

"The $SO_2$ mass is then distributed vertically in a Gaussian profile centred 1 km below the maximum altitude (sigma of 2 km, confined to the vertical extent of the maximum plume altitude down 2 km, truncating the Gaussian distribution at $\sigma/2$) recorded in the emission inventory, over 6 hours around the identified date and time of peak mixing ratio as default."

and before:

"From this analysis, we extract the space-time point exhibiting the maximum stratospheric $SO_2$ mixing ratios observed by IASI as the optimal estimate for both, timing and geographical location for injecting the plume into the stratosphere."

We only use the (horizontal) geographical location and timing from IASI, and the plume altitude from the emission inventory from Schallock et al. (2023). We apologize that this was misleading, as we did not specify "horizontal" explicitly. We slightly reformulated to clarify:

" ... From this analysis, we extract the horizontal space-time point exhibiting the maximum stratospheric $SO_2$ mixing ratios observed by IASI as the optimal estimate for both, timing and geographical location for injecting the plume into the stratosphere.

The $SO_2$ mass is then distributed vertically in a Gaussian profile centred 1 km below the maximum altitude (sigma of 2 km, confined to the vertical extent of the maximum plume altitude down 2 km, truncating the Gaussian distribution at $\sigma/2$) provided in the emission inventory from Schallock et al. (2023), at the horizontal geographical location derived from IASI observations, over 6 hours around the identified date and time of peak mixing ratio. ... "

Regarding the namelist in the supplement, we reformulated as follows:

"The 54 strong injections with optimized horizontal geographical location and timing of the stratospheric entry point (e. g. Raikoke 2019, Ulawun 2019, Taal 2020) are marked accordingly."

9) Lines 265-267 – The wording here needs reducing: "To comprehensively evaluate volcanic SO2 emissions with the EVER submodel and the impact of simplifications and adjustments to emissions data, and we conducted a reference simulations along with a series of sensitivity simulations of stratospheric SO2".

Suggest to simplify to "For the Nabro eruption, we carried out a series of EMAC simulations to assess the sensitivity to difference emissions source parameters".

Putting "Nabro" earlier in the sentence so the reader can realise these sensitivity runs are for this one particular eruption.

The subject of the manuscript is EVER, so that doesn't need to be stated, but

since the model is not mentioned at any point in section 3.2.1, do need to specify EMAC there.

The "We applied a horizontal resolution..." needs to be impersonal tense, to "These simulations applied a horizontal resolution..." or similar.

10) Line 267 — Change "We applied a horizontal resolution of T63..." to "For the stratospheric injecting case, the simulations were at T63 horizontal resolution

Depending on how change 8 specifies the model resolutions in section 3.2, this sentence could be reduced, but actually I think good here to re-iterate as many readers may go straight to read this subsection.

We combined both comments and reformulated to:

"For the Nabro eruption, we carried out a series of EMAC simulations to assess the sensitivity of stratospheric $SO_2$ burdens to varying emissions source parameters. The simulations are performed at T63 horizontal resolution (approx. $190 \times 190$ km at the equator), with 90 vertical levels up to 0.01 hPa and a model timestep of 8 minutes."

11) Line 270 — The text here says "MIM1" but shoudn't this be simply "MIM"?

We actually mean MIM1 here. This is the version 1 of MIM (we state this now clearly in the manuscript) based on Jöckel et al. (2016) and Pöschl et al. (2000). There is also a version MIM2 (Taraborrelli et al., 2009). For that reason, the differentiation is necessary. However, we switched the citation from Jöckel et al. (2006) to Jöckel et al. (2016), as it is explicitly called MIM1 there as well (including a comprehensive evaluation).

13) Line 305 — Change "fixed lower integration limits" to "fixed lower plume-altitude"

We actually do not refer to the lower plume-altitude here. The plume altitude distribution is given above. The lower integration limit is only used her for the calculation of stratospheric $SO_2$ burden and sAOD, as lower altitude, from which we integrate/sum over the $SO_2$ mass and the aerosol extinction. This is only the definition of the $SO_2$ burden and sAOD here. We slightly adjusted the wording in the revised manuscript to clarify this.

18) Lines 337 – Change "In addition to the evaluation of explosive volcanic eruptions..." (remember it's not the eruptions being evaluated here) to "As well as evaluating emissions from explosive eruptions..." .

And then delete "following" later in the sentence so it's focused to "analysing a series of"

To avoid repetition of "evaluate", we reformulated to:

"In addition to emissions from explosive volcanic eruptions, we evaluate the new submodel's capability to simulate emissions of degassing volcanoes by analysing a series of eruptive fissures ..."

20) Line 345 – change "reducing the high computational cost due to the increased horizontal resolution and decreased timestep" to "with a decreased timestep to reduce the computational cost at this increased horizontal resolution".

We actually had to decrease the timestep to 2.5 minutes to account for the Courant-Friedrichs-Lewy criterion, and this additionally increases the computational cost. We reformulated the paragraph in combination with the previous comment to:

" Simulations for the degassing case are performed at a horizontal resolution of T255 (approximately $50 \times 50$ km at the equator) and 31 model levels (up to an altitude of about 30 km) to capture the tropospheric transport. For these simulations, we use a simpler chemical mechanism (compared to the T63 simulations) in the global model covering the basic tropospheric chemistry (as we do not focus on the stratosphere here), including $O_3$, OH, NOx, NOy and basic sulfur chemistry (see supplement for details), to reduce the computational cost at this increased horizontal resolution and decreased timestep of 2.5 minutes. ... "

22) Lines 362-363 – Add a full-stop after "simulated SO2 columns" and resume with a new sentence "For each day in...", then being able to add "(the first month post-eruption)" to clarify why June 2018. Also replace "and subsequently use this linear relationship to derive" with "a linear relationship is used to derive".

Actually, putting a full stop here would change the meaning of the sentence. The fit is performed for data at each day in June 2018, so we changed "for each day" to "at each day" to make this clearer. June 2018 is chosen as the emissions were most intense in that month. We added "(the month of most intense $SO_2$ emissions)" here.

**References**

Jöckel, P., H. Tost, A. Pozzer, C. Brühl, J. Buchholz, L. Ganzeveld, P. Hoor, A. Kerkweg, M. G. Lawrence, R. Sander, B. Steil, G. Stiller, M. Tanarhte, D. Taraborrelli, J. van Aardenne, and J. Lelieveld (2006). "The atmospheric chemistry general circulation model ECHAM5/MESSy1: consistent simulation of ozone from the surface to the mesosphere". *Atmospheric Chemistry and Physics* 6.12, pp. 5067–5104. DOI: `10.5194/acp-6-5067-2006`.

Jöckel, P., H. Tost, A. Pozzer, M. Kunze, O. Kirner, C. A. M. Brenninkmeijer, S. Brinkop, D. S. Cai, C. Dyroff, J. Eckstein, F. Frank, H. Garny, K.-D. Gottschaldt, P. Graf, V. Grewe, A. Kerkweg, B. Kern, S. Matthes, M. Mertens, S. Meul, M. Neumaier, M. Nützel, S. Oberländer-Hayn, R. Ruhnke, T. Runde, R. Sander, D. Scharffe, and A. Zahn (2016). "Earth System Chemistry integrated Modelling (ESCiMo) with the Modular Earth Submodel System (MESSy) version 2.51". *Geoscientific Model Development* 9.3, pp. 1153–1200. DOI: `10.5194/gmd-9-1153-2016`.

Pöschl, U., R. von Kuhlmann, N. Poisson, and P. J. Crutzen (2000). "Development and intercomparison of condensed isoprene oxidation mechanisms for global atmospheric modeling". *Journal of Atmospheric Chemistry* 37.1, pp. 29–52. DOI: `10.1023/A:1006391009798`.

Schallock, J., C. Brühl, C. Bingen, M. Höpfner, L. Rieger, and J. Lelieveld (2023). "Reconstructing volcanic radiative forcing since 1990, using a comprehensive emission inventory and spatially resolved sulfur injections from satellite data in a chemistry-climate model". *Atmospheric Chemistry and Physics* 23.2, pp. 1169–1207. DOI: `10.5194/acp-23-1169-2023`.

Taraborrelli, D., M. G. Lawrence, T. M. Butler, R. Sander, and J. Lelieveld (2009). "Mainz Isoprene Mechanism 2 (MIM2): an isoprene oxidation mechanism for regional and global atmospheric modelling". *Atmospheric Chemistry and Physics* 9.8, pp. 2751–2777. DOI: `10.5194/acp-9-2751-2009`.